# Statistical Inference with M-Estimators on Adaptively Collected Data

**Kelly W. Zhang**
Department of Computer Science
Harvard University
`kellywzhang@seas.harvard.edu`

**Lucas Janson**
Departments of Statistics
Harvard University
`ljanson@fas.harvard.edu`

**Susan A. Murphy**
Departments of Statistics and Computer Science
Harvard University
`samurphy@fas.harvard.edu`

## Abstract

Bandit algorithms are increasingly used in real-world sequential decision-making problems. Associated with this is an increased desire to be able to use the resulting datasets to answer scientific questions like: Did one type of ad lead to more purchases? In which contexts is a mobile health intervention effective? However, classical statistical approaches fail to provide valid confidence intervals when used with data collected with bandit algorithms. Alternative methods have recently been developed for simple models (e.g., comparison of means). Yet there is a lack of general methods for conducting statistical inference using more complex models on data collected with (contextual) bandit algorithms; for example, current methods cannot be used for valid inference on parameters in a logistic regression model for a binary reward. In this work, we develop theory justifying the use of M-estimators—which includes estimators based on empirical risk minimization as well as maximum likelihood—on data collected with adaptive algorithms, including (contextual) bandit algorithms. Specifically, we show that M-estimators, modified with particular adaptive weights, can be used to construct asymptotically valid confidence regions for a variety of inferential targets.

## 1 Introduction

Due to the need for interventions that are personalized to users, (contextual) bandit algorithms are increasingly used to address sequential decision making problems in health-care [Yom-Tov et al., 2017, Liao et al., 2020], online education [Liu et al., 2014, Shaikh et al., 2019], and public policy [Kasy and Sautmann, 2021, Caria et al., 2020]. Contextual bandits personalize, that is, minimize regret, by learning to choose the best intervention in each context, i.e., the action that leads to the greatest expected reward. Besides the goal of regret minimization, another critical goal in these real-world problems is to be able to use the resulting data collected by bandit algorithms to advance scientific knowledge [Liu et al., 2014, Erraqabi et al., 2017]. By scientific knowledge, we mean information gained by using the data to conduct a variety of statistical analyses, including confidence interval construction and hypothesis testing. **While regret minimization is a *within*-experiment learning objective, gaining scientific knowledge from the resulting adaptively collected data is a *between*-experiment learning objective**, which ultimately helps with regret minimization between deployments of bandit algorithms. Note that the data collected by bandit algorithms are *adaptively collected* because previously observed contexts, actions, and rewards are used to inform what actions to select in future timesteps.

35th Conference on Neural Information Processing Systems (NeurIPS 2021).

There are a variety of between-experiment learning questions encountered in real-life applications of bandit algorithms. For example, in real-life sequential decision-making problems there are often a number of additional scientifically interesting outcomes besides the reward that are collected during the experiment. In the online advertising setting, the reward might be whether an ad is clicked on, but one may be interested in the outcome of amount of money spent or the subsequent time spent on the advertiser's website. If it was found that an ad had high click-through rate, but low amounts of money was spent after clicking on the ad, one may redesign the reward used in the next bandit experiment. One type of statistical analysis would be to construct confidence intervals for the relative effect of the actions on multiple outcomes (in addition to the reward) conditional on the context. Furthermore, due to engineering and practical limitations, some of the variables that might be useful as context are often not accessible to the bandit algorithm online. If after-study analyses find some such contextual variables to have sufficiently strong influence on the relative usefulness of an action, this might lead investigators to ensure these variables are accessible to the bandit algorithm in the next experiment.

As discussed above, we can gain scientific knowledge from data collected with (contextual) bandit algorithms by constructing confidence intervals and performing hypothesis tests for unknown quantities such as the expected outcome for different actions in various contexts. Unfortunately, standard statistical methods developed for i.i.d. data fail to provide valid inference when applied to data collected with common bandit algorithms. For example, assuming the sample mean of rewards for an arm is approximately normal can lead to unreliable confidence intervals and inflated type-1 error; see Section 3.1 for an illustration. Recently statistical inference methods have been developed for data collected using bandit algorithms [Hadad et al., 2019, Deshpande et al., 2018, Zhang et al., 2020]; however, these methods are limited to inference for parameters of simple models. There is a lack of general statistical inference methods for data collected with (contextual) bandit algorithms in more complex data-analytic settings, including parameters in non-linear models for outcomes; for example, there are currently no methods for constructing valid confidence intervals for the parameters of a logistic regression model for binary outcomes or for constructing confidence intervals based on robust estimators like minimizers of the Huber loss function.

In this work we show that a wide variety of estimators which are frequently used both in science and industry on i.i.d. data, namely, M-estimators [Van der Vaart, 2000], can be used to conduct valid inference on data collected with (contextual) bandit algorithms when adjusted with particular adaptive weights, i.e., weights that are a function of previously collected data. Different forms of adaptive weights are used by existing methods for simple models [Deshpande et al., 2018, Hadad et al., 2019, Zhang et al., 2020]. Our work is a step towards developing a general framework for statistical inference on data collected with adaptive algorithms, including (contextual) bandit algorithms.

## 2   Problem Formulation

We assume that the data we have after running a contextual bandit algorithm is comprised of contexts $\{X_t\}_{t=1}^T$, actions $\{A_t\}_{t=1}^T$, and primary outcomes $\{Y_t\}_{t=1}^T$. $T$ is deterministic and known. We assume that rewards are a deterministic function of the primary outcomes, i.e., $R_t = f(Y_t)$ for some known function $f$. We are interested in constructing confidence regions for the parameters of the conditional distribution of $Y_t$ given $(X_t, A_t)$. Below we consider $T \to \infty$ in order to derive the asymptotic distributions of estimators and construct asymptotically valid confidence intervals. We allow the action space $\mathcal{A}$ to be finite or infinite. We use potential outcome notation [Imbens and Rubin, 2015] and let $\{Y_t(a) : a \in \mathcal{A}\}$ denote the potential outcomes of the primary outcome and let $Y_t := Y_t(A_t)$ be the observed outcome. We assume a stochastic contextual bandit environment in which $\{X_t, Y_t(a) : a \in \mathcal{A}\} \overset{i.i.d.}{\sim} \mathcal{P} \in \mathbf{P}$ for $t \in [1 : T]$; the contextual bandit environment distribution $\mathcal{P}$ is in a space of possible environment distributions $\mathbf{P}$. We define the history $\mathcal{H}_t := \{X_{t'}, A_{t'}, Y_{t'}\}_{t'=1}^t$ for $t \geq 1$ and $\mathcal{H}_0 := \emptyset$. Actions $A_t \in \mathcal{A}$ are selected according to policies $\pi := \{\pi_t\}_{t \geq 1}$, which define action selection probabilities $\pi_t(A_t, X_t, \mathcal{H}_{t-1}) := \mathbb{P}(A_t | \mathcal{H}_{t-1}, X_t)$. Even though the potential outcomes are i.i.d., the *observed* data $\{X_t, A_t, Y_t\}_{t=1}^T$ are *not* because the actions are selected using policies $\pi_t$ which are a function of past data, $\mathcal{H}_{t-1}$. Non-independence of observations is a key property of adaptively collected data.

We are interested in constructing confidence regions for some unknown $\theta^*(\mathcal{P}) \in \Theta \subset \mathbb{R}^d$, which is a parameter of the conditional distribution of $Y_t$ given $(X_t, A_t)$. This work focuses on the setting in which we have a well-specified model for $Y_t$. Specifically, we assume that $\theta^*(\mathcal{P})$ is a conditionally

maximizing value of criterion $m_\theta$, i.e., for all $\mathcal{P} \in \mathbf{P}$,

$$\theta^*(\mathcal{P}) \in \underset{\theta \in \Theta}{\operatorname{argmax}} \, \mathbb{E}_\mathcal{P}\left[m_\theta(Y_t, X_t, A_t)|X_t, A_t\right] \quad \text{w.p. 1.} \tag{1}$$

Note that $\theta^*(\mathcal{P})$ does not depend on $(X_t, A_t)$ and it is an implicit modelling assumption that such a $\theta^*(\mathcal{P})$ exists for a given $m_\theta$. Note that this formulation includes semi-parametric models, e.g., the model could constrain the conditional mean of $Y_t$ to be linear in some function of the actions and context, but allow the residuals to follow any mean-zero distribution, including ones that depend on the actions and/or contexts.

To estimate $\theta^*(\mathcal{P})$, we build on M-estimation [Huber, 1992], which classically selects the estimator $\hat{\theta}$ to be the $\theta \in \Theta$ that maximizes the empirical analogue of Equation (1):

$$\hat{\theta}_T := \underset{\theta \in \Theta}{\operatorname{argmax}} \, \frac{1}{T}\sum_{t=1}^{T} m_\theta(Y_t, X_t, A_t). \tag{2}$$

For example, in a classical linear regression setting with $|\mathcal{A}| < \infty$ actions, a natural choice for $m_\theta$ is the negative of the squared loss function, $m_\theta(Y_t, X_t, A_t) = -(Y_t - X_t^\top \theta_{A_t})^2$. When $Y_t$ is binary, a natural choice is instead the negative log-likelihood function for a logistic regression model, i.e., $m_\theta(Y_t, X_t, A_t) = -[Y_t X_t^\top \theta_{A_t} - \log(1 + \exp(X_t^\top \theta_{A_t}))]$. More generally, $m_\theta$ is commonly chosen to be a log-likelihood function or the negative of a robust loss function such as the Huber loss. If the data, $\{X_t, A_t, Y_t\}_{t=1}^T$, were independent across time, classical approaches could be used to prove the consistency and asymptotic normality of M-estimators [Van der Vaart, 2000]. However, on data collected with bandit algorithms, standard M-estimators like the ordinary least-squares estimator fail to provide valid confidence intervals [Hadad et al., 2019, Deshpande et al., 2018, Zhang et al., 2020]. In this work, we show that M-estimators can still be used to provide valid statistical inference on adaptively collected data when adjusted with well-chosen adaptive weights.

## 3  Adaptively Weighted M-Estimators

We consider a weighted M-estimating criteria with adaptive weights $W_t \in \sigma(\mathcal{H}_{t-1}, X_t, A_t)$ given by $W_t = \sqrt{\frac{\pi_t^{\text{sta}}(A_t, X_t)}{\pi_t(A_t, X_t, \mathcal{H}_{t-1})}}$. Here $\{\pi_t^{\text{sta}}\}_{t \geq 1}$ are pre-specified *stabilizing policies* that do not depend on data $\{Y_t, X_t, A_t\}_{t \geq 1}$. A default choice for the stabilizing policy when the action space is of size $|\mathcal{A}| < \infty$ is just $\pi_t^{\text{sta}}(a, x) = 1/|\mathcal{A}|$ for all $x, a$, and $t$; we discuss considerations for the choice of $\{\pi_t^{\text{sta}}\}_{t=1}^T$ in Section 3.3. We call these weights *square-root importance weights* because they are the square-root of the standard importance weights [Hammersley, 2013, Wang et al., 2017]. Our proposed estimator for $\theta^*(\mathcal{P})$, $\hat{\theta}_T$, is the maximizer of a *weighted* version of the M-estimation criterion of Equation (2):

$$\hat{\theta}_T := \underset{\theta \in \Theta}{\operatorname{argmax}} \, \frac{1}{T}\sum_{t=1}^{T} W_t m_\theta(Y_t, X_t, A_t) =: \underset{\theta \in \Theta}{\operatorname{argmax}} \, M_T(\theta).$$

Note that $M_T(\theta)$ defined above depends on both the data $\{X_t, A_t, Y_t\}_{t=1}^T$ and weights $\{W_t\}_{t=1}^T$. We provide asymptotically valid confidence regions for $\theta^*(\mathcal{P})$ by deriving the asymptotic distribution of $\hat{\theta}_T$ as $T \to \infty$ and by proving that the convergence in distribution is *uniform* over $\mathcal{P} \in \mathbf{P}$. Such convergence allows us to construct a uniformly asymptotically valid $1 - \alpha$ level confidence region, $C_T(\alpha)$, for $\theta^*(\mathcal{P})$, which is a confidence region that satisfies

$$\liminf_{T \to \infty} \inf_{\mathcal{P} \in \mathbf{P}} \mathbb{P}_{\mathcal{P}, \pi}\left(\theta^*(\mathcal{P}) \in C_T(\alpha)\right) \geq 1 - \alpha. \tag{3}$$

If $C_T(\alpha)$ were *not* uniformly valid, then there would exist an $\epsilon > 0$ such that for *every* sample size $T$, $C_T(\alpha)$'s coverage would be below $1 - \alpha - \epsilon$ for some worst-case $P_T \in \mathbf{P}$. Confidence regions which are asymptotically valid, but not *uniformly* asymptotically valid, fail to be reliable in practice [Leeb and Pötscher, 2005, Romano et al., 2012]. Note that on i.i.d. data it is generally straightforward to show that estimators that converge in distribution do so uniformly; however, as discussed in Zhang et al. [2020] and Appendix D, this is not the case on data collected with bandit algorithms.

To construct uniformly valid confidence regions for $\theta^*(\mathcal{P})$ we prove that $\hat{\theta}_T$ is uniformly asymptotically normal in the following sense:

$$\Sigma_T(\mathcal{P})^{-1/2} \ddot{M}_T(\hat{\theta}_T)\sqrt{T}(\hat{\theta}_T - \theta^*(\mathcal{P})) \xrightarrow{D} \mathcal{N}(0, I_d) \text{ uniformly over } \mathcal{P} \in \mathbf{P}, \tag{4}$$

where $\ddot{M}_T(\theta) := \frac{\partial^2}{\partial^2\theta}M_T(\theta)$ and $\Sigma_T(\mathcal{P}) := \frac{1}{T}\sum_{t=1}^{T}\mathbb{E}_{\mathcal{P},\pi_t^{\text{sta}}}\left[\dot{m}_{\theta^*(\mathcal{P})}(Y_t, X_t, A_t)^{\otimes 2}\right]$. We define $\dot{m}_\theta := \frac{\partial}{\partial\theta}m_\theta$. Similarly we define respectively $\ddot{m}_\theta$ and $\dddot{m}_\theta$ as the second and third partial derivatives of $m_\theta$ with respect to $\theta$. For any vector $z$ we define $z^{\otimes 2} := zz^\top$.

## 3.1 Intuition for Square-Root Importance Weights

The critical role of the square-root importance weights $W_t = \sqrt{\frac{\pi_t^{\text{sta}}(A_t, X_t)}{\pi_t(A_t, X_t, \mathcal{H}_{t-1})}}$ is to adjust for instability in the *variance* of M-estimators due to the bandit algorithm. These weights act akin to standard importance weights when squared and adjust a key term in the variance of M-estimators from depending on adaptive policies $\{\pi_t\}_{t=1}^{T}$, which can be ill-behaved, to depending on the pre-specified stabilizing policies $\{\pi_t^{\text{sta}}\}_{t=1}^{T}$. See Zhang et al. [2020] and Deshpande et al. [2018] for more discussion of the ill-behavior of the action selection probabilities for common bandit algorithms, which occurs particularly when there is no unique optimal policy.

As an illustrative example, consider the least-squares estimators in a finite-arm linear contextual bandit setting. Assume that $\mathbb{E}_{\mathcal{P}}[Y_t|X_t, A_t = a] = X_t^\top\theta_a^*(\mathcal{P})$ w.p. 1. We focus on estimating $\theta_a^*(\mathcal{P})$ for some $a \in \mathcal{A}$. The least-squares estimator corresponds to an M-estimator with $m_{\theta_a}(Y_t, X_t, A_t) = -\mathbb{1}_{A_t=a}(Y_t - X_t^\top\theta_a)^2$. The adaptively weighted least-squares (AW-LS) estimator is $\hat{\theta}_{T,a}^{\text{AW-LS}} := \text{argmax}_{\theta_a}\{-\sum_{t=1}^{T}W_t\mathbb{1}_{A_t=a}(Y_t - X_t^\top\theta_a)^2\}$. For simplicity, suppose that the stabilizing policy does not change with $t$ and drop the index $t$ to get $\pi^{\text{sta}}$. Taking the derivative of this criterion, we get $0 = \sum_{t=1}^{T}W_t\mathbb{1}_{A_t=a}X_t(Y_t - X_t^\top\hat{\theta}_{T,a}^{\text{AW-LS}})$, and rearranging terms gives

$$\frac{1}{\sqrt{T}}\sum_{t=1}^{T}W_t\mathbb{1}_{A_t=a}X_tX_t^\top\left(\hat{\theta}_{T,a}^{\text{AW-LS}} - \theta_a^*(\mathcal{P})\right) = \frac{1}{\sqrt{T}}\sum_{t=1}^{T}W_t\mathbb{1}_{A_t=a}X_t\left(Y_t - X_t^\top\theta_a^*(\mathcal{P})\right). \quad (5)$$

Note that the right hand side of Equation (5) is a martingale difference sequence with respect to history $\{\mathcal{H}_t\}_{t=0}^{T}$ because $\mathbb{E}_{\mathcal{P},\pi}[W_t\mathbb{1}_{A_t=a}(Y_t - X_t^\top\theta_a^*(\mathcal{P}))|\mathcal{H}_{t-1}] = 0$ for all $t$; by law of iterated expectations and since $W_t \in \sigma(\mathcal{H}_{t-1}, X_t, A_t)$, $\mathbb{E}_{\mathcal{P},\pi}[W_t\mathbb{1}_{A_t=a}(Y_t - X_t^\top\theta_a^*(\mathcal{P}))|\mathcal{H}_{t-1}]$ equals

$$\mathbb{E}_{\mathcal{P}}\left[W_t\pi_t(a, X_t, \mathcal{H}_{t-1})\mathbb{E}_{\mathcal{P}}\left[Y_t - X_t^\top\theta_a^*(\mathcal{P})|\mathcal{H}_{t-1}, X_t, A_t = a\right]\Big|\mathcal{H}_{t-1}\right]$$

$$\underset{(i)}{=} \mathbb{E}_{\mathcal{P}}\left[W_t\pi_t(a, X_t, \mathcal{H}_{t-1})\mathbb{E}_{\mathcal{P}}\left[Y_t - X_t^\top\theta_a^*(\mathcal{P})|X_t, A_t = a\right]\Big|\mathcal{H}_{t-1}\right]\underset{(ii)}{=} 0.$$

(i) holds by our i.i.d. potential outcomes assumption. (ii) holds since $\mathbb{E}_{\mathcal{P}}[Y_t|X_t, A_t = a] = X_t^\top\theta_a^*(\mathcal{P})$. We prove that (5) is uniformly asymptotically normal by applying a martingale central limit theorem (Appendix B.4). The key condition in this theorem is that the conditional variance converges uniformly, for which it is sufficient to show that the conditional covariance of $W_t\mathbb{1}_{A_t=a}\left(Y_t - X_t^\top\theta_a^*(\mathcal{P})\right)$ given $\mathcal{H}_{t-1}$ equals some positive-definite matrix $\Sigma(\mathcal{P})$ for every $t$, i.e.,

$$\mathbb{E}_{\mathcal{P},\pi}\left[W_t^2\mathbb{1}_{A_t=a}X_tX_t^\top\left(Y_t - X_t^\top\theta_a^*(\mathcal{P})\right)^2\Big|\mathcal{H}_{t-1}\right] = \Sigma(\mathcal{P}). \quad (6)$$

By law of iterated expectations, $\mathbb{E}_{\mathcal{P},\pi}[W_t^2\mathbb{1}_{A_t=a}X_tX_t^\top(Y_t - X_t^\top\theta_a^*(\mathcal{P}))^2|\mathcal{H}_{t-1}]$ equals

$$\mathbb{E}_{\mathcal{P}}\left[\mathbb{E}_{\mathcal{P},\pi}\left[\frac{\pi^{\text{sta}}(A_t, X_t)}{\pi_t(A_t, X_t, \mathcal{H}_{t-1})}\mathbb{1}_{A_t=a}X_tX_t^\top\left(Y_t - X_t^\top\theta_a^*(\mathcal{P})\right)^2\Big|\mathcal{H}_{t-1}, X_t\right]\Big|\mathcal{H}_{t-1}\right] \quad (7)$$

$$\underset{(a)}{=} \mathbb{E}_{\mathcal{P}}\left[\mathbb{E}_{\mathcal{P},\pi^{\text{sta}}}\left[\mathbb{1}_{A_t=a}X_tX_t^\top\left(Y_t - X_t^\top\theta_a^*(\mathcal{P})\right)^2\Big|\mathcal{H}_{t-1}, X_t\right]\Big|\mathcal{H}_{t-1}\right]$$

$$\underset{(b)}{=} \mathbb{E}_{\mathcal{P}}\left[\mathbb{E}_{\mathcal{P},\pi^{\text{sta}}}\left[\mathbb{1}_{A_t=a}X_tX_t^\top\left(Y_t - X_t^\top\theta_a^*(\mathcal{P})\right)^2\Big|X_t\right]\Big|\mathcal{H}_{t-1}\right]$$

$$\underset{(c)}{=} \mathbb{E}_{\mathcal{P}}\left[\mathbb{E}_{\mathcal{P},\pi^{\text{sta}}}\left[\mathbb{1}_{A_t=a}X_tX_t^\top\left(Y_t - X_t^\top\theta_a^*(\mathcal{P})\right)^2\Big|X_t\right]\right]$$

$$\underset{(d)}{=} \mathbb{E}_{\mathcal{P},\pi^{\text{sta}}}[\mathbb{1}_{A_t=a}X_tX_t^\top(Y_t - X_t^\top\theta_a^*(\mathcal{P}))^2] =: \Sigma(\mathcal{P}).$$

Above, (a) holds because the importance weights change the sampling measure from the adaptive policy $\pi_t$ to the pre-specified stabilizing policy $\pi^{\text{sta}}$. (b) holds by our i.i.d. potential outcomes

assumption and because $\pi^{\text{sta}}$ is a pre-specified policy. (c) holds because $X_t$ does not depend on $\mathcal{H}_{t-1}$ by our i.i.d. potential outcomes assumption. (d) holds by the law of iterated expectations. Note that $\Sigma(\mathcal{P})$ does not depend on $t$ because $\pi^{\text{sta}}$ is not time-varying. In contrast, without the adaptive weighting, i.e., when $W_t = 1$, the conditional covariance of $\mathbb{1}_{A_t=a} \left( Y_t - X_t^\top \theta_a^*(\mathcal{P}) \right)$ on $\mathcal{H}_{t-1}$ is a random variable, due to the adaptive policy $\pi_t$.

In Figure 1 we plot the empirical distributions of the z-statistic for the least-squares estimator both with and without adaptive weighting. We consider a two-armed bandit with $A_t \in \{0, 1\}$. Let $\theta_1^*(\mathcal{P}) := \mathbb{E}_\mathcal{P}[Y_t(1)]$ and $m_{\theta_1}(Y_t, A_t) := -A_t(Y_t - \theta_1)^2$. The unweighted version, i.e., the ordinary least-squares (OLS) estimator, is $\hat{\theta}_{T,1}^{\text{OLS}} := \operatorname{argmax}_{\theta_1} \frac{1}{T} \sum_{t=1}^{T} m_{\theta_1}(Y_t, A_t)$. The adaptively weighted version is $\hat{\theta}_{T,1}^{\text{AW-LS}} := \operatorname{argmax}_{\theta_1} \frac{1}{T} \sum_{t=1}^{T} W_t m_{\theta_1}(Y_t, A_t)$. We collect data using Thompson Sampling and use a uniform stabilizing policy where $\pi^{\text{sta}}(1) = \pi^{\text{sta}}(0) = 0.5$. It is clear that the least-squares estimator with adaptive weighting has a z-statistic that is much closer to a normal distribution.

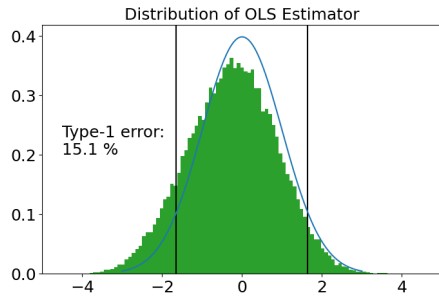
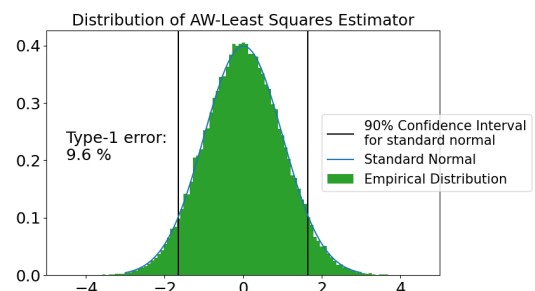

Figure 1: The empirical distributions of the weighted and unweighted least-squares estimators for $\theta_1^*(\mathcal{P}) := \mathbb{E}_\mathcal{P}[Y_t(1)]$ in a two arm bandit setting where $\mathbb{E}_\mathcal{P}[Y_t(1)] = \mathbb{E}_\mathcal{P}[Y_t(0)] = 0$. We perform Thompson Sampling with $\mathcal{N}(0, 1)$ priors, $\mathcal{N}(0, 1)$ errors, and $T = 1000$. Specifically, we plot $\sqrt{\sum_{t=1}^{T} A_t} (\hat{\theta}_{T,1}^{\text{OLS}} - \theta_1^*(\mathcal{P}))$ on the left and $\left( \frac{1}{\sqrt{T}} \sum_{t=1}^{T} \sqrt{\frac{0.5}{\pi_t(1)}} A_t \right) (\hat{\theta}_{T,1}^{\text{AW-LS}} - \theta_1^*(\mathcal{P}))$ on the right.

The square-root importance weights are a form of variance stabilizing weights, akin to those introduced in Hadad et al. [2019] for estimating means and differences in means on data collected with multi-armed bandits. In fact, in the special case that $|\mathcal{A}| < \infty$ and $\phi(X_t, A_t) = [\mathbb{1}_{A_t=1}, \mathbb{1}_{A_t=2}, ..., \mathbb{1}_{A_t=|\mathcal{A}|}]^\top$, the adaptively weighted least-squares estimator is equivalent to the weighted average estimator of Hadad et al. [2019]. See Section 4 for more on Hadad et al. [2019].

## 3.2 Asymptotic Normality and Confidence Regions

We now discuss conditions under which the adaptively weighted M-estimators are asymptotically normal in the sense of Equation (4). In general, our conditions differ from those made for standard M-estimators on i.i.d. data because (i) the data is adaptively collected, i.e., $\pi_t$ can depend on $\mathcal{H}_{t-1}$ and (ii) we ensure uniform convergence over $\mathcal{P} \in \mathbf{P}$, which is stronger than guaranteeing convergence pointwise for each $\mathcal{P} \in \mathbf{P}$.

**Condition 1** (Stochastic Bandit Environment). *Potential outcomes* $\{X_t, Y_t(a) : a \in \mathcal{A}\} \overset{i.i.d.}{\sim} \mathcal{P} \in \mathbf{P}$ *over* $t \in [1 : T]$.

Condition 1 implies that $Y_t$ is independent of $\mathcal{H}_{t-1}$ given $X_t$ and $A_t$, and the conditional distribution $Y_t \mid X_t, A_t$ is invariant over time. Also note that action space $\mathcal{A}$ can be finite or infinite.

**Condition 2** (Differentiable). *The first three derivatives of* $m_\theta(y, x, a)$ *with respect to* $\theta$ *exist for every* $\theta \in \Theta$, *every* $a \in \mathcal{A}$, *and every* $(x, y)$ *in the joint support of* $\{\mathcal{P} : \mathcal{P} \in \mathbf{P}\}$.

**Condition 3** (Bounded Parameter Space). *For all* $\mathcal{P} \in \mathbf{P}$, $\theta^*(\mathcal{P}) \in \Theta$, *a bounded open subset of* $\mathbb{R}^d$.

**Condition 4** (Lipschitz). *There exists some real-valued function* $g$ *such that (i)* $\sup_{\mathcal{P} \in \mathbf{P}, t \geq 1} \mathbb{E}_{\mathcal{P}, \pi_t^{\text{sta}}}[g(Y_t, X_t, A_t)^2]$ *is bounded and (ii) for all* $\theta, \theta' \in \Theta$,

$$|m_\theta(Y_t, X_t, A_t) - m_{\theta'}(Y_t, X_t, A_t)| \leq g(Y_t, X_t, A_t)\|\theta - \theta'\|_2.$$

Conditions 3 and 4 together restrict the complexity of the function $m$ in order to ensure a martingale law of large numbers result holds uniformly over functions $\{m_\theta : \theta \in \Theta\}$; this is used to prove the consistency of $\hat{\theta}_T$. Similar conditions are commonly used to prove consistency of M-estimators based on i.i.d. data, although the boundedness of the parameter space can be dropped when $m_\theta$ is a concave function of $\theta$ for all $Y_t, A_t, X_t$ (as it is in many canonical examples such as least squares) [Van der Vaart, 2000, Engle, 1994, Bura et al., 2018]; we expect that a similar result would hold for adaptively weighted M-estimators.

**Condition 5** (Moments). *The fourth moments of $m_{\theta^*(\mathcal{P})}(Y_t, X_t, A_t)$, $\dot{m}_{\theta^*(\mathcal{P})}(Y_t, X_t, A_t)$, and $\ddot{m}_{\theta^*(\mathcal{P})}(Y_t, X_t, A_t)$ with respect to $\mathcal{P}$ and policy $\pi_t^{\mathrm{sta}}$ are bounded uniformly over $\mathcal{P} \in \boldsymbol{P}$ and $t \geq 1$. For all sufficiently large $T$, the minimum eigenvalue of $\Sigma_{T,P} := \frac{1}{T}\sum_{t=1}^{T} \mathbb{E}_{\mathcal{P}, \pi_t^{\mathrm{sta}}}\left[\dot{m}_{\theta^*(\mathcal{P})}(Y_t, X_t, A_t)^{\otimes 2}\right]$ is bounded above $\delta_{\dot{m}^2} > 0$ for all $\mathcal{P} \in \boldsymbol{P}$.*

Condition 5 is similar to those of Van der Vaart [2000, Theorem 5.41]. However, to guarantee uniform convergence we assume that moment bounds hold uniformly over $\mathcal{P} \in \boldsymbol{P}$ and $t \geq 1$.

**Condition 6** (Third Derivative Domination). *For $B \in \mathbb{R}^{d \times d \times d}$, we define $\|B\|_1 := \sum_{i=1}^{d}\sum_{j=1}^{d}\sum_{k=1}^{d}|B_{i,j,k}|$. There exists a function $\dddot{m}(Y_t, X_t, A_t) \in \mathbb{R}^{d \times d \times d}$ such that (i) $\sup_{\mathcal{P} \in \boldsymbol{P}, t \geq 1}\mathbb{E}_{\mathcal{P}, \pi_t^{\mathrm{sta}}}\left[\|\dddot{m}(Y_t, X_t, A_t)\|_1^2\right]$ is bounded and (ii) for all $\mathcal{P} \in \boldsymbol{P}$ there exists some $\epsilon_{\dddot{m}} > 0$ such that the following holds with probability 1,*

$$\sup_{\theta \in \Theta: \|\theta - \theta^*(\mathcal{P})\| \leq \epsilon_{\dddot{m}}} \|\dddot{m}_\theta(Y_t, X_t, A_t)\|_1 \leq \|\dddot{m}(Y_t, X_t, A_t)\|_1.$$

Condition 6 is again similar to those in classical M-estimator asymptotic normality proofs [Van der Vaart, 2000, Theorem 5.41].

**Condition 7** (Maximizing Solution).
*(i) For all $\mathcal{P} \in \boldsymbol{P}$, there exists a $\theta^*(\mathcal{P}) \in \Theta$ such that (a) $\theta^*(\mathcal{P}) \in \operatorname{argmax}_{\theta \in \Theta}\mathbb{E}_{\mathcal{P}}\left[m_\theta(Y_t, X_t, A_t)\big|X_t, A_t\right]$ w.p. 1, (b) $\mathbb{E}_{\mathcal{P}}\left[\dot{m}_{\theta^*(\mathcal{P})}(Y_t, X_t, A_t)\big|X_t, A_t\right] = 0$ w.p. 1, and (c) $\mathbb{E}_{\mathcal{P}}\left[\ddot{m}_{\theta^*(\mathcal{P})}(Y_t, X_t, A_t)\big|X_t, A_t\right] \preceq 0$ w.p. 1.*
*(ii) There exists some positive definite matrix $H$ such that $-\frac{1}{T}\sum_{t=1}^{T}\mathbb{E}_{\mathcal{P}, \pi_t^{\mathrm{sta}}}\left[\ddot{m}_{\theta^*(\mathcal{P})}(Y_t, X_t, A_t)\right] \succeq H$ for all $\mathcal{P} \in \boldsymbol{P}$ and all sufficiently large $T$.*

For matrices $A, B$, we define $A \succeq B$ to mean that $A - B$ is positive semi-definite, as used above. Condition 7 (i) ensures that $\theta^*(\mathcal{P})$ is a conditionally maximizing solution for all contexts $X_t$ and actions $A_t$; this ensures that $\{\dot{m}_{\theta^*(\mathcal{P})}(Y_t, X_t, A_t)\}_{t=1}^{T}$ is a martingale difference sequence with respect to $\{\mathcal{H}_t\}_{t=1}^{T}$. Note it does not require $\theta^*(\mathcal{P})$ to always be a conditionally *unique* optimal solution. Condition 7 (ii) is related to the local curvature at the maximizing solution and the analogous condition in the i.i.d. setting is trivially satisfied; we specifically use this condition to ensure we can replace $\ddot{M}(\theta^*(\mathcal{P}))$ with $\ddot{M}(\hat{\theta}_T)$ in our asymptotic normality result, i.e., that $\ddot{M}(\theta^*(\mathcal{P}))^{-1}\ddot{M}(\hat{\theta}_T) \xrightarrow{P} I_d$ uniformly over $\mathcal{P} \in \boldsymbol{P}$.

**Condition 8** (Well-Separated Solution). *For all sufficiently large $T$, for any $\epsilon > 0$, there exists some $\delta > 0$ such that for all $\mathcal{P} \in \boldsymbol{P}$,*

$$\inf_{\theta \in \Theta: \|\theta - \theta^*(\mathcal{P})\|_2 > \epsilon} \left\{ \frac{1}{T}\sum_{t=1}^{T}\mathbb{E}_{\mathcal{P}, \pi_t^{\mathrm{sta}}}\left[m_{\theta^*(\mathcal{P})}(Y_t, X_t, A_t) - m_\theta(Y_t, X_t, A_t)\right] \right\} \geq \delta.$$

A well-separated solution condition akin to Condition 8 is commonly assumed in order to prove consistency of M-estimators, e.g., see Van der Vaart [2000, Theorem 5.7]. Note that the difference between Condition 7 (i) and Condition 8 is that the former is a conditional statement (conditional on $X_t, A_t$) and the latter is a marginal statement (marginal over $X_t, A_t$, where $A_t$ is chosen according to stabilizing policies $\pi_t^{\mathrm{sta}}$). Condition 7 (i) means there is a $\theta^*(\mathcal{P})$ solution for all contexts $X_t$ and actions $A_t$ that does not need to be unique, however Condition 8 assumes that marginally over $X_t, A_t$ there is a well-separated solution.

**Condition 9** (Bounded Importance Ratios). *$\{\pi_t^{\mathrm{sta}}\}_{t=1}^{T}$ do not depend on data $\{Y_t, X_t, A_t\}_{t=1}^{T}$. For all $t \geq 1$, $\rho_{\min} \leq \frac{\pi_t^{\mathrm{sta}}(A_t, X_t)}{\pi_t(A_t, X_t, \mathcal{H}_{t-1})} \leq \rho_{\max}$ w.p. 1 for some constants $0 < \rho_{\min} \leq \rho_{\max} < \infty$.*

Note that Condition 9 implies that for a stabilizing policy that is not time-varying, the action selection probabilities of the bandit algorithm $\pi_t(A_t, X_t, \mathcal{H}_{t-1})$ must be bounded away from zero w.p. 1. Similar boundedness assumptions are also made in the off-policy evaluation literature [Thomas and Brunskill, 2016, Kallus and Uehara, 2020]. We discuss this condition further in Sections 3.3 and 6.

**Theorem 1** (Uniform Asymptotic Normality of Adaptively Weighted M-Estimators). *Under Conditions 1-9 we have that $\hat{\theta}_T \xrightarrow{P} \theta^*(\mathcal{P})$ uniformly over $\mathcal{P} \in \boldsymbol{P}$. Additionally,*

$$\Sigma_T(\mathcal{P})^{-1/2} \ddot{M}_T(\hat{\theta}_T) \sqrt{T}(\hat{\theta}_T - \theta^*(\mathcal{P})) \xrightarrow{D} \mathcal{N}(0, I_d) \text{ uniformly over } \mathcal{P} \in \boldsymbol{P}. \quad (8)$$

The asymptotic normality result of equation (8) guarantees that for $d$-dimensional $\theta^*(\mathcal{P})$,

$$\liminf_{T \to \infty} \inf_{\mathcal{P} \in \boldsymbol{P}} \mathbb{P}_{\mathcal{P}, \pi} \left( \left[ \Sigma_T(\mathcal{P})^{-1/2} \ddot{M}_T(\hat{\theta}_T) \sqrt{T}(\hat{\theta}_T - \theta^*(\mathcal{P})) \right]^{\otimes 2} \le \chi^2_{d, (1-\alpha)} \right) = 1 - \alpha.$$

Above $\chi^2_{d, (1-\alpha)}$ is the $1 - \alpha$ quantile of the $\chi^2$ distribution with $d$ degrees of freedom. Note that the region $C_T(\alpha) := \left\{ \theta \in \Theta : [\Sigma_T(\mathcal{P})^{-1/2} \ddot{M}_T(\hat{\theta}_T) \sqrt{T}(\hat{\theta}_T - \theta^*(\mathcal{P}))]^{\otimes 2} \le \chi^2_{d, (1-\alpha)} \right\}$ defines a $d$-dimensional hyper-ellipsoid confidence region for $\theta^*(\mathcal{P})$. Also note that since $\ddot{M}_T(\hat{\theta}_T)$ does not concentrate under standard bandit algorithms, we cannot use standard arguments to justify treating $\hat{\theta}_T$ as multivariate normal with covariance $\ddot{M}_T(\hat{\theta}_T)^{-1} \Sigma_T(\mathcal{P}) \ddot{M}_T(\hat{\theta}_T)^{-1}$. Nevertheless, Theorem 1 can be used to guarantee valid confidence regions for subset of entries in $\theta^*(\mathcal{P})$ by using projected confidence regions [Nickerson, 1994]. Projected confidence regions take a confidence region for all parameters $\theta^*(\mathcal{P})$ and project it onto the lower dimensional space on which the subset of target parameters lie (Appendix A.2).

### 3.3 Choice of Stabilizing Policy

When the action space is bounded, using weights $W_t = 1/\sqrt{\pi_t(A_t, X_t, \mathcal{H}_{t-1})}$ is equivalent to using square-root importance weights with a stabilizing policy that selects actions uniformly over $\mathcal{A}$; this is because weighted M-estimators are invariant to all weights being scaled by the same constant. It can make sense to choose a non-uniform stabilizing policy in order to prevent the square-root importance weights from growing too large and to ensure Condition 9 holds; disproportionately up-weighting a few observations can lead to unstable estimators. Note that an analogue of our stabilizing policy exists in the causal inference literature, namely, "stabilized weights" use a probability density in the numerator of the weights to prevent them from becoming too large [Robins et al., 2000].

We now discuss how to choose stabilizing policies $\{\pi_t^{\text{sta}}\}_{t \ge 1}$ in order to minimize the asymptotic variance of adaptively weighted M-estimators. We focus on the adaptively weighted least-squares estimator when we have a linear outcome model $\mathbb{E}_{\mathcal{P}}[Y_t | X_t, A_t] = X_t^\top \theta_{A_t}$:

$$\hat{\theta}^{\text{AW-LS}} := \underset{\theta \in \Theta}{\operatorname{argmax}} \left\{ \frac{1}{T} \sum_{t=1}^{T} W_t \left( Y_t - X_t^\top \theta_{A_t} \right)^2 \right\}. \quad (9)$$

Recall that our use of adaptive weights is to adjust for instability in the variance of M-estimators induced by the bandit algorithm in order to construct valid confidence regions; note that weighted estimators are not typically used for this reason. On i.i.d. data, the least-squares criterion is weighted like in Equation (9) in order to minimize the variance of estimators under noise heteroskedasticity; in this setting, the best linear unbiased estimator has weights $W_t = 1/\sigma^2(A_t, X_t)$ where $\sigma^2(A_t, X_t) := \mathbb{E}_{\mathcal{P}}[(Y_t - X_t^\top \theta_{A_t}^*(\mathcal{P}))^2 | X_t, A_t]$; this up-weights the importance of observations with low noise variance. Intuitively, if we do not need to variance stabilize, $\{W_t\}_{t \ge 1}$ should be determined by the relative importance of minimizing the errors for different observations, i.e., their noise variance.

In light of this observation, we expect that under homoskedastic noise there is no reason to up-weight some observations over others. This would recommend choosing the stabilizing policy to make $W_t = \sqrt{\pi_t^{\text{sta}}(A_t, X_t)/\pi_t(A_t, X_t, \mathcal{H}_{t-1})}$ as close to 1 as possible, subject to the constraint that the stabilizing policies are pre-specified, i.e., $\{\pi_t^{\text{sta}}\}_{t \ge 1}$ do not depend on data $\{Y_t, X_t, A_t\}_{t \ge 1}$ (see Appendix C for details). Since adjusting for heteroskedasticity and variance stabilization are distinct uses of weights, under heteroskedasticity, we recommend that the weights are combined in the following sense: $W_t = (1/\sigma^2(A_t, X_t)) \sqrt{\pi_t^{\text{sta}}(A_t, X_t)/\pi_t(A_t, X_t, \mathcal{H}_{t-1})}$. This would mean that to minimize variance, we still want to choose the stabilizing policies to make $\pi_t^{\text{sta}}(A_t, X_t)/\pi_t(A_t, X_t, \mathcal{H}_{t-1})$ as close to 1 possible, subject to the pre-specified constraint.

# 4 Related Work

Villar et al. [2015] and Rafferty et al. [2019] empirically illustrate that classical ordinary least squares (OLS) inference methods have inflated Type-1 error when used on data collected with a variety of regret-minimizing multi-armed bandit algorithms. Chen et al. [2020] prove that the OLS estimator is asymptotically normal on data collected with an $\epsilon$-greedy algorithm, but their results do not cover settings in which there is no unique optimal policy, e.g., a multi-arm bandit with two identical arms (Appendix E). Recent work has discussed the non-normality of OLS on data collected with bandit algorithms when there is no unique optimal policy and proposed alternative methods for statistical inference. A common thread between these methods is that they all utilize a form of *adaptive weighting*. Deshpande et al. [2018] introduced the W-decorrelated estimator, which adjusts the OLS estimator with a sum of adaptively weighted residuals. In the multi-armed bandit setting, the W-decorrelated estimator up-weights observations from early in the study and down-weights observations from later in the study [Zhang et al., 2020]. In the batched bandit setting, Zhang et al. [2020] show that the Z-statistics for the OLS estimators computed separately on each batch are jointly asymptotically normal. Standardizing the OLS statistic for each batch effectively adaptively re-weights the observations in each batch.

Hadad et al. [2019] introduce adaptively weighted versions of both the standard augmented-inverse propensity weighted estimator (AW-AIPW) and the sample mean (AWA) for estimating parameters of simple models on data collected with bandit algorithms. They introduce a class of adaptive "variance stabilizing" weights, for which the variance of a normalized version of their estimators converges in probability to a constant. In their discussion section they note open questions, two of which this work addresses: 1) "What additional estimators can be used for normal inference with adaptively collected data?" and 2) How do their results generalize to more complex sampling designs, like data collected with contextual bandit algorithms? We demonstrate that variance stabilizing adaptive weights can be used to modify a large class of M-estimators to guarantee valid inference. This generalization allows us to perform valid inference for a large class of important inferential targets: parameters of models for expected outcomes that are context dependent.

Recently, adaptive weighting has also been used in off-policy evaluation methods for when the behavior policy (policy used to collect the data) is a contextual bandit algorithm [Bibaut et al., 2021, Zhan et al., 2021]. In this literature the estimand is the value, or average expected reward, of a pre-specified policy (note this is a scalar value). In contrast, in our work we are interested in constructing confidence regions for parameters of a model for an outcome (that could be the reward)—for example, this could be parameters of a logistic regression model for a binary outcome. We believe in the future there could be theory that could unify these adaptive weighting methods for these different estimands.

An alternative to using asymptotic approximations to construct confidence intervals is to use high-probability confidence bounds. These bounds provide stronger guarantees than those based on asymptotic approximations, as they are guaranteed to hold for finite samples. The downside is that these bounds are typically much wider, which is why much of classical statistics uses asymptotic approximations. Here we do the same. In Section 5, we empirically compare our to the self-normalized martingale bound [Abbasi-Yadkori et al., 2011], a high-probability bound commonly used in the bandit literature.

# 5 Simulation Results

In this section, $R_t = Y_t$. We consider two settings: a continuous reward setting and a binary reward setting. In the continuous reward setting, the rewards are generated with mean $\mathbb{E}_{\mathcal{P}}[R_t|X_t, A_t] = \tilde{X}_t^\top \theta_0^*(\mathcal{P}) + A_t \tilde{X}_t^\top \theta_1^*(\mathcal{P})$ and noise drawn from a student's $t$ distribution with five degrees of freedom; here $\tilde{X}_t = [1, X_t] \in \mathbb{R}^3$ ($X_t$ with intercept term), actions $A_t \in \{0, 1\}$, and parameters $\theta_0^*(\mathcal{P}), \theta_1^*(\mathcal{P}) \in \mathbb{R}^3$. In the binary reward setting, the reward $R_t$ is generated as a Bernoulli with success probability $\mathbb{E}_{\mathcal{P}}[R_t|X_t, A_t] = [1 + \exp(-\tilde{X}_t^\top \theta_0^*(\mathcal{P}) - A_t \tilde{X}_t^\top \theta_1^*(\mathcal{P}))]^{-1}$. Furthermore, in both simulation settings we set $\theta_0^*(\mathcal{P}) = [0.1, 0.1, 0.1]$ and $\theta_1^*(\mathcal{P}) = [0, 0, 0]$, so there is no unique optimal arm; we call vector parameter $\theta_1^*(\mathcal{P})$ the *advantage* of selecting $A_t = 1$ over $A_t = 0$. Also in both settings, the contexts $X_t$ are drawn i.i.d. from a uniform distribution.

In both simulation settings we collect data using Thompson Sampling with a linear model for the expected reward and normal priors [Agrawal and Goyal, 2013] (so even when the reward is

binary). We constrain the action selection probabilities with *clipping* at a rate of $0.05$; this means that while typical Thompson Sampling produces action selection probabilities $\pi_t^{\text{TS}}(A_t, X_t, \mathcal{H}_{t-1})$, we instead use action selection probabilities $\pi_t(A_t, X_t, \mathcal{H}_{t-1}) = 0.05 \vee \left( 0.95 \wedge \pi_t^{\text{TS}}(A_t, X_t, \mathcal{H}_{t-1}) \right)$ to select actions. We constrain the action selection probabilities in order to ensure weights $W_t$ are bounded when using a uniform stabilizing policy; see Sections 3.2 and 6 for more discussion on this boundedness assumption. Also note that increasing the amount the algorithm explores (clipping) decreases the expected width of confidence intervals constructed on the resulting data (see Section 6).

To analyze the data, in the continuous reward setting, we use least-squares estimators with a correctly specified model for the expected reward, i.e., M-estimators with $m_\theta(R_t, X_t, A_t) = -(R_t - \tilde{X}_t^\top \theta_0 - A_t \tilde{X}_t^\top \theta_1)^2$. We consider both the unweighted and adaptively weighted versions. We also compare to the self-normalized martingale bound [Abbasi-Yadkori et al., 2011] and the W-decorrelated estimator [Deshpande et al., 2018], as they were both developed for the linear expected reward setting. For the self-normalized martingale bound, which requires explicit bounds on the parameter space, we set $\Theta = \{\theta \in \mathbb{R}^6 : \|\theta\|_2 \le 6\}$. In the binary reward setting, we also assume a correctly specified model for the expected reward. We use both unweighted and adaptively weighted maximum likelihood estimators (MLEs), which correspond to an M-estimators with $m_\theta(R_t, X_t, A_t)$ set to the negative log-likelihood of $R_t$ given $X_t, A_t$. We solve for these estimators using Newton–Raphson optimization and do not put explicit bounds on the parameter space $\Theta$ (note in this case $m_\theta$ is concave in $\theta$ [Agresti, 2015, Chapter 5.4.2]). See Appendix A for additional details and simulation results.

In Figure 4 we plot the empirical coverage probabilities and volumes of 90% confidence regions for $\theta^*(\mathcal{P}) := [\theta_0^*(\mathcal{P}), \theta_1^*(\mathcal{P})]$ and $\theta_1^*(\mathcal{P})$ in both the continuous and binary reward settings. While the confidence regions based on the unweighted least-squares estimator (OLS) and the unweighted MLE have significant undercoverage that does not improve as $T$ increases, the confidence regions based on the adaptively weighted versions, AW-LS and AW-MLE, have very reliable coverage. For the confidence regions for $\theta_1^*(\mathcal{P})$ based on the AW-LS and AW-MLE, we include both projected confidence regions (for which we have theoretical guarantees) and non-projected confidence regions. The confidence regions based on projections are conservative but nevertheless have comparable volume to those based on OLS and MLE respectively. We do not prove theoretical guarantees for the non-projection confidence regions for AW-LS and AW-MLE, however they perform well across in our simulations. Both types of confidence regions based on AW-LS have significantly smaller volumes than those constructed using the self-normalized martingale bound and W-decorrelated estimator. Note that the W-decorrelated estimator and self-normalized martingale bounds are designed for linear contextual bandits and are thus not applicable for the logistic regression model setting. The confidence regions constructed using the self-normalized martingale bound have reliable coverage as well, but are very conservative. Empirically, we found that the coverage probabilities of the confidence regions based on the W-decorrelated estimator were very sensitive to the choice of tuning parameters. We use $5,000$ Monte-Carlo repetitions and the error bars plotted are standard errors.

## 6 Discussion

**Immediate questions** We assume that ratios $\pi_t^{\text{sta}}(A_t, X_t)/\pi_t(A_t, X_t, \mathcal{H}_{t-1})$ are bounded for our theoretical results; this precludes $\pi_t(A_t, X_t, \mathcal{H}_{t-1})$ from going to zero for a fixed stabilizing policy. For simple models, e.g., the AW-LS estimator, we can let these ratios grow at a certain rate and still guarantee asymptotic normality (Appendix B.5); we conjecture similar results hold more generally.

**Generality and robustness** This work assumes that we have a well-specified model for the outcome $Y_t$, i.e., that $\theta^*(\mathcal{P}) \in \operatorname{argmax}_{\theta \in \Theta} \mathbb{E}_\mathcal{P}[m_\theta(Y_t, X_t, A_t) | X_t, A_t]$ w.p. 1. Our theorems use this assumption to ensure that $\{W_t \dot{m}_\theta(Y_t, X_t, A_t)\}_{t \ge 1}$ is a martingale difference sequence with respect to $\{\mathcal{H}_t\}_{t \ge 0}$. On i.i.d. data it is common to define $\theta^*(\mathcal{P})$ to be the best *projected* solution, i.e., $\theta_0(\mathcal{P}) \in \operatorname{argmax}_{\theta \in \Theta} \mathbb{E}_{\mathcal{P},\pi}[m_\theta(Y_t, X_t, A_t)]$. Note that the best projected solution, $\theta^*(\mathcal{P})$, depends on the distribution of the action selection policy $\pi$. It would be ideal to also be able to perform inference for a projected solution on adaptively collected data.

Another natural question is whether adaptive weighting methods work in Markov Decision Processes (MDP) environments. Taking the AW-LS estimator introduced in Section 3.1 as an example, our conditional variance derivation in Equation (7) fails to hold in an MDP setting, specifically equality (c). However, the conditional variance condition can be satisfied if we instead use weights $W_t = \{[\pi_t^{\text{sta}}(A_t, X_t) p^{\text{sta}}(X_t)]/[\pi_t(A_t, X_t, \mathcal{H}_{t-1}) \mathbb{P}_\mathcal{P}(X_t | X_{t-1}, A_{t-1})]\}^{1/2}$ where $\mathbb{P}_\mathcal{P}$ are the state

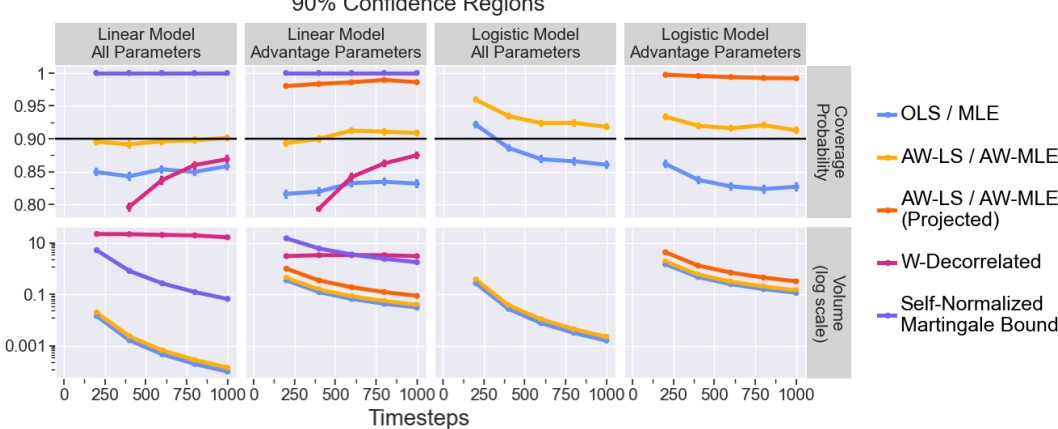

Figure 2: Empirical coverage probabilities (upper row) and volume (lower row) of 90% confidence ellipsoids. The left two columns are for the linear reward model setting (t-distributed rewards) and the right two columns are for the logistic regression model setting (Bernoulli rewards). We consider confidence ellipsoids for all parameters $\theta^*(\mathcal{P})$ and for advantage parameters $\theta_1^*(\mathcal{P})$ for both settings.

transition probabilities and $p^{\text{sta}}$ is a pre-specified distribution over states. In general though we do not expect to know the transition probabilities $\mathbb{P}_{\mathcal{P}}$ and if we tried to estimate them, our theory would require the estimator to have error $o_p(1/\sqrt{T})$, *below* the parametric rate.

**Trading-off regret minimization and statistical inference objectives** In sequential decision-making problems there is a fundamental trade-off between minimizing regret and minimizing estimation error for parameters of the environment using the resulting data [Bubeck et al., 2009, Dean et al., 2018]. Given this trade-off there are many open problems regarding how to minimize regret while still guaranteeing a certain amount of power or expected confidence interval width, e.g., developing sample size calculators for use in justifying the number of users in a mobile health trial, and developing new adaptive algorithms [Liu et al., 2014, Erraqabi et al., 2017, Yao et al., 2020].

## Acknowledgements and Disclosure of Funding

We thank Yash Nair for feedback on early drafts of this work.

Research reported in this paper was supported by National Institute on Alcohol Abuse and Alcoholism (NIAAA) of the National Institutes of Health under award number R01AA23187, National Institute on Drug Abuse (NIDA) of the National Institutes of Health under award number P50DA039838, National Cancer Institute (NCI) of the National Institutes of Health under award number U01CA229437, and by NIH/NIBIB and OD award number P41EB028242. The content is solely the responsibility of the authors and does not necessarily represent the official views of the National Institutes of Health.

This material is based upon work supported by the National Science Foundation Graduate Research Fellowship Program under Grant No. DGE1745303. Any opinions, findings, and conclusions or recommendations expressed in this material are those of the author(s) and do not necessarily reflect the views of the National Science Foundation.

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
