# Appendix

## Table of Contents

## A   Simulations

### A.1   Simulation Details

**Simulation Environment**

- Each dimension of $X_t$ is sampled independently from Uniform$(0, 5)$.
- $\theta^*(\mathcal{P}) = [\theta_0^*(\mathcal{P}), \theta_1^*(\mathcal{P})] = [0.1, 0.1, 0.1, 0, 0, 0]$, where $\theta_0^*(\mathcal{P}), \theta_1^*(\mathcal{P}) \in \mathbb{R}^3$.
  Below also include simulations where $[\theta_0^*(\mathcal{P}), \theta_1^*(\mathcal{P})] = [0.1, 0.1, 0.1, 0.2, 0.1, 0]$.
- t-Distributed rewards:  $R_t | X_t, A_t \sim t_5 + \tilde{X}_t^\top \theta_0^*(\mathcal{P}) + A_t \tilde{X}_t^\top \theta_1^*(\mathcal{P})$, where $t_5$ is a t-distribution with 5 degrees of freedom.
- Bernoulli rewards: $R_t | X_t, A_t \sim \text{Bernoulli}(expit(\nu_t))$ for $\nu_t = \tilde{X}_t^\top \theta_0^*(\mathcal{P}) + A_t \tilde{X}_t^\top \theta_1^*(\mathcal{P})$ and $expit(x) = \frac{1}{1+\exp(-x)}$.
- Poisson rewards: $R_t | X_t, A_t \sim \text{Poisson}(\exp(\nu_t))$ for $\nu_t = \tilde{X}_t^\top \theta_0^*(\mathcal{P}) + A_t \tilde{X}_t^\top \theta_1^*(\mathcal{P})$.

**Algorithm**

- Thompson Sampling with $\mathcal{N}(0, I_d)$ priors on each arm.
- 0.05 clipping
- Pre-processing rewards before received by algorithm:
  - Bernoulli: $2R_t - 1$
  - Poisson: $0.6R_t$

**Compute Time and Resources**    All simulations run within a few hours on a MacBook Pro.

## A.2 Details on Constructing of Confidence Regions

For notational convenience, we define $Z_t = [\tilde{X}_t, A_t \tilde{X}_t]$.

### A.2.1 Least Squares Estimators

- $\hat{\theta}_T = \left( \sum_{t=1}^T W_t Z_t Z_t^\top \right)^{-1} \sum_{t=1}^T W_t Z_t R_t$

  - For unweighted least squares, $W_t = 1$ and we call the estimator $\hat{\theta}_T^{\text{OLS}}$.
  - For adaptively weighted least squares, $W_t = \frac{1}{\sqrt{\pi_t(A_t, X_t, \mathcal{H}_{t-1})}}$; this is equivalent to using square-root importance weights with a uniform stabilizing policy. We call the estimator $\hat{\theta}_T^{\text{AW-LS}}$.

- We assume homoskedastic errors and estimate the noise variance $\sigma^2$ as follows:

$$\hat{\sigma}_T^2 = \frac{1}{T} \sum_{t=1}^T (R_t - Z_t^\top \hat{\theta}_T)^2.$$

- We use a Hotelling t-squared test statistic to construct confidence regions for $\theta^*(\mathcal{P})$:

$$C_T(\alpha) = \left\{ \theta \in \mathbb{R}^d : \left[ \hat{\Sigma}_T^{-1/2} \left( \frac{1}{T} \sum_{t=1}^T W_t Z_t Z_t^\top \right) \sqrt{T}(\hat{\theta}_T - \theta) \right]^{\otimes 2} \right.$$
$$\left. \leq \frac{d(T-1)}{T-d} F_{d,T-d}(1-\alpha) \right\}. \quad (10)$$

  - For the unweighted least-squares estimator we use the following variance estimator: $\hat{\Sigma}_T = \hat{\sigma}_T^2 \frac{1}{T} \sum_{t=1}^T Z_t Z_t^\top$.
  - For the AW-Least Squares estimator we use the following variance estimator: $\hat{\Sigma}_T = \hat{\sigma}_T^2 \frac{1}{T} \sum_{t=1}^T \frac{1}{\pi_t(A_t, X_t, \mathcal{H}_{t-1})}^{A_t} \frac{1}{1-\pi_t(A_t, X_t, \mathcal{H}_{t-1})}^{1-A_t} Z_t Z_t^\top$.

- To construct (non-projected) confidence regions for $\theta_1^*(\mathcal{P}) \in \mathbb{R}^{d_1}$ we treat the unweighted least squares / AW-LS estimators, $\hat{\theta}_{T,1}$, as $\mathcal{N}\left( \theta_1^*(\mathcal{P}), \frac{1}{T} \left( \frac{1}{T} \sum_{t=1}^T W_t Z_t Z_t^\top \right)^{-1} \hat{\Sigma}_T \left( \frac{1}{T} \sum_{t=1}^T W_t Z_t Z_t^\top \right)^{-1} \right)$. We use a Hotelling t-squared test statistic to construct confidence regions for $\theta_1^*(\mathcal{P})$:

$$C_T(\alpha) = \left\{ \theta_1 \in \mathbb{R}^{d_1} : \left[ V_{1,T}^{-1/2} \sqrt{T}(\hat{\theta}_{T,1} - \theta_1) \right]^{\otimes 2} \leq \frac{d_1(T-1)}{T-d_1} F_{d_1,T-d_1}(1-\alpha) \right\},$$

  where $V_{1,T}$ is the lower right $d_1 \times d_1$ block of matrix $\left( \frac{1}{T} \sum_{t=1}^T W_t Z_t Z_t^\top \right)^{-1} \hat{\Sigma}_T \left( \frac{1}{T} \sum_{t=1}^T W_t Z_t Z_t^\top \right)^{-1}$. Recall that for the unweighted least squares estimator $W_t = 1$ and for AW-LS $W_t = \frac{1}{\sqrt{\pi_t(A_t, X_t, \mathcal{H}_{t-1})}}$.

- For the AW-least squares estimator, we also construct projected confidence regions for $\theta_1^*(\mathcal{P})$ using the confidence region defined in equation (10). See Section A.2.5 below for more details on constructing projected confidence regions.

### A.2.2 MLE Estimators

| Distribution | $\nu$ | $b(\nu)$ | $b'(\nu)$ | $b''(\nu)$ | $b'''(\nu)$ |
|---|---|---|---|---|---|
| $\mathcal{N}(\mu, 1)$ | $\mu$ | $\frac{1}{2}\nu^2$ | $\nu = \mu$ | $1$ | $0$ |
| $\text{Poisson}(\lambda)$ | $\log \lambda$ | $\exp(\nu)$ | $\exp(\nu) = \lambda$ | $\exp(\nu) = \lambda$ | $\exp(\nu) = \lambda$ |
| $\text{Bernoulli}(p)$ | $\log\left(\frac{p}{1-p}\right)$ | $\log(1 + e^\nu)$ | $\frac{e^\nu}{1+e^\nu} = p$ | $\frac{e^\nu}{(1+e^\nu)^2} = p(1-p)$ | $p(1-p)(1-2p)$ |

- $\hat{\theta}_T$ is the root of the score function:
$$0 = \sum_{t=1}^{T} W_t \left( R_t - b'(\hat{\theta}_T^\top Z_t) \right) Z_t.$$

  We use Newton Raphson optimization to solve for $\hat{\theta}_T$.
    - For unweighted MLE, $W_t = 1$.
    - For AW-MLE, $W_t = \frac{1}{\sqrt{\pi_t(A_t, X_t, \mathcal{H}_{t-1})}}$; this is equivalent to using square-root importance weights with a uniform stabilizing policy.

- Second derivative of score function: $-\sum_{t=1}^{T} b''(\hat{\theta}_T^\top Z_t) Z_t Z_t^\top$.
- We use a Hotelling t-squared test statistic to construct confidence regions for $\theta^*(\mathcal{P})$:

$$C_T(\alpha) = \left\{ \theta \in \mathbb{R}^d : \left[ \hat{\Sigma}_T^{-1/2} \left( \frac{1}{T} \sum_{t=1}^{T} W_t b''(\hat{\theta}_T^\top Z_t) Z_t Z_t^\top \right) \sqrt{T}(\hat{\theta}_T - \theta) \right]^{\otimes 2} \right.$$
$$\left. \leq \frac{d(T-1)}{T-d} F_{d, T-d}(1-\alpha) \right\}. \quad (11)$$

    - For the MLE variance estimator, we use $\hat{\Sigma}_T = \frac{1}{T} \sum_{t=1}^{T} b''(\hat{\theta}_T^\top Z_t) Z_t Z_t^\top$.
    - For the AW-MLE variance estimator, we use $\hat{\Sigma}_T = \frac{1}{T} \sum_{t=1}^{T} \frac{1}{\pi_t(A_t, X_t, \mathcal{H}_{t-1})}^{A_t} \frac{1}{1-\pi_t(A_t, X_t, \mathcal{H}_{t-1})}^{1-A_t} b''(\hat{\theta}_T^\top Z_t) Z_t Z_t^\top$.

- To construct (non-projected) confidence regions for $\theta_1^*(\mathcal{P}) \in \mathbb{R}^{d_1}$ we treat the MLE / AW-MLE estimators, $\hat{\theta}_{T,1}$, as $\mathcal{N}\left( \theta_1^*(\mathcal{P}), \frac{1}{T} \left( \frac{1}{T} \sum_{t=1}^{T} W_t b''(\hat{\theta}_T^\top Z_t) Z_t Z_t^\top \right) \hat{\Sigma}_T^{-1} \left( \frac{1}{T} \sum_{t=1}^{T} W_t b''(\hat{\theta}_T^\top Z_t) Z_t Z_t^\top \right) \right)$.
  We use a Hotelling t-squared test statistic to construct confidence regions for $\theta_1^*(\mathcal{P})$:

$$C_T(\alpha) = \left\{ \theta_1 \in \mathbb{R}^{d_1} : \left[ V_{1,T}^{-1/2} \sqrt{T}(\hat{\theta}_{T,1} - \theta_1) \right]^{\otimes 2} \leq \frac{d_1(T-1)}{T-d_1} F_{d_1, T-d_1}(1-\alpha) \right\},$$

  where $V_{1,T}$ is the lower right $d_1 \times d_1$ block of matrix $\left( \frac{1}{T} \sum_{t=1}^{T} W_t b''(\hat{\theta}_T^\top Z_t) Z_t Z_t^\top \right) \hat{\Sigma}_T^{-1} \left( \frac{1}{T} \sum_{t=1}^{T} W_t b''(\hat{\theta}_T^\top Z_t) Z_t Z_t^\top \right)$.

- For the AW-MLE estimator, we also construct projected confidence regions for $\theta_1^*(\mathcal{P})$ using the confidence region defined in equation (11). See Section A.2.5 below for more details on constructing projected confidence regions.

### A.2.3 W-Decorrelated

The following is based on Algorithm 1 of Deshpande et al. [2018].

- The W-decorrelated estimator for $\theta^*(\mathcal{P})$ is constructed as follows with adaptive weights for $W_t \in \mathbb{R}^d$:
$$\hat{\theta}_T^{\text{WD}} = \hat{\theta}_T^{\text{OLS}} + \sum_{t=1}^{T} W_t(R_t - \tilde{X}_t^\top \hat{\theta}_T^{\text{OLS}}).$$

- The weights are set as follows:
  $W_1 = 0 \in \mathbb{R}^d$ and $W_t = (I_d - \sum_{s=1}^{t} \sum_{u=1}^{t} W_s Z_u^\top) Z_t \frac{1}{\lambda_T + \|Z_t\|_2^2}$ for $t > 1$.

- We choose $\lambda_T = \text{mineig}_{0.01}(Z_t Z_t^\top) / \log T$ and $\text{mineig}_\alpha(Z_t Z_t^\top)$ represents the $\alpha$ quantile of the minimum eigenvalue of $Z_t Z_t^\top$. This is similar to the procedure used in the simulations of Deshpande et al. [2018] and is guided by Proposition 5 in their paper.

- We assume homoskedastic errors and estimate the noise variance $\sigma^2$ as follows:
$$\hat{\sigma}_T^2 = \frac{1}{T} \sum_{t=1}^{T} (R_t - Z_t^\top \hat{\theta}_T^{\text{OLS}})^2.$$

- To construct confidence ellipsoids for $\theta^*(\mathcal{P})$ are constructed using a Hotelling t-squared statistic:

$$C_T(\alpha) = \left\{ \theta \in \mathbb{R}^d : (\hat{\theta}_T^{\text{WD}} - \theta)^\top V_T^{-1}(\hat{\theta}_T^{\text{WD}} - \theta) \leq \frac{d(T-1)}{T-d} F_{d,T-d}(1-\alpha) \right\}$$

where $V_T = \hat{\sigma}_T^2 \sum_{t=1}^T W_t W_t^\top$.

- To construct confidence ellipsoids for $\theta_1^*(\mathcal{P}) \in \mathbb{R}^{d_1}$ with the following confidence ellipsoid where $V_{T,1}$ is the lower right $d_1 \times d_1$ block of matrix $V_T$:

$$C_T(\alpha) = \left\{ \theta_1 \in \mathbb{R}^{d_1} : (\hat{\theta}_{T,1}^{\text{WD}} - \theta_1)^\top V_{T,1}^{-1}(\hat{\theta}_{T,1}^{\text{WD}} - \theta_1) \leq \frac{d_1(T-1)}{T-d_1} F_{d_1,T-d_1}(1-\alpha) \right\}.$$

### A.2.4 Self-Normalized Martingale Bound

We construct $1 - \alpha$ confidence region using the following equation taken from Theorem 2 of Abbasi-Yadkori et al. [2011]:

$$C_T(\alpha) = \left\{ \theta \in \Theta : (\hat{\theta}_T - \theta)^\top V_T(\hat{\theta}_T - \theta) \leq \sigma \sqrt{2 \log \left( \frac{\det(V_T)^{1/2} \det(\lambda I_d)^{-1/2}}{\alpha} \right)} + \lambda^{1/2} S \right\}.$$

- $\hat{\theta}_T = \left( \lambda I_d + \sum_{t=1}^T Z_t Z_t^\top \right)^{-1} \sum_{t=1}^T Z_t R_t$.
- $V_T = I_d \lambda + \sum_{t=1}^T Z_t Z_t^\top$.
- $\lambda = 1$ (ridge regression regularization parameter).
- $\sigma = 1$ (assumes rewards are $\sigma$-subgaussian).
- $S = 6$, where it is assumed that $\|\theta^*(\mathcal{P})\| \leq S$ (recall that in our simulations $\theta^*(\mathcal{P}) \in \mathbb{R}^6$).
- $\Theta = \{ \theta \in \mathbb{R}^6 : \|\theta\|_2 \leq 6 \}$.
- For constructing confidence regions for $\theta^*(\mathcal{P})$, we use projected confidence regions.

### A.2.5 Construction of Projected Confidence Regions

We are interested in getting the confidence ellipsoid of the projection of a $d$-dimensional ellipsoid onto $p$-dimensional space, for $p < d$.

- Defining the original $d$-dimensional ellipsoid, for $\mathbf{x} \in \mathbb{R}^d$ and $\mathbf{B} \in \mathbb{R}^{d \times d}$:

$$\mathbf{x}^\top \mathbf{B} \mathbf{x} = 1$$

- Partitioning the matrix $\mathbf{B}$ and vector $\mathbf{x}$:
  For $y \in \mathbb{R}^{d-p}$ and $z \in \mathbb{R}^p$.

$$\mathbf{x} = \begin{bmatrix} \mathbf{y} \\ \mathbf{z} \end{bmatrix}$$

For $\mathbf{C} \in \mathbb{R}^{d-p \times d-p}$, $\mathbf{E} \in \mathbb{R}^{p \times p}$, and $\mathbf{D} \in \mathbb{R}^{d-p \times p}$.

$$\mathbf{B} = \begin{bmatrix} \mathbf{C} & \mathbf{D} \\ \mathbf{D}^\top & \mathbf{E} \end{bmatrix}$$

- Gradient of $\mathbf{x}^\top \mathbf{B} \mathbf{x}$ with respect to $\mathbf{x}$:

$$(\mathbf{B} + \mathbf{B}^\top)\mathbf{x} = 2\mathbf{B}\mathbf{x} = \begin{bmatrix} \mathbf{C} & \mathbf{D} \\ \mathbf{D}^\top & \mathbf{E} \end{bmatrix} \begin{bmatrix} \mathbf{y} \\ \mathbf{z} \end{bmatrix}.$$

Since we are projecting onto the p-dimensional space, our projection is such that the gradient of $\mathbf{x}^\top \mathbf{B} \mathbf{x}$ with respect to $\mathbf{y}$ is zero, which means

$$\mathbf{C}\mathbf{y} + \mathbf{D}\mathbf{z} = 0.$$

This means in the projection that $\mathbf{y} = -\mathbf{C}^{-1}\mathbf{D}\mathbf{z}$.

- Returning to our definition of the ellipsoid, plugging in $\mathbf{z}$, we have that

$$1 = \mathbf{x}^\top \mathbf{B} \mathbf{x} = \begin{bmatrix} \mathbf{y}^\top & \mathbf{z}^\top \end{bmatrix} \begin{bmatrix} \mathbf{C} & \mathbf{D} \\ \mathbf{D}^\top & \mathbf{E} \end{bmatrix} \begin{bmatrix} \mathbf{y} \\ \mathbf{z} \end{bmatrix} = \mathbf{y}^\top \mathbf{C} \mathbf{y} + 2\mathbf{z}^\top \mathbf{D}^\top \mathbf{y} + \mathbf{z}^\top \mathbf{E} \mathbf{z}$$

$$= (\mathbf{C}^{-1}\mathbf{D}\mathbf{z})^\top \mathbf{C}(\mathbf{C}^{-1}\mathbf{D}\mathbf{z}) - 2\mathbf{z}^\top \mathbf{D}^\top (\mathbf{C}^{-1}\mathbf{D}\mathbf{z}) + \mathbf{z}^\top \mathbf{E} \mathbf{z}$$

$$= \mathbf{z}^\top \mathbf{D}^\top \mathbf{C}^{-1}\mathbf{D}\mathbf{z} - 2\mathbf{z}^\top \mathbf{D}^\top \mathbf{C}^{-1}\mathbf{D}\mathbf{z} + \mathbf{z}^\top \mathbf{E} \mathbf{z}$$

$$= \mathbf{z}^\top (\mathbf{E} - \mathbf{D}^\top \mathbf{C}^{-1}\mathbf{D})\mathbf{z}.$$

Thus the equation for the final projected ellipsoid is

$$\mathbf{z}^\top (\mathbf{E} - \mathbf{D}^\top \mathbf{C}^{-1}\mathbf{D})\mathbf{z} = 1.$$

### A.3 Additional Simulation Results

In addition to the continuous reward and a binary reward settings, here we also consider a discrete count reward setting. In this discrete reward setting, the reward $R_t$ is generated from a Poisson distribution with expectation $\mathbb{E}_{\mathcal{P}}[R_t|X_t, A_t] = \exp(\tilde{X}_t^\top \theta_0^*(\mathcal{P}) - A_t \tilde{X}_t^\top \theta_1^*(\mathcal{P}))$. All other data generation methods are equivalent to those used for the other simulation settings. Additionally we will consider the setting in which $\theta^*(\mathcal{P}) = [0.1, 0.1, 0.1, 0.2, 0.1, 0]$ for the continuous reward, binary reward, and discrete count settings.

To analyze the data, in the discrete count reward setting, we assume a correctly specified model for the expected reward. We use both unweighted and adaptively weighted maximum likelihood estimators (MLEs), which correspond to an M-estimators with $m_\theta(R_t, X_t, A_t)$ set to the negative log-likelihood of $R_t$ given $X_t, A_t$. We solve for these estimators using Newton–Raphson optimization and do not put explicit bounds on the parameter space $\Theta$.

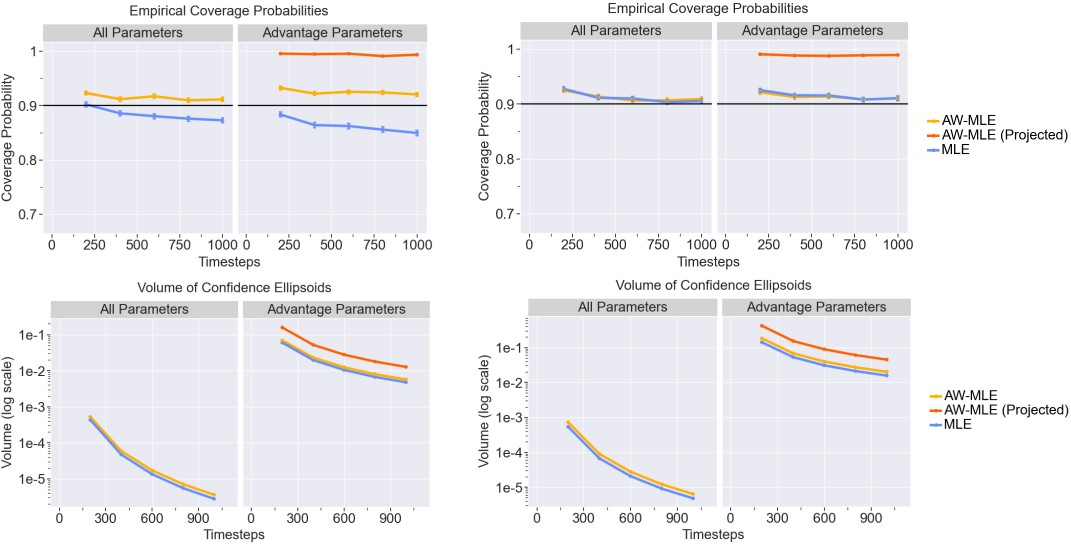

Figure 3: **Poisson Rewards:** Empirical coverage probabilities for 90% confidence ellipsoids for parameters $\theta^*(\mathcal{P})$ and parameters $\theta_1^*(\mathcal{P})$ (top row). We also plot the volumes of these 90% confidence ellipsoids for $\theta^*(\mathcal{P})$ and parameters $\theta_1^*(\mathcal{P})$ (bottom row). We set the true parameters to $\theta^*(\mathcal{P}) = [0.1, 0.1, 0.1, 0, 0, 0]$ (left) and to $\theta^*(\mathcal{P}) = [0.1, 0.1, 0.1, 0.2, 0.1, 0]$ (right).

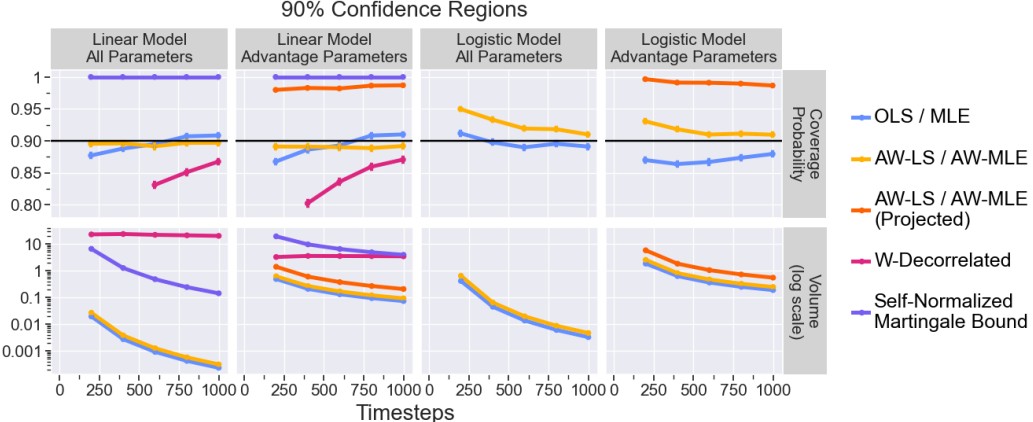

Figure 4: Empirical coverage probabilities (upper row) and volume (lower row) of 90% confidence ellipsoids. In these simulations, $\theta^*(\mathcal{P}) = [0.1, 0.1, 0.1, 0.2, 0.1, 0]$. The left two columns are for the linear reward model setting (t-distributed rewards) and the right two columns are for the logistic regression model setting (Bernoulli rewards). We consider confidence ellipsoids for all parameters $\theta^*(\mathcal{P})$ and for advantage parameters $\theta_1^*(\mathcal{P})$ for both settings.

In Figure 5, we plot the mean squared errors of all estimators for all three simulation settings (same simulation hyperparameters as described previously for the respective simulation settings).

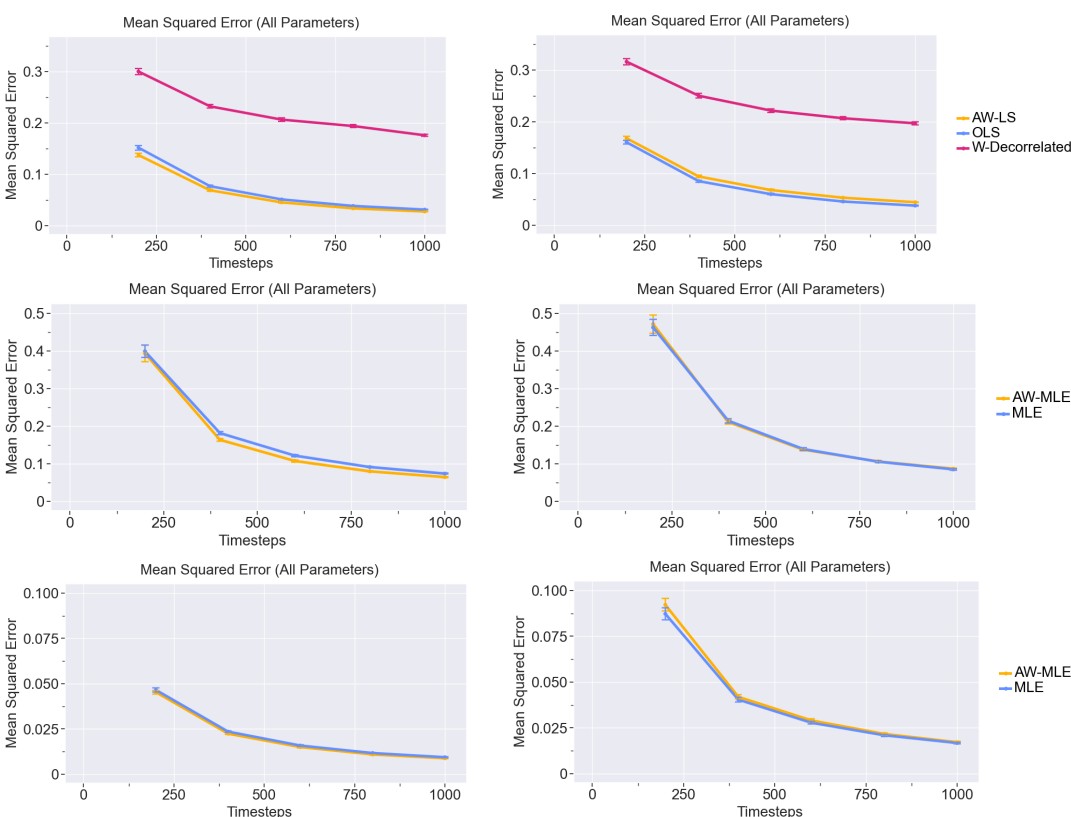

Figure 5: Mean squared error estimators of $\theta^*(\mathcal{P})$ for linear model (top), logistic regression model (middle), and generalized linear model for Poisson rewards (bottom). We consider simulations with $\theta^*(\mathcal{P}) = [0.1, 0.1, 0.1, 0, 0, 0]$ (left) and simulations with $\theta^*(\mathcal{P}) = [0.1, 0.1, 0.1, 0.2, 0.1, 0]$ (right).

# B Asymptotic Results

Throughout, $\|\cdot\|$ refers to the $L_2$ norm.

## B.1 Definitions

Here we define convergence in probability and distribution that is uniform over the true parameter. We follow the definitions are based on those in Kasy [2019] and Van Der Vaart and Wellner [1996, Chapter 1.12].

**Definition 1** (Uniform Convergence in Probability). *Let $\{Z_T(\mathcal{P})\}_{T\geq 1}$ be a sequence of random variables whose distributions are defined by some $\mathcal{P} \in \mathbf{P}$ and some nuisance component $\eta$. We say that $Z_T(\mathcal{P}) \xrightarrow{P} c$ uniformly over $\mathcal{P} \in \mathbf{P}$ as $T \to \infty$ if for any $\epsilon > 0$,*

$$\sup_{\mathcal{P}\in\mathbf{P}} \mathbb{P}_{\mathcal{P},\eta}\left(\|Z_T(\mathcal{P}) - c\| > \epsilon\right) \to 0. \tag{12}$$

*For simplicity of notation, throughout we denote $Z_T(\mathcal{P}) - c = o_{\mathcal{P}\in\mathbf{P}}(1)$ to mean $Z_T(\mathcal{P}) \xrightarrow{P} c$ uniformly over $\mathcal{P} \in \mathbf{P}$ as $T \to \infty$.*

**Definition 2** (Uniformly Stochastically Bounded). *Let $\{Z_T(\mathcal{P})\}_{T\geq 1}$ be a sequence of random variables whose distributions are defined by some $\mathcal{P} \in \mathbf{P}$ and some nuisance component $\eta$. We say that $Z_T(\mathcal{P})$ is uniformly stochastically bounded over $\mathcal{P} \in \mathbf{P}$ as $T \to \infty$ if for any $\epsilon > 0$ there exists some $k < \infty$ such that*

$$\limsup_{T\to\infty} \sup_{\mathcal{P}\in\mathbf{P}} \mathbb{P}_{\mathcal{P},\eta}\left(\|Z_T(\mathcal{P})\| > k\right) < \epsilon.$$

*Similarly we denote $Z_T(P) = O_{\mathcal{P}\in\mathbf{P}}(1)$ to mean $Z_T(\mathcal{P})$ is stochastically bounded uniformly over $\mathcal{P} \in \mathbf{P}$ as $T \to \infty$.*

**Definition 3** (Uniform Convergence in Distribution). *Let $Z(\mathcal{P}) \in \mathbb{R}^{d_z}$ and $\{Z_T(\mathcal{P})\}_{T\geq 1} \in \mathbb{R}^{d_z}$ be a sequence of random variables whose distributions are defined by some $\mathcal{P} \in \mathbf{P}$ and some nuisance component $\eta$. We say that $Z_T(\mathcal{P}) \xrightarrow{D} Z(\mathcal{P})$ uniformly over $\mathcal{P} \in \mathbf{P}$ as $T \to \infty$ if*

$$\sup_{\mathcal{P}\in\mathbf{P}} \sup_{f\in BL_1} \left| \mathbb{E}_{\mathcal{P},\eta}\left[f\left(Z_T(\mathcal{P})\right)\right] - \mathbb{E}_{\mathcal{P},\eta}\left[f\left(Z(\mathcal{P})\right)\right] \right| \to 0, \tag{13}$$

*where $BL_1$ is the set of functions $f : \mathbb{R}^{d_z} \to \mathbb{R}$ with $\|f(z)\|_\infty \leq 1$ and $|f(z) - f(z')| \leq \|z - z'\|$ for all $z, z' \in \mathbb{R}^{d_Z}$.*

As discussed in Kasy [2019], Equation (12) holds if and only if for any $\epsilon > 0$ and any sequence $\{\mathcal{P}_T\}_{T\geq 1}$ such that $\mathcal{P}_T \in \mathbf{P}$ for all $T \geq 1$, $\mathbb{P}_{\mathcal{P}_T,\eta}\left(\|Z_T(\mathcal{P}_T) - c\| > \epsilon\right) \to 0$.

Similarly, Equation (13) holds if and only if for any sequence $\{\mathcal{P}_T\}_{T\geq 1}$ such that $\mathcal{P}_T \in \mathbf{P}$ for all $T \geq 1$, $\sup_{f\in BL_1} \left| \mathbb{E}_{\mathcal{P}_T,\eta}\left[f\left(Z_T(\mathcal{P}_T)\right)\right] - \mathbb{E}_{\mathcal{P}_T,\eta}\left[f\left(Z(\mathcal{P}_T)\right)\right] \right| \to 0$.

## B.2 Consistency

We prove the first part of Theorem 1, i.e., that $\hat{\theta}_T \xrightarrow{P} \theta^*(\mathcal{P})$ uniformly over $\mathcal{P} \in \mathbf{P}$. We abbreviate $m_\theta(Y_t, X_t, A_t)$ with $m_{\theta,t}$. By definition of $\hat{\theta}_T$,

$$\sum_{t=1}^{T} W_t m_{\hat{\theta}_T,t} = \sup_{\theta\in\Theta} \sum_{t=1}^{T} W_t m_{\theta,t} \geq \sum_{t=1}^{T} W_t m_{\theta^*(\mathcal{P}),t}.$$

Note that $\|\hat{\theta}_T - \theta^*(\mathcal{P})\| > \epsilon > 0$ implies that

$$\sup_{\theta\in\Theta:\|\theta-\theta^*(\mathcal{P})\|>\epsilon} \sum_{t=1}^{T} W_t m_{\theta,t} = \sup_{\theta\in\Theta} \sum_{t=1}^{T} W_t m_{\theta,t}.$$

Thus, the above two results imply the following inequality:

$$\sup_{\mathcal{P}\in\mathbf{P}} \mathbb{P}_{\mathcal{P},\pi}\left(\|\hat{\theta}_T - \theta^*(\mathcal{P})\| > \epsilon\right) \leq \sup_{\mathcal{P}\in\mathbf{P}} \mathbb{P}_{\mathcal{P},\pi}\left(\sup_{\theta\in\Theta:\|\theta-\theta^*(\mathcal{P})\|>\epsilon} \sum_{t=1}^{T} W_t m_{\theta,t} \geq \sum_{t=1}^{T} W_t m_{\theta^*(\mathcal{P}),t}\right)$$

$$= \sup_{\mathcal{P}\in\mathbf{P}} \mathbb{P}_{\mathcal{P},\pi}\left(\sup_{\theta\in\Theta:\|\theta-\theta^*(\mathcal{P})\|>\epsilon}\left\{\frac{1}{T}\sum_{t=1}^{T}W_t m_{\theta,t}\right\} - \frac{1}{T}\sum_{t=1}^{T}W_t m_{\theta^*(\mathcal{P}),t}\geq 0\right)$$

$$= \sup_{\mathcal{P}\in\mathbf{P}} \mathbb{P}_{\mathcal{P},\pi}\left(\sup_{\theta\in\Theta:\|\theta-\theta^*(\mathcal{P})\|>\epsilon}\left\{\frac{1}{T}\sum_{t=1}^{T}W_t m_{\theta,t} - \mathbb{E}_{\mathcal{P},\pi}[W_t m_{\theta,t}|\mathcal{H}_{t-1}] + \mathbb{E}_{\mathcal{P},\pi}[W_t m_{\theta,t}|\mathcal{H}_{t-1}]\right\}\right.$$
$$\left. - \frac{1}{T}\sum_{t=1}^{T}\left\{W_t m_{\theta^*(\mathcal{P}),t} - \mathbb{E}_{\mathcal{P},\pi}[W_t m_{\theta^*(\mathcal{P}),t}|\mathcal{H}_{t-1}] + \mathbb{E}_{\mathcal{P},\pi}[W_t m_{\theta^*(\mathcal{P}),t}|\mathcal{H}_{t-1}]\right\}\geq 0\right).$$

By triangle inequality,

$$\leq \sup_{\mathcal{P}\in\mathbf{P}} \mathbb{P}_{\mathcal{P},\pi}\left(\underbrace{\sup_{\theta\in\Theta:\|\theta-\theta^*(\mathcal{P})\|>\epsilon}\left\{\frac{1}{T}\sum_{t=1}^{T}\left(W_t m_{\theta,t} - \mathbb{E}_{\mathcal{P},\pi}[W_t m_{\theta,t}|\mathcal{H}_{t-1}]\right)\right\}}_{(a)}\right.$$

$$+ \underbrace{\sup_{\theta\in\Theta:\|\theta-\theta^*(\mathcal{P})\|>\epsilon}\left\{\frac{1}{T}\sum_{t=1}^{T}\mathbb{E}_{\mathcal{P},\pi}\left[W_t(m_{\theta,t} - m_{\theta^*(\mathcal{P}),t})\big|\mathcal{H}_{t-1}\right]\right\}}_{(b)}$$

$$\left. - \underbrace{\frac{1}{T}\sum_{t=1}^{T}\left\{W_t m_{\theta^*(\mathcal{P}),t} - \mathbb{E}_{\mathcal{P},\pi}[W_t m_{\theta^*(\mathcal{P}),t}|\mathcal{H}_{t-1}]\right\}}_{(c)}\geq 0\right) \to 0. \quad (14)$$

We now show that the limit in Equation (14) above holds.

- Regarding term (c), by moment bounds of Condition 5 and Lemma 1,
  $\frac{1}{T}\sum_{t=1}^{T}\left\{W_t m_{\theta^*(\mathcal{P}),t} - \mathbb{E}_{\mathcal{P},\pi}[W_t m_{\theta^*(\mathcal{P}),t}|\mathcal{H}_{t-1}]\right\} = o_{\mathcal{P}\in\mathbf{P}}(1).$
- Regarding term (a), by Lemma 2,
  $\sup_{\theta\in\Theta:\|\theta-\theta^*(\mathcal{P})\|>\epsilon}\left\{\frac{1}{T}\sum_{t=1}^{T}\left(W_t m_{\theta,t} - \mathbb{E}_{\mathcal{P},\pi}[W_t m_{\theta,t}|\mathcal{H}_{t-1}]\right)\right\} = o_{\mathcal{P}\in\mathbf{P}}(1).$

Thus it is sufficient to show that term (b) is such that for some $\delta' > 0$,

$$\sup_{\theta\in\Theta:\|\theta-\theta^*(\mathcal{P})\|>\epsilon}\left\{\frac{1}{T}\sum_{t=1}^{T}\mathbb{E}_{\mathcal{P},\pi}[W_t(m_{\theta,t} - m_{\theta^*(\mathcal{P}),t})|\mathcal{H}_{t-1}]\right\}\leq -\delta' \text{ w.p. } 1. \quad (15)$$

By law of iterated expectations,

$$\sup_{\theta\in\Theta:\|\theta-\theta^*(\mathcal{P})\|>\epsilon}\left\{\frac{1}{T}\sum_{t=1}^{T}\mathbb{E}_{\mathcal{P},\pi}[W_t(m_{\theta,t} - m_{\theta^*(\mathcal{P}),t})|\mathcal{H}_{t-1}]\right\}$$

$$= \sup_{\theta\in\Theta:\|\theta-\theta^*(\mathcal{P})\|>\epsilon}\left\{\frac{1}{T}\sum_{t=1}^{T}\mathbb{E}_{\mathcal{P}}\left[\int_{a\in\mathcal{A}}\pi_t(a,X_t,\mathcal{H}_{t-1})\mathbb{E}_{\mathcal{P}}[W_t(m_{\theta,t} - m_{\theta^*(\mathcal{P}),t})|\mathcal{H}_{t-1},X_t,A_t = a]da\Big|\mathcal{H}_{t-1}\right]\right\}.$$

Since $W_t \in \sigma(\mathcal{H}_{t-1}, X_t, A_t)$, we have that $\mathbb{E}_{\mathcal{P}}[W_t(m_{\theta,t} - m_{\theta^*(\mathcal{P}),t})|\mathcal{H}_{t-1},X_t,A_t = a] = W_t\mathbb{E}_{\mathcal{P}}[m_{\theta,t} - m_{\theta^*(\mathcal{P}),t}|\mathcal{H}_{t-1},X_t,A_t = a]$. By Condition 1, we have that $W_t\mathbb{E}_{\mathcal{P}}[m_{\theta,t} - m_{\theta^*(\mathcal{P}),t}|\mathcal{H}_{t-1},X_t,A_t = a] = W_t\mathbb{E}_{\mathcal{P}}[m_{\theta,t} - m_{\theta^*(\mathcal{P}),t}|X_t,A_t = a]$. Thus we have,

$$= \sup_{\theta\in\Theta:\|\theta-\theta^*(\mathcal{P})\|>\epsilon}\left\{\frac{1}{T}\sum_{t=1}^{T}\mathbb{E}_{\mathcal{P}}\left[\int_{a\in\mathcal{A}}\pi_t(a,X_t,\mathcal{H}_{t-1})W_t\mathbb{E}_{\mathcal{P}}[m_{\theta,t} - m_{\theta^*(\mathcal{P}),t}|X_t,A_t = a]da\Big|\mathcal{H}_{t-1}\right]\right\}.$$

Since for all $\theta \in \Theta$, $\mathbb{E}_{\mathcal{P}}[m_{\theta,t} - m_{\theta^*(\mathcal{P}),t}|X_t,A_t] \leq 0$ with probability 1 by Condition 7 and since $0 < \frac{W_t}{\sqrt{\rho_{\max}}} \leq 1$ with probability 1 by Condition 9,

$$\leq \sup_{\theta\in\Theta:\|\theta-\theta^*(\mathcal{P})\|>\epsilon}\left\{\frac{1}{T\sqrt{\rho_{\max}}}\sum_{t=1}^{T}\mathbb{E}_{\mathcal{P}}\left[\int_{a\in\mathcal{A}}\pi_t(a,X_t,\mathcal{H}_{t-1})W_t^2\mathbb{E}_{\mathcal{P}}[m_{\theta,t} - m_{\theta^*(\mathcal{P}),t}|X_t,A_t = a]da\Big|\mathcal{H}_{t-1}\right]\right\}.$$

Since $W_t^2 = \frac{\pi_t^{\text{sta}}(A_t, X_t)}{\pi_t(A_t, X_t, \mathcal{H}_{t-1})}$,

$$= \sup_{\theta \in \Theta : \|\theta - \theta^*(\mathcal{P})\| > \epsilon} \left\{ \frac{1}{T\sqrt{\rho_{\max}}} \sum_{t=1}^{T} \mathbb{E}_{\mathcal{P}} \left[ \int_{a \in \mathcal{A}} \pi_t^{\text{sta}}(a, X_t) \mathbb{E}_{\mathcal{P}}[m_{\theta,t} - m_{\theta^*(\mathcal{P}),t} | X_t, A_t = a] da \Big| \mathcal{H}_{t-1} \right] \right\}.$$

By Condition 1 and since $\pi_t^{\text{sta}}$ is pre-specified, we can drop the conditioning on $\mathcal{H}_{t-1}$, i.e.,

$$= \sup_{\theta \in \Theta : \|\theta - \theta^*(\mathcal{P})\| > \epsilon} \left\{ \frac{1}{T\sqrt{\rho_{\max}}} \sum_{t=1}^{T} \mathbb{E}_{\mathcal{P}} \left[ \int_{a \in \mathcal{A}} \pi_t^{\text{sta}}(a, X_t) \mathbb{E}_{\mathcal{P}}[m_{\theta,t} - m_{\theta^*(\mathcal{P}),t} | X_t, A_t = a] da \right] \right\}.$$

By law of iterated expectations,

$$= \sup_{\theta \in \Theta : \|\theta - \theta^*(\mathcal{P})\| > \epsilon} \left\{ \frac{1}{T\sqrt{\rho_{\max}}} \sum_{t=1}^{T} \mathbb{E}_{\mathcal{P}, \pi_t^{\text{sta}}} \left[ m_{\theta,t} - m_{\theta^*(\mathcal{P}),t} \right] \right\} \leq -\frac{1}{\sqrt{\rho_{\max}}} \delta.$$

The last inequality above holds for some $\delta > 0$ for all sufficiently large $T$ by Condition 8. Thus Equation (15) holds for $\delta' = \frac{1}{\sqrt{\rho_{\max}}} \delta$.

## B.3 Asymptotic Normality

We prove the second part of Theorem 1, i.e., that

$$\Sigma_T(\mathcal{P})^{-1/2} \ddot{M}_T(\hat{\theta}_T) \sqrt{T}(\hat{\theta}_T - \theta^*(\mathcal{P})) \xrightarrow{D} \mathcal{N}(0, I_d) \text{ uniformly over } \mathcal{P} \in \mathbf{P}. \tag{16}$$

### B.3.1 Main Argument

The three results we show to ensure Equation (16) holds are as follows:

$$\Sigma_T(\mathcal{P})^{-1/2} \sqrt{T} \dot{M}_T(\theta^*(\mathcal{P})) \xrightarrow{D} \mathcal{N}(0, I_d) \text{ uniformly over } \mathcal{P} \in \mathbf{P}. \tag{17}$$

For $\ddot{\epsilon}_{\ddot{m}} > 0$ as defined in Condition 6,

$$\sup_{\theta \in \Theta : \|\theta - \theta^*(\mathcal{P})\| \leq \epsilon_{\dddot{m}}} \left\| \dddot{M}_T(\theta) \right\|_1 = O_{\mathcal{P} \in \mathbf{P}}(1). \tag{18}$$

For matrix $H$ positive definite,

$$-\ddot{M}_T(\theta^*(\mathcal{P})) \succeq H + o_{\mathcal{P} \in \mathbf{P}}(1). \tag{19}$$

For a reminder on the notation of $o_{\mathcal{P} \in \mathbf{P}}(1)$ and $O_{\mathcal{P} \in \mathbf{P}}(1)$ see definitions 12 and 2. For now, we assume that Equations (17), (18), and (19) hold; we will show they hold in Sections B.3.2, B.3.3, and B.3.4 respectively. Our argument is based on Van der Vaart [2000, Theorem of 5.41].

By differentiability Condition 2, since $\hat{\theta}_T$ is the maximizer of criterion $M_T(\theta)$,

$$0 = \dot{M}_T(\hat{\theta}_T).$$

By differentiability Condition 2 again and Taylor's theorem we have that for some random $\tilde{\theta}_T$ on the line segment between $\theta^*(\mathcal{P})$ and $\hat{\theta}_T$,

$$0 = \dot{M}_T(\hat{\theta}_T) = \dot{M}_T(\theta^*(\mathcal{P})) + \ddot{M}_T(\theta^*(\mathcal{P}))(\hat{\theta}_T - \theta^*(\mathcal{P})) + \frac{1}{2}(\hat{\theta}_T - \theta^*(\mathcal{P}))^\top \dddot{M}_T(\tilde{\theta}_T)(\hat{\theta}_T - \theta^*(\mathcal{P})).$$

By rearranging terms and multiplying by $\sqrt{T}$,

$$-\sqrt{T}\dot{M}_T(\theta^*(\mathcal{P})) = \ddot{M}_T(\theta^*(\mathcal{P}))\sqrt{T}(\hat{\theta}_T - \theta^*(\mathcal{P})) + \frac{1}{2}(\hat{\theta}_T - \theta^*(\mathcal{P}))^\top \dddot{M}_T(\tilde{\theta}_T)\sqrt{T}(\hat{\theta}_T - \theta^*(\mathcal{P}))$$

$$= \left[ \ddot{M}_T(\theta^*(\mathcal{P})) + \frac{1}{2}(\hat{\theta}_T - \theta^*(\mathcal{P}))^\top \dddot{M}_T(\tilde{\theta}_T) \right] \sqrt{T}(\hat{\theta}_T - \theta^*(\mathcal{P})).$$

Note that by the above equation and Equation (17), we have that

$$\Sigma_T(\mathcal{P})^{-1/2}\left[\ddot{M}_T(\theta^*(\mathcal{P})) + \frac{1}{2}(\hat{\theta}_T - \theta^*(\mathcal{P}))^\top \dddot{M}_T(\tilde{\theta}_T)\right]\sqrt{T}(\hat{\theta}_T - \theta^*(\mathcal{P}))$$

$$\xrightarrow{D} \mathcal{N}(0, I_d) \text{ uniformly over } \mathcal{P} \in \mathbf{P}. \quad (20)$$

By Equation (19), the probability that $\ddot{M}_T(\theta^*(\mathcal{P}))$ is invertible goes to 1 uniformly over $\mathcal{P} \in \mathbf{P}$. Thus by Equation (20), we have that

$$\Sigma_T(\mathcal{P})^{-1/2}\left[I_d + \frac{1}{2}(\hat{\theta}_T - \theta^*(\mathcal{P}))^\top \dddot{M}_T(\tilde{\theta}_T)\ddot{M}_T(\theta^*(\mathcal{P}))^{-1}\right]\ddot{M}_T(\theta^*(\mathcal{P}))\sqrt{T}(\hat{\theta}_T - \theta^*(\mathcal{P}))$$

$$= \left[I_d + \frac{1}{2}\Sigma_T(\mathcal{P})^{-1/2}(\hat{\theta}_T - \theta^*(\mathcal{P}))^\top \dddot{M}_T(\tilde{\theta}_T)\ddot{M}_T(\theta^*(\mathcal{P}))^{-1}\Sigma_T(\mathcal{P})^{1/2}\right]$$

$$\Sigma_T(\mathcal{P})^{-1/2}\ddot{M}_T(\theta^*(\mathcal{P}))\sqrt{T}(\hat{\theta}_T - \theta^*(\mathcal{P})) \xrightarrow{D} \mathcal{N}(0, I_d) \text{ uniformly over } \mathcal{P} \in \mathbf{P}. \quad (21)$$

We now show that $\frac{1}{2}\Sigma_T(\mathcal{P})^{-1/2}(\hat{\theta}_T - \theta^*(\mathcal{P}))^\top \dddot{M}_T(\tilde{\theta}_T)\ddot{M}_T(\theta^*(\mathcal{P}))^{-1}\Sigma_T(\mathcal{P})^{1/2} = o_{\mathcal{P}\in\mathbf{P}}(1)$. It is sufficient to show that $\|\Sigma_T(\mathcal{P})^{-1/2}\|\|\hat{\theta}_T - \theta^*(\mathcal{P})\|\|\dddot{M}_T(\tilde{\theta}_T)\|_1\|\ddot{M}_T(\theta^*(\mathcal{P}))^{-1}\|\|\Sigma_T(\mathcal{P})^{1/2}\| = o_{\mathcal{P}\in\mathbf{P}}(1)$.

- By Condition 5, the minimum eigenvalue of $\Sigma_T(\mathcal{P})$ is bounded uniformly above some constant greater than zero, so $\sup_{\mathcal{P}\in\mathbf{P}}\|\Sigma_T(\mathcal{P})^{-1/2}\| = O(1)$.

- By uniform consistency of $\hat{\theta}_T$, $\|\hat{\theta}_T - \theta^*(\mathcal{P})\| = o_{\mathcal{P}\in\mathbf{P}}(1)$.

- By uniform consistency of $\hat{\theta}_T$, $\mathbb{1}_{\|\tilde{\theta}_T - \theta^*(\mathcal{P})\| \le \epsilon_{\tilde{m}}} = o_{\mathcal{P}\in\mathbf{P}}(1)$. Thus by Equation (18), $\dddot{M}_T(\tilde{\theta}_T) = O_{\mathcal{P}\in\mathbf{P}}(1)$.

- By Equation (19), the minimum eigenvalue of $-\ddot{M}_T(\theta^*(\mathcal{P}))^{-1}$ is bounded above that of positive definite matrix $H$. Thus $\|\ddot{M}_T(\theta^*(\mathcal{P}))^{-1}\| = O_{\mathcal{P}\in\mathbf{P}}(1)$.

- By Condition 5, $\sup_{\mathcal{P}\in\mathbf{P}}\|\Sigma_T(\mathcal{P})^{1/2}\| = O(1)$.

Thus, by Slutsky's Theorem and Equation (21), we have that

$$\Sigma_T(\mathcal{P})^{-1/2}\ddot{M}_T(\theta^*(\mathcal{P}))\sqrt{T}(\hat{\theta}_T - \theta^*(\mathcal{P})) \xrightarrow{D} \mathcal{N}(0, I_d) \text{ uniformly over } \mathcal{P} \in \mathbf{P}. \quad (22)$$

Lastly, to show our desired result, that $\Sigma_T(\mathcal{P})^{-1/2}\ddot{M}_T(\hat{\theta}_T)\sqrt{T}(\hat{\theta}_T - \theta^*(\mathcal{P})) \xrightarrow{D} \mathcal{N}(0, I_d)$ uniformly over $\mathcal{P} \in \mathbf{P}$, by Equation (22) and Slutsky's Theorem it is sufficient to show that $\Sigma_T(\mathcal{P})^{-1/2}\ddot{M}_T(\hat{\theta}_T)\ddot{M}_T(\theta^*(\mathcal{P}))^{-1}\Sigma_T(\mathcal{P})^{1/2} \xrightarrow{P} I_d$ uniformly over $\mathcal{P} \in \mathbf{P}$. Note if we can show that $\ddot{M}_T(\hat{\theta}_T)\ddot{M}_T(\theta^*(\mathcal{P}))^{-1} \xrightarrow{P} I_d$ uniformly over $\mathcal{P} \in \mathbf{P}$, then $\Sigma_T(\mathcal{P})^{-1/2}\ddot{M}_T(\hat{\theta}_T)\ddot{M}_T(\theta^*(\mathcal{P}))^{-1}\Sigma_T(\mathcal{P})^{1/2} = \Sigma_T(\mathcal{P})^{-1/2}\left[I_d + o_{\mathcal{P}\in\mathbf{P}}(1)\right]\Sigma_T(\mathcal{P})^{1/2} = I_d + \Sigma_T(\mathcal{P})^{-1/2}o_{\mathcal{P}\in\mathbf{P}}(1)\Sigma_T(\mathcal{P})^{1/2} = I_d + o_{\mathcal{P}\in\mathbf{P}}(1)$. The last limit holds since $\|\Sigma_T(\mathcal{P})^{-1/2}\| = O_{\mathcal{P}\in\mathbf{P}}(1)$ and $\|\Sigma_T(\mathcal{P})^{1/2}\| = O_{\mathcal{P}\in\mathbf{P}}(1)$ by Condition 5 (use the same argument as that used in the bullet points below Equation (21)).

Thus it is sufficient to show that $\ddot{M}_T(\hat{\theta}_T)\ddot{M}_T(\theta^*(\mathcal{P}))^{-1} \xrightarrow{P} I_d$ uniformly over $\mathcal{P} \in \mathbf{P}$. By Taylor's Theorem, for some random $\bar{\theta}_T$ on the line segment between $\hat{\theta}_T$ and $\theta^*(\mathcal{P})$,

$$\ddot{M}_T(\hat{\theta}_T) = \ddot{M}_T(\theta^*(\mathcal{P})) + \dddot{M}_T(\bar{\theta}_T)(\hat{\theta}_T - \theta^*(\mathcal{P})).$$

Recall that the probability the inverse of $\ddot{M}_T(\theta^*(\mathcal{P}))$ exists goes to 1 by Equation (19) (use the same argument as that used in the bullet points below Equation (21)). Thus we have that $\ddot{M}_T(\hat{\theta}_T)\ddot{M}_T(\theta^*(\mathcal{P}))^{-1}$ equals the following:

$$\left[\ddot{M}_T(\theta^*(\mathcal{P})) + \dddot{M}_T(\bar{\theta}_T)(\hat{\theta}_T - \theta^*(\mathcal{P}))\right]\ddot{M}_T(\theta^*(\mathcal{P}))^{-1}$$

$$= I_d + \dddot{M}_T(\bar{\theta}_T)(\hat{\theta}_T - \theta^*(\mathcal{P}))\ddot{M}_T(\theta^*(\mathcal{P}))^{-1}$$

Note that $\dddot{M}_T(\bar{\theta}_T)(\hat{\theta}_T - \theta^*(\mathcal{P}))\ddot{M}_T(\theta^*(\mathcal{P}))^{-1} = o_{\mathcal{P}\in\mathbf{P}}(1)$ because

- By uniform consistency of $\hat\theta_T$, $\mathbb{1}_{\|\tilde\theta_T - \theta^*(\mathcal{P})\| \le \epsilon_{\dddot m}} = o_{\mathcal{P}\in\mathbf{P}}(1)$. Thus by Equation (18), $\dddot M_T(\tilde\theta_T) = O_{\mathcal{P}\in\mathbf{P}}(1)$.

- By uniform consistency of $\hat\theta_T$, $\|\hat\theta_T - \theta^*(\mathcal{P})\| = o_{\mathcal{P}\in\mathbf{P}}(1)$.

- By Equation (19), $\|\ddot M_T(\theta^*(\mathcal{P}))^{-1}\| = O_{\mathcal{P}\in\mathbf{P}}(1)$.

### B.3.2  Asymptotic Normality of $\Sigma_T(\mathcal{P})^{-1/2}\sqrt{T}\dot M_T(\theta^*(\mathcal{P}))$

We will show that Equation (17) holds by applying a martingale central limit theorem. For notational convenience, we let $\dot m_{\theta,t} := \dot m_\theta(Y_t, X_t, A_t)$. Note that by definition $\Sigma_T(\mathcal{P})^{-1/2}\sqrt{T}\dot M_T(\theta^*(\mathcal{P})) = \Sigma_T(\mathcal{P})^{-1/2}\frac{1}{\sqrt{T}}\sum_{t=1}^T W_t \dot m_{\theta^*(\mathcal{P}),t}$. We first show that $\left\{\Sigma_T(\mathcal{P})^{-1/2}\frac{1}{\sqrt{T}}W_t \dot m_{\theta^*(\mathcal{P}),t}\right\}_{t=1}^T$ is a martingale difference sequence with respect to $\{\mathcal{H}_t\}_{t=0}^T$. For any $t \in [1:T]$,

$$\mathbb{E}_{\mathcal{P},\pi}\left[\frac{1}{\sqrt{T}}\Sigma_T(\mathcal{P})^{-1/2}W_t\mathbf{c}^\top \dot m_{\theta^*(\mathcal{P}),t}\middle|\mathcal{H}_{t-1}\right]$$

$$\underset{(a)}{=} \frac{1}{\sqrt{T}}\mathbb{E}_{\mathcal{P},\pi}\left[\mathbb{E}_{\mathcal{P}}\left[\Sigma_T(\mathcal{P})^{-1/2}W_t\mathbf{c}^\top \dot m_{\theta^*(\mathcal{P}),t}\middle|\mathcal{H}_{t-1}, X_t, A_t\right]\middle|\mathcal{H}_{t-1}\right]$$

$$\underset{(b)}{=} \frac{1}{\sqrt{T}}\Sigma_T(\mathcal{P})^{-1/2}\mathbb{E}_{\mathcal{P},\pi}\left[W_t\mathbf{c}^\top\mathbb{E}_{\mathcal{P}}\left[\dot m_{\theta^*(\mathcal{P}),t}\middle|\mathcal{H}_{t-1}, X_t, A_t\right]\middle|\mathcal{H}_{t-1}\right]\underset{(c)}{=} 0$$

- Above, (a) holds by law of iterated expectations.

- (b) holds since $W_t \in \sigma(\mathcal{H}_{t-1}, X_t, A_t)$ and since $\Sigma_T(\mathcal{P})$ are a function of stabilizing policies $\{\pi_t^{\mathrm{sta}}\}_{t\ge 1}$, which are pre-specified.

- By Condition 1, $\mathbb{E}_{\mathcal{P}}\left[\dot m_{\theta^*(\mathcal{P}),t}\middle|\mathcal{H}_{t-1}, X_t, A_t\right] = \mathbb{E}_{\mathcal{P}}\left[\dot m_{\theta^*(\mathcal{P}),t}\middle|X_t, A_t\right]$. Equality (c) holds because $\mathbb{E}_{\mathcal{P}}\left[\dot m_{\theta^*(\mathcal{P}),t}\middle|X_t, A_t\right] = 0$ with probability 1 by Condition 7; note that $\theta^*(\mathcal{P})$ is a critical point of $\mathbb{E}_{\mathcal{P}}[m_{\theta,t}|X_t, A_t]$.

By Cramer-Wold device, to show that Equation (17) holds, it is sufficient to show that for any fixed $\mathbf{c} \in \mathbb{R}^d$ with $\|\mathbf{c}\|_2 = 1$, that $\mathbf{c}^\top\Sigma_T(\mathcal{P})^{-1/2}\frac{1}{\sqrt{T}}\sum_{t=1}^T W_t \dot m_{\theta^*(\mathcal{P}),t} \overset{D}{\to} \mathcal{N}\left(0, \mathbf{c}^\top I_d \mathbf{c}\right)$ uniformly over $\mathcal{P} \in \mathbf{P}$. We now apply Theorem 2, a uniform version of the martingale central limit theorem of Dvoretzky [1972]; while the original theorem holds for any fixed $\mathcal{P}$, we can show uniform convergence in distribution by ensuring that the conditions of the theorem hold uniformly over $\mathcal{P} \in \mathbf{P}$ (see Definition 3). By Theorem 2, it is sufficient to show that the following two conditions hold:

1. **Conditional Variance:** $\frac{1}{T}\sum_{t=1}^T\mathbb{E}_{\mathcal{P},\pi}\left[\left\{\mathbf{c}^\top\Sigma_T(\mathcal{P})^{-1/2}W_t\dot m_{\theta^*(\mathcal{P}),t}\right\}^2\middle|\mathcal{H}_{t-1}\right] \overset{P}{\to} \sigma^2$ uniformly over $\mathcal{P} \in \mathbf{P}$.

2. **Conditional Lindeberg:** For any $\delta > 0$,
$\frac{1}{T}\sum_{t=1}^T\mathbb{E}_{\mathcal{P},\pi}\left[\left\{\mathbf{c}^\top\Sigma_T(\mathcal{P})^{-1/2}W_t\dot m_{\theta^*(\mathcal{P}),t}\right\}^2\mathbb{1}_{|\mathbf{c}^\top\Sigma_T(\mathcal{P})^{-1/2}W_t\dot m_{\theta^*(\mathcal{P}),t}|>\delta\sqrt{T}}\middle|\mathcal{H}_{t-1}\right] \overset{P}{\to} 0$ uniformly over $\mathcal{P} \in \mathbf{P}$.

**1. Conditional Variance**

$$\frac{1}{T}\sum_{t=1}^T\mathbb{E}_{\mathcal{P},\pi}\left[\left(\mathbf{c}^\top W_t\Sigma_T(\mathcal{P})^{-1/2}\dot m_{\theta^*(\mathcal{P}),t}\right)^2\middle|\mathcal{H}_{t-1}\right]$$

$$= \frac{1}{T}\sum_{t=1}^T\mathbb{E}_{\mathcal{P},\pi}\left[W_t^2\mathbf{c}^\top\Sigma_T(\mathcal{P})^{-1/2}\dot m_{\theta^*(\mathcal{P}),t}^{\otimes 2}\Sigma_T(\mathcal{P})^{-1/2}\mathbf{c}\middle|\mathcal{H}_{t-1}\right]$$

$$\underset{(a)}{=} \mathbf{c}^\top\Sigma_T(\mathcal{P})^{-1/2}\left\{\frac{1}{T}\sum_{t=1}^T\mathbb{E}_{\mathcal{P},\pi}\left[W_t^2\dot m_{\theta^*(\mathcal{P}),t}^{\otimes 2}\middle|\mathcal{H}_{t-1}\right]\right\}\Sigma_T(\mathcal{P})^{-1/2}\mathbf{c}$$

$$\underset{(b)}{=} \mathbf{c}^\top \Sigma_T(\mathcal{P})^{-1/2} \left\{ \frac{1}{T} \sum_{t=1}^T \mathbb{E}_\mathcal{P} \left[ \int_{a \in \mathcal{A}} \pi_t(a, X_t, \mathcal{H}_{t-1}) \mathbb{E}_\mathcal{P} \left[ W_t^2 \dot{m}_{\theta^*(\mathcal{P}),t}^{\otimes 2} | \mathcal{H}_{t-1}, X_t, A_t = a \right] da \middle| \mathcal{H}_{t-1} \right] \right\} \Sigma_T(\mathcal{P})^{-1/2} \mathbf{c}$$

$$\underset{(c)}{=} \mathbf{c}^\top \Sigma_T(\mathcal{P})^{-1/2} \left\{ \frac{1}{T} \sum_{t=1}^T \mathbb{E}_\mathcal{P} \left[ \int_{a \in \mathcal{A}} \pi_t^{\text{sta}}(a, X_t) \mathbb{E}_\mathcal{P} \left[ \dot{m}_{\theta^*(\mathcal{P}),t}^{\otimes 2} | \mathcal{H}_{t-1}, X_t, A_t = a \right] da \middle| \mathcal{H}_{t-1} \right] \right\} \Sigma_T(\mathcal{P})^{-1/2} \mathbf{c}$$

$$\underset{(d)}{=} \mathbf{c}^\top \Sigma_T(\mathcal{P})^{-1/2} \left\{ \frac{1}{T} \sum_{t=1}^T \mathbb{E}_\mathcal{P} \left[ \mathbb{E}_{\mathcal{P}, \pi_t^{\text{sta}}} \left[ \dot{m}_{\theta^*(\mathcal{P}),t}^{\otimes 2} | X_t \right] \middle| \mathcal{H}_{t-1} \right] \right\} \Sigma_T(\mathcal{P})^{-1/2} \mathbf{c}$$

$$\underset{(e)}{=} \mathbf{c}^\top \Sigma_T(\mathcal{P})^{-1/2} \left\{ \frac{1}{T} \sum_{t=1}^T \mathbb{E}_{\mathcal{P}, \pi_t^{\text{sta}}} \left[ \dot{m}_{\theta^*(\mathcal{P}),t}^{\otimes 2} \right] \right\} \Sigma_T(\mathcal{P})^{-1/2} \mathbf{c}$$

$$\underset{(f)}{=} \mathbf{c}^\top \Sigma_T(\mathcal{P})^{-1/2} \Sigma_T(P) \Sigma_T(\mathcal{P})^{-1/2} \mathbf{c} = \mathbf{c}^\top I_d \mathbf{c}$$

- Above, (a) holds since $\Sigma_T(\mathcal{P})$ are a function of stabilizing policies $\{\pi_t^{\text{sta}}\}_{t \geq 1}$, which are pre-specified.
- Equality (b) holds by law of iterated expectations.
- Equality (c) holds since $W_t = \sqrt{\frac{\pi_t^{\text{sta}}(A_t, X_t)}{\pi_t(A_t, X_t, \mathcal{H}_{t-1})}} \in \sigma(\mathcal{H}_{t-1}, X_t, A_t)$.
- Equality (d) holds because by Condition [1], $\mathbb{E}_\mathcal{P}[\dot{m}_{\theta^*(\mathcal{P}),t}^{\otimes 2} | \mathcal{H}_{t-1}, X_t, A_t = a] = \mathbb{E}_\mathcal{P}[\dot{m}_{\theta^*(\mathcal{P}),t}^{\otimes 2} | X_t, A_t = a]$ and by law of iterated expectations.
- Equality (e) holds because by Condition [1], the distribution of $X_t$ does not depend on $\mathcal{H}_{t-1}$, so $\mathbb{E}_\mathcal{P} \left[ \mathbb{E}_{\mathcal{P}, \pi_t^{\text{sta}}} \left[ \dot{m}_{\theta^*(\mathcal{P}),t}^{\otimes 2} | X_t \right] \middle| \mathcal{H}_{t-1} \right] = \mathbb{E}_\mathcal{P} \left[ \mathbb{E}_{\mathcal{P}, \pi_t^{\text{sta}}} \left[ \dot{m}_{\theta^*(\mathcal{P}),t}^{\otimes 2} | X_t \right] \right] = \mathbb{E}_{\mathcal{P}, \pi_t^{\text{sta}}} \left[ \dot{m}_{\theta^*(\mathcal{P}),t}^{\otimes 2} \right]$; the last equality holds by law of iterated expectations.
- Equality (f) holds by definition.

## 2. Conditional Lindeberg

$$\frac{1}{T} \sum_{t=1}^T \mathbb{E}_{\mathcal{P}, \pi} \left[ \left( \mathbf{c}^\top W_t \Sigma_T(\mathcal{P})^{-1/2} \dot{m}_{\theta^*(\mathcal{P}),t} \right)^2 \mathbb{1}_{\left| \mathbf{c}^\top W_t \Sigma_T(\mathcal{P})^{-1/2} \dot{m}_{\theta^*(\mathcal{P}),t} \right| > \delta \sqrt{T}} \middle| \mathcal{H}_{t-1} \right]$$

$$= \frac{1}{T} \sum_{t=1}^T \mathbb{E}_{\mathcal{P}, \pi} \left[ W_t^2 \mathbf{c}^\top \Sigma_T(\mathcal{P})^{-1/2} \dot{m}_{\theta^*(\mathcal{P}),t}^{\otimes 2} \Sigma_T(\mathcal{P})^{-1/2} \mathbf{c} \mathbb{1}_{\left| \mathbf{c}^\top W_t \Sigma_T(\mathcal{P})^{-1/2} \dot{m}_{\theta^*(\mathcal{P}),t} \right| > \delta \sqrt{T}} \middle| \mathcal{H}_{t-1} \right]$$

$$\underset{(a)}{\leq} \frac{1}{T^2 \delta^2} \sum_{t=1}^T \mathbb{E}_{\mathcal{P}, \pi} \left[ W_t^4 \left( \mathbf{c}^\top \Sigma_T(\mathcal{P})^{-1/2} \dot{m}_{\theta^*(\mathcal{P}),t}^{\otimes 2} \Sigma_T(\mathcal{P})^{-1/2} \mathbf{c} \right)^2 \middle| \mathcal{H}_{t-1} \right]$$

$$\underset{(b)}{\leq} \frac{\rho_{\max}}{T^2 \delta^2} \sum_{t=1}^T \mathbb{E}_{\mathcal{P}, \pi} \left[ W_t^2 \left( \mathbf{c}^\top \Sigma_T(\mathcal{P})^{-1/2} \dot{m}_{\theta^*(\mathcal{P}),t}^{\otimes 2} \Sigma_T(\mathcal{P})^{-1/2} \mathbf{c} \right)^2 \middle| \mathcal{H}_{t-1} \right]$$

$$\underset{(c)}{=} \frac{\rho_{\max}}{T^2 \delta^2} \sum_{t=1}^T \mathbb{E}_\mathcal{P} \left[ \int_{a \in \mathcal{A}} \pi_t(a, X_t, \mathcal{H}_{t-1}) \mathbb{E}_\mathcal{P} \left[ W_t^2 \left( \mathbf{c}^\top \Sigma_T(\mathcal{P})^{-1/2} \dot{m}_{\theta^*(\mathcal{P}),t}^{\otimes 2} \Sigma_T(\mathcal{P})^{-1/2} \mathbf{c} \right)^2 \middle| \mathcal{H}_{t-1}, X_t, A_t = a \right] da \middle| \mathcal{H}_{t-1} \right]$$

$$\underset{(d)}{=} \frac{\rho_{\max}}{T^2 \delta^2} \sum_{t=1}^T \mathbb{E}_\mathcal{P} \left[ \int_{a \in \mathcal{A}} \pi_t^{\text{sta}}(a, X_t) \mathbb{E}_\mathcal{P} \left[ \left( \mathbf{c}^\top \Sigma_T(\mathcal{P})^{-1/2} \dot{m}_{\theta^*(\mathcal{P}),t}^{\otimes 2} \Sigma_T(\mathcal{P})^{-1/2} \mathbf{c} \right)^2 \middle| \mathcal{H}_{t-1}, X_t, A_t = a \right] da \middle| \mathcal{H}_{t-1} \right]$$

$$\underset{(e)}{=} \frac{\rho_{\max}}{T^2 \delta^2} \sum_{t=1}^T \mathbb{E}_\mathcal{P} \left[ \mathbb{E}_\mathcal{P} \left[ \left( \mathbf{c}^\top \Sigma_T(\mathcal{P})^{-1/2} \dot{m}_{\theta^*(\mathcal{P}),t}^{\otimes 2} \Sigma_T(\mathcal{P})^{-1/2} \mathbf{c} \right)^2 \middle| X_t \right] \middle| \mathcal{H}_{t-1} \right]$$

$$\underset{(f)}{=} \frac{\rho_{\max}}{T^2 \delta^2} \sum_{t=1}^T \mathbb{E}_{\mathcal{P}, \pi_t^{\text{sta}}} \left[ \left( \mathbf{c}^\top \Sigma_T(\mathcal{P})^{-1/2} \dot{m}_{\theta^*(\mathcal{P}),t}^{\otimes 2} \Sigma_T(\mathcal{P})^{-1/2} \mathbf{c} \right)^2 \right] \underset{(g)}{\to} 0$$

- Above, inequality (a) holds because $\mathbb{1}_{|W_t \mathbf{c}^\top \Sigma_T(\mathcal{P})^{-1/2} \dot{m}_{\theta^*(\mathcal{P}),t}| > \sqrt{T}\delta} = 1$ if and only if $W_t^2 \frac{1}{T\delta^2} \mathbf{c}^\top \Sigma_T(\mathcal{P})^{-1/2} \dot{m}_{\theta^*(\mathcal{P}),t}^{\otimes 2} \Sigma_T(\mathcal{P})^{-1/2} \mathbf{c} > 1$.

- Inequality (b) holds because by Condition 9, $W_t^2 \leq \rho_{\max}$ with probability 1.

- Equality (c) holds by the law of iterated expectations.

- Equality (d) holds since $W_t = \sqrt{\frac{\pi_t^{\mathrm{sta}}(A_t, X_t)}{\pi_t(A_t, X_t, \mathcal{H}_{t-1})}} \in \sigma(\mathcal{H}_{t-1}, X_t, A_t)$.

- Equality (e) holds because by Condition 1,
$$\mathbb{E}_{\mathcal{P}}\left[ (\mathbf{c}^\top \Sigma_T(\mathcal{P})^{-1/2} \dot{m}_{\theta^*(\mathcal{P}),t}^{\otimes 2} \Sigma_T(\mathcal{P})^{-1/2} \mathbf{c})^2 \big| \mathcal{H}_{t-1}, X_t, A_t = a \right] = \mathbb{E}_{\mathcal{P}}\left[ (\mathbf{c}^\top \Sigma_T(\mathcal{P})^{-1/2} \dot{m}_{\theta^*(\mathcal{P}),t}^{\otimes 2} \Sigma_T(\mathcal{P})^{-1/2} \mathbf{c})^2 \big| X_t \right]$$ and by law of iterated expectations.

- Equality (f) holds since the distribution of $X_t$ does not depend on $\mathcal{H}_{t-1}$ by Condition 1 and by law of iterated expectations.

- Regarding limit (g), it is sufficient to show that $\frac{1}{T} \sum_{t=1}^T \mathbb{E}_{\mathcal{P}, \pi_t^{\mathrm{sta}}}\left[ \left( \mathbf{c}^\top \Sigma_T(\mathcal{P})^{-1/2} \dot{m}_{\theta^*(\mathcal{P}),t}^{\otimes 2} \Sigma_T(\mathcal{P})^{-1/2} \mathbf{c} \right)^2 \right]$ is uniformly bounded over $\mathcal{P} \in \mathbf{P}$ for all sufficiently large $T$. By Condition 5, the minimum eigenvalue of $\Sigma_T(P)$ is bounded above zero uniformly over $\mathcal{P} \in \mathbf{P}$ for all sufficiently large $T$; this bounds the maximum eigenvalue of $\Sigma_T(P)^{-1}$. Also by Condition 5 the fourth moment of $\dot{m}_{\theta^*(\mathcal{P}),t}$ with respect to $\mathcal{P}$ and policy $\pi_t^{\mathrm{sta}}$ is uniformly bounded over $\mathcal{P} \in \mathbf{P}$ and $t \geq 1$. With these two properties we have that $\frac{1}{T} \sum_{t=1}^T \mathbb{E}_{\mathcal{P}, \pi_t^{\mathrm{sta}}}\left[ \left( \mathbf{c}^\top \Sigma_T(\mathcal{P})^{-1/2} \dot{m}_{\theta^*(\mathcal{P}),t}^{\otimes 2} \Sigma_T(\mathcal{P})^{-1/2} \mathbf{c} \right)^2 \right]$ is uniformly bounded over $\mathcal{P} \in \mathbf{P}$ for all sufficiently large $T$.

### B.3.3  Showing that $\sup_{\theta \in \Theta : \|\theta - \theta^*(\mathcal{P})\| \leq \epsilon_{\ddot{m}}} \left\| \dddot{M}_T(\theta) \right\|_1$ is bounded in probability

Recall that for any $B \in \mathbb{R}^{d \times d \times d}$, we denote $\|B\|_1 = \sum_{i=1}^d \sum_{j=1}^d \sum_{k=1}^d |B_{i,j,k}|$. We abbreviate $\dddot{m}_\theta(Y_t, X_t, A_t)$ with $\dddot{m}_{\theta,t}$.

By triangle inequality, $\left\| \dddot{M}_T(\theta) \right\|_1 = \left\| \frac{1}{T} \sum_{t=1}^T W_t \dddot{m}_{\theta,t} \right\|_1 \leq \frac{1}{T} \sum_{t=1}^T W_t \left\| \dddot{m}_{\theta,t} \right\|_1$. Thus we have that

$$\sup_{\theta \in \Theta : \|\theta - \theta^*(\mathcal{P})\| \leq \epsilon_{\ddot{m}}} \left\| \dddot{M}_T(\theta) \right\|_1 \leq \sup_{\theta \in \Theta : \|\theta - \theta^*(\mathcal{P})\| \leq \epsilon_{\ddot{m}}} \frac{1}{T} \sum_{t=1}^T W_t \left\| \dddot{m}_{\theta,t} \right\|_1.$$

By Condition 6 (ii), there exists a function $\dddot{m}$ (note it is not indexed by $\theta$) such that for all $\mathcal{P} \in \mathbf{P}$, we have that $\sup_{\theta \in \Theta : \|\theta - \theta^*(\mathcal{P})\| \leq \epsilon_{\ddot{m}}} \left\| \dddot{m}_{\theta,t} \right\|_1 \leq \left\| \dddot{m}(Y_t, X_t, A_t) \right\|_1$.

$$\leq \frac{1}{T} \sum_{t=1}^T W_t \left\| \dddot{m}(Y_t, X_t, A_t) \right\|_1.$$

Adding and subtracting $\frac{1}{T} \sum_{t=1}^T \mathbb{E}_{\mathcal{P}, \pi}\left[ W_t \left\| \dddot{m}(Y_t, X_t, A_t) \right\|_1 | \mathcal{H}_{t-1} \right]$,

$$= \frac{1}{T} \sum_{t=1}^T W_t \left\| \dddot{m}(Y_t, X_t, A_t) \right\|_1 - \mathbb{E}_{\mathcal{P}, \pi}\left[ W_t \left\| \dddot{m}(Y_t, X_t, A_t) \right\|_1 | \mathcal{H}_{t-1} \right] + \mathbb{E}_{\mathcal{P}, \pi}\left[ W_t \left\| \dddot{m}(Y_t, X_t, A_t) \right\|_1 | \mathcal{H}_{t-1} \right].$$

By second moment bounds on $\left\| \dddot{m}(Y_t, X_t, A_t) \right\|_1$ from Condition 6 (i), by Lemma 1, we have that $\frac{1}{T} \sum_{t=1}^T W_t \left\| \dddot{m}(Y_t, X_t, A_t) \right\|_1 - \mathbb{E}_{\mathcal{P}, \pi}\left[ W_t \left\| \dddot{m}(Y_t, X_t, A_t) \right\|_1 | \mathcal{H}_{t-1} \right] = o_{\mathcal{P} \in \mathbf{P}}(1)$.

$$= o_{\mathcal{P} \in \mathbf{P}}(1) + \frac{1}{T} \sum_{t=1}^T \mathbb{E}_{\mathcal{P}, \pi}\left[ W_t \left\| \dddot{m}(Y_t, X_t, A_t) \right\|_1 | \mathcal{H}_{t-1} \right]$$

Since by Condition 9, $\frac{W_t}{\sqrt{\rho_{\min}}} \geq 1$ with probability 1,

$$\leq o_{\mathcal{P} \in \mathbf{P}}(1) + \frac{1}{T\sqrt{\rho_{\min}}} \sum_{t=1}^T \mathbb{E}_{\mathcal{P}, \pi}\left[ W_t^2 \left\| \dddot{m}(Y_t, X_t, A_t) \right\|_1 | \mathcal{H}_{t-1} \right]$$

Since $W_t^2 = \frac{\pi_t^{\text{sta}}(A_t, X_t)}{\pi_t(A_t, X_t, \mathcal{H}_{t-1})}$ and by Condition 1,

$$= o_{\mathcal{P} \in \mathbf{P}}(1) + \frac{1}{T\sqrt{\rho_{\min}}} \sum_{t=1}^{T} \mathbb{E}_{\mathcal{P}, \pi_t^{\text{sta}}} \left[ \|\dddot{m}(Y_t, X_t, A_t)\|_1 \right] = O_{\mathcal{P} \in \mathbf{P}}(1).$$

Note that by Jensen's inequality, $\mathbb{E}_{\mathcal{P}, \pi_t^{\text{sta}}} \left[ \|\dddot{m}(Y_t, X_t, A_t)\|_1 \right] \leq \sqrt{\mathbb{E}_{\mathcal{P}, \pi_t^{\text{sta}}} \left[ \|\dddot{m}(Y_t, X_t, A_t)\|_1^2 \right]}$. By Condition 6 (i), $\sup_{\mathcal{P} \in \mathbf{P}, t \geq 1} \mathbb{E}_{\mathcal{P}, \pi_t^{\text{sta}}} \left[ \|\dddot{m}(Y_t, X_t, A_t)\|_1^2 \right]$ is bounded, which implies the final limit above.

### B.3.4  Lower bounding $-\ddot{M}_T(\theta^*(\mathcal{P}))$

We now show that $-\ddot{M}_T(\theta^*(\mathcal{P})) \succeq H + o_{\mathcal{P} \in \mathbf{P}}(1)$, for positive definite matrix $H$ introduced in Condition 7 (ii).

By Condition 5 and Lemma 1, $\frac{1}{T} \sum_{t=1}^{T} W_t \ddot{m}_{\theta^*(\mathcal{P}), t} - \mathbb{E}_{\mathcal{P}, \pi} \left[ W_t \ddot{m}_{\theta^*(\mathcal{P}), t} | \mathcal{H}_{t-1} \right] = o_{\mathcal{P} \in \mathbf{P}}(1)$, so

$$-\ddot{M}_T(\theta^*(\mathcal{P})) = -\frac{1}{T} \sum_{t=1}^{T} W_t \ddot{m}_{\theta^*(\mathcal{P}), t} = o_{\mathcal{P} \in \mathbf{P}}(1) - \frac{1}{T} \sum_{t=1}^{T} \mathbb{E}_{\mathcal{P}, \pi} \left[ W_t \ddot{m}_{\theta^*(\mathcal{P}), t} | \mathcal{H}_{t-1} \right]$$

By law of iterated expectations,

$$= o_{\mathcal{P} \in \mathbf{P}}(1) - \frac{1}{T} \sum_{t=1}^{T} \mathbb{E}_{\mathcal{P}, \pi} \left[ W_t \mathbb{E}_{\mathcal{P}} \left[ \ddot{m}_{\theta^*(\mathcal{P}), t} | \mathcal{H}_{t-1}, X_t, A_t \right] | \mathcal{H}_{t-1} \right]$$

By Condition 1,

$$= o_{\mathcal{P} \in \mathbf{P}}(1) - \frac{1}{T} \sum_{t=1}^{T} \mathbb{E}_{\mathcal{P}, \pi} \left[ W_t \mathbb{E}_{\mathcal{P}} \left[ \ddot{m}_{\theta^*(\mathcal{P}), t} | X_t, A_t \right] | \mathcal{H}_{t-1} \right]$$

By Condition 7, we have that $\mathbb{E}_{\mathcal{P}} \left[ \ddot{m}_{\theta^*(\mathcal{P}), t} | X_t, A_t \right] \preceq 0$; recall that $\theta^*(\mathcal{P})$ is a maximizing value of $\mathbb{E}_{\mathcal{P}, \pi} \left[ m_{\theta, t} | X_t, A_t \right]$. Also since $\frac{W_t}{\sqrt{\rho_{\max}}} \leq 1$ with probability 1 by Condition 9,

$$\succeq o_{\mathcal{P} \in \mathbf{P}}(1) - \frac{1}{T\sqrt{\rho_{\max}}} \sum_{t=1}^{T} \mathbb{E}_{\mathcal{P}, \pi} \left[ W_t^2 \mathbb{E}_{\mathcal{P}, \pi} \left[ \ddot{m}_{\theta^*(\mathcal{P}), t} | X_t, A_t \right] | \mathcal{H}_{t-1} \right]$$

Since $W_t^2 = \frac{\pi_t^{\text{sta}}(A_t, X_t)}{\pi_t(A_t, X_t, \mathcal{H}_{t-1})}$,

$$= o_{\mathcal{P} \in \mathbf{P}}(1) - \frac{1}{T\sqrt{\rho_{\max}}} \sum_{t=1}^{T} \mathbb{E}_{\mathcal{P}, \pi_t^{\text{sta}}} \left[ \ddot{m}_{\theta^*(\mathcal{P}), t} | \mathcal{H}_{t-1} \right]$$

Note that for any $t \geq 1$, $\mathbb{E}_{\mathcal{P}, \pi_t^{\text{sta}}} \left[ \ddot{m}_{\theta^*(\mathcal{P}), t} | \mathcal{H}_{t-1} \right] = \mathbb{E}_{\mathcal{P}, \pi_t^{\text{sta}}} \left[ \ddot{m}_{\theta^*(\mathcal{P}), t} \right]$ because $\{\pi_t^{\text{sta}}\}_{t \geq 1}$ are pre-specified. Recall that by Condition 7 for all sufficiently large $T$, $-\frac{1}{T} \sum_{t=1}^{T} \mathbb{E}_{\mathcal{P}, \pi_t^{\text{sta}}} \left[ \ddot{m}_{\theta^*(\mathcal{P}), t} \right] \succeq H$ for all $\mathcal{P} \in \mathbf{P}$. Thus our final result is that

$$- \ddot{M}_T(\theta^*(\mathcal{P})) \succeq H + o_{\mathcal{P} \in \mathbf{P}}(1). \tag{23}$$

### B.4  Lemmas and Other Helpful Results

**Theorem 2** (Uniform Martingale Central Limit Theorem). *Let $\{Z_T(\mathcal{P})\}_{T \geq 1}$ be a sequence of random variables whose distributions are defined by some $\mathcal{P} \in \mathbf{P}$ and some nuisance component $\eta$. Moreover, let $\{Z_T(\mathcal{P})\}_{T \geq 1}$ be a martingale difference sequence with respect to $\mathcal{F}_t$, meaning $\mathbb{E}_{\mathcal{P}, \eta}[Z_t(\mathcal{P}) | \mathcal{F}_{t-1}] = 0$ for all $t \geq 1$ and $\mathcal{P} \in \mathbf{P}$.*

*(a) $\frac{1}{T} \sum_{t=1}^{T} \mathbb{E}_{\mathcal{P}, \eta}[Z_t(\mathcal{P})^2 | \mathcal{F}_{t-1}] \xrightarrow{P} \sigma^2$ uniformly over $\mathcal{P} \in \mathbf{P}$, where $\sigma^2$ is a constant $0 < \sigma^2 < \infty$.*

*(b)* For any $\epsilon > 0$, $\frac{1}{T}\sum_{t=1}^{T}\mathbb{E}_{\mathcal{P},\eta}[Z_t(\mathcal{P})^2 \mathbb{1}_{|Z_t(\mathcal{P})|>\epsilon}|\mathcal{F}_{t-1}] \overset{P}{\to} 0$ *uniformly over* $\mathcal{P} \in \boldsymbol{P}$.

*Under the above conditions,*

$$\frac{1}{\sqrt{T}}\sum_{t=1}^{T}Z_t(\mathcal{P}) \overset{D}{\to} \mathcal{N}(0, \sigma^2) \text{ uniformly over } \mathcal{P} \in \boldsymbol{P}.$$

**Proof:** By by Kasy [2019, Lemma 1], it is sufficient to show that for any sequence $\{\mathcal{P}_T\}_{T=1}^{\infty}$ with $\mathcal{P}_T \in \boldsymbol{P}$ for all $T \geq 1$, $\frac{1}{\sqrt{T}}\sum_{t=1}^{T}Z_t(\mathcal{P}_T) \overset{D}{\to} \mathcal{N}(0, \sigma^2)$. In this setting, since $\mathcal{P}_T$ depends on $T$, we consider triangular array asymptotics and additionally index by $T$, e.g., $\mathcal{F}_{T,t}$.

Note that $\frac{1}{T}\sum_{t=1}^{T}\mathbb{E}_{\mathcal{P}_T,\eta}[Z_t(\mathcal{P}_T)^2|\mathcal{F}_{T,t-1}] \overset{P}{\to} \sigma^2$, by Kasy [2019, Lemma 1] and condition (a) above.

Also, for any $\epsilon > 0$, $\frac{1}{T}\sum_{t=1}^{T}\mathbb{E}_{\mathcal{P}_T,\eta}\left[Z_t(\mathcal{P}_T)^2 \mathbb{1}_{|Z_t(\mathcal{P}_T)|>\epsilon}|\mathcal{F}_{T,t-1}\right] \overset{P}{\to} 0$, by Kasy [2019, Lemma 1] and condition (b) above.

Thus by the martingale central limit theorem of Dvoretzky [1972], we have that for the sequence $\{\mathcal{P}_T\}_{T=1}^{\infty}$,

$$\frac{1}{\sqrt{T}}\sum_{t=1}^{T}Z_t(\mathcal{P}_T) \overset{D}{\to} \mathcal{N}(0, 1).$$

Since the sequence $\{\mathcal{P}_T\}_{T=1}^{\infty}$ were chosen arbitrarily from $\boldsymbol{P}$, the desired result is implied again by Kasy [2019, Lemma 1].

**Lemma 1.** *Let $f(Y_t, X_t, A_t) \in \mathbb{R}^{d_f}$ be a function such that* $\sup_{\mathcal{P} \in \boldsymbol{P}, t \geq 1}\mathbb{E}_{\mathcal{P},\pi_t^{\mathrm{sta}}}\left[\|f(Y_t, X_t, A_t)\|^2\right] < m$ *for some $m < \infty$. Under Conditions 1 and 9,*

$$\frac{1}{\sqrt{T}}\sum_{t=1}^{T}\left\{W_t f(Y_t, X_t, A_t) - \mathbb{E}_{\mathcal{P},\pi}[W_t f(Y_t, X_t, A_t)|\mathcal{H}_{t-1}]\right\} = O_{\mathcal{P} \in \boldsymbol{P}}(1). \tag{24}$$

*Note that the above equation implies that*

$$\frac{1}{T}\sum_{t=1}^{T}\left\{W_t f(Y_t, X_t, A_t) - \mathbb{E}_{\mathcal{P},\pi}[W_t f(Y_t, X_t, A_t)|\mathcal{H}_{t-1}]\right\} = o_{\mathcal{P} \in \boldsymbol{P}}(1).$$

Lemma 1 is a type of martingale weak law of large number result and the proof is similar to the weak law of large numbers proofs for i.i.d. random variables.

**Proof:** We denote the $k^{\mathrm{th}} \in [1:d_f]$ dimension of vector $f(Y_t, X_t, A_t)$ as $f^k(Y_t, X_t, A_t)$. It is sufficient to show the result for any dimension of vector $f(Y_t, X_t, A_t)$. For notational convenience, let $f_t := f^k(Y_t, X_t, A_t)$. Let $\epsilon > 0$.

$$\sup_{\mathcal{P} \in \boldsymbol{P}}\mathbb{P}_{\mathcal{P},\pi}\left(\left|\frac{1}{\sqrt{T}}\sum_{t=1}^{T}\left\{W_t f_t - \mathbb{E}_{\mathcal{P},\pi}[W_t f_t|\mathcal{H}_{t-1}]\right\}\right| > \epsilon\right)$$

$$\underset{(a)}{\leq} \frac{1}{T\epsilon^2}\sup_{\mathcal{P} \in \boldsymbol{P}}\mathbb{E}_{\mathcal{P},\pi}\left[\left(\sum_{t=1}^{T}\left\{W_t f_t - \mathbb{E}_{\mathcal{P},\pi}[W_t f_t|\mathcal{H}_{t-1}]\right\}\right)^2\right]$$

$$\underset{(b)}{=} \frac{1}{T\epsilon^2}\sup_{\mathcal{P} \in \boldsymbol{P}}\sum_{t=1}^{T}\mathbb{E}_{\mathcal{P},\pi}\left[\left\{W_t f_t - \mathbb{E}_{\mathcal{P},\pi}[W_t f_t|\mathcal{H}_{t-1}]\right\}^2\right]$$

$$\underset{(c)}{\leq} \frac{1}{T\epsilon^2}\sup_{\mathcal{P} \in \boldsymbol{P}}\sum_{t=1}^{T}\mathbb{E}_{\mathcal{P},\pi}\left[W_t^2 f_t^2\right]$$

$$\underset{(d)}{=} \frac{1}{T\epsilon^2}\sup_{\mathcal{P} \in \boldsymbol{P}}\sum_{t=1}^{T}\mathbb{E}_{\mathcal{P}}\left[\int_{a \in \mathcal{A}}W_t^2 \pi_t(a, X_t, \mathcal{H}_{t-1})\mathbb{E}_{\mathcal{P}}[f_t^2|\mathcal{H}_{t-1}, X_t, A_t = a]da\right]$$

$$\underset{(e)}{=} \frac{1}{T\epsilon^2} \sup_{\mathcal{P}\in\mathbf{P}} \sum_{t=1}^{T} \mathbb{E}_{\mathcal{P}} \left[ \int_{a\in\mathcal{A}} \pi_t^{\text{sta}}(a, X_t) \mathbb{E}_{\mathcal{P}}[f_t^2 | \mathcal{H}_{t-1}, X_t, A_t = a] da \right]$$

$$\underset{(f)}{=} \frac{1}{T\epsilon^2} \sup_{\mathcal{P}\in\mathbf{P}} \sum_{t=1}^{T} \mathbb{E}_{\mathcal{P}, \pi_t^{\text{sta}}} \left[ f_t^2 \right] \underset{(g)}{\leq} \frac{4m}{\epsilon^2}$$

- Above (a) holds by Chebyshev's inequality.
- (b) holds because the above terms form a martingale difference sequence with respect to $\mathcal{H}_{t-1}$, i.e., $\mathbb{E}_{\mathcal{P}, \pi} \left[ W_t f_t - \mathbb{E}_{\mathcal{P}, \pi}[W_t f_t | \mathcal{H}_{t-1}] \big| \mathcal{H}_{t-1} \right] = 0$; this implies that cross terms disappear, i.e., for $t > s$,

$$\mathbb{E}_{\mathcal{P}, \pi} \left[ \left( W_t f_t - \mathbb{E}_{\mathcal{P}, \pi}[W_t f_t | \mathcal{H}_{t-1}] \right) \left( W_s f_s - \mathbb{E}_{\mathcal{P}, \pi}[W_s f_s | \mathcal{H}_{s-1}] \right) \right]$$

$$= \mathbb{E}_{\mathcal{P}, \pi} \left[ \mathbb{E}_{\mathcal{P}, \pi} \left[ \left( W_t f_t - \mathbb{E}_{\mathcal{P}, \pi}[W_t f_t | \mathcal{H}_{t-1}] \right) \left( W_s f_s - \mathbb{E}_{\mathcal{P}, \pi}[W_s f_s | \mathcal{H}_{s-1}] \right) \Big| \mathcal{H}_{t-1} \right] \right]$$

Since $s > t$,

$$= \mathbb{E}_{\mathcal{P}, \pi} \left[ \left( W_s f_s - \mathbb{E}_{\mathcal{P}, \pi}[W_s f_s | \mathcal{H}_{s-1}] \right) \mathbb{E}_{\mathcal{P}, \pi} \left[ W_t f_t - \mathbb{E}_{\mathcal{P}, \pi}[W_t f_t | \mathcal{H}_{t-1}] \Big| \mathcal{H}_{t-1} \right] \right] = 0.$$

- (c) holds because $\mathbb{E}_{\mathcal{P}, \pi} \left[ \{ W_t f_t - \mathbb{E}_{\mathcal{P}, \pi}[W_t f_t | \mathcal{H}_{t-1}] \}^2 \right] = \mathbb{E}_{\mathcal{P}, \pi} \left[ W_t^2 f_t^2 \right] - \mathbb{E}_{\mathcal{P}, \pi} \left[ \mathbb{E}_{\mathcal{P}, \pi}[W_t f_t | \mathcal{H}_{t-1}]^2 \right] \leq \mathbb{E}_{\mathcal{P}, \pi} \left[ W_t^2 f_t^2 \right]$.
- (d) holds by law of iterated expectations.
- (e) holds because $W_t = \sqrt{\frac{\pi_t^{\text{sta}}(A_t, X_t)}{\pi_t(A_t, X_t, \mathcal{H}_{t-1})}}$.
- (f) holds since by Condition 1, $\mathbb{E}_{\mathcal{P}}[f_t^2 | \mathcal{H}_{t-1}, X_t, A_t] = \mathbb{E}_{\mathcal{P}}[f_t^2 | X_t, A_t]$ and by law of iterated expectations $\mathbb{E}_{\mathcal{P}, \pi_t^{\text{sta}}} \left[ f_t^2 \right] = \mathbb{E}_{\mathcal{P}} \left[ \int_{a\in\mathcal{A}} \pi_t^{\text{sta}}(a, X_t) \mathbb{E}_{\mathcal{P}}[f_t^2 | X_t, A_t = a] da \right]$.
- (g) holds since $\sup_{\mathcal{P}\in\mathbf{P}, t\geq 1} \mathbb{E}_{\mathcal{P}, \pi_t^{\text{sta}}} \left[ f_t^2 \right] < m < \infty$.

**Lemma 2.** *Let $m_{\theta, t} := m_\theta(Y_t, X_t, A_t)$. Under Conditions 1, 3, 4, 5, 7 and 9,*

$$\sup_{\theta\in\Theta} \left\{ \frac{1}{T} \sum_{t=1}^{T} W_t m_{\theta, t} - \mathbb{E}_{\mathcal{P}, \pi}[W_t m_{\theta, t} | \mathcal{H}_{t-1}] \right\} = O_{\mathcal{P}\in\mathbf{P}}(1). \tag{25}$$

Lemma 1 is a type of martingale functionally uniform law of large number result and the proof is similar to the functionally uniform law of large numbers proofs for i.i.d. random variables Van Der Vaart and Wellner [1996, Theorem 2.4.1].

**Proof:**

**Finite Bracketing Number:** Let $\delta > 0$. We construct a set $B_\delta$ which is made up of pairs of functions $(l, u)$. We show that we can find $B_\delta$ that satisfies the following:

(a) For any $\theta \in \Theta$, we can find $(l, u) \in B_\delta$ such that
 (i) $l(y, x, a) \leq m_\theta(y, x, a) \leq u(y, x, a)$ for all $(x, y)$ in the joint support of $\{\mathcal{P} \in \mathbf{P}\}$ and all $a \in \mathcal{A}$.
 (ii) $\sup_{\mathcal{P}\in\mathbf{P}, t\geq 1} \mathbb{E}_{\mathcal{P}, \pi_t^{\text{sta}}} \left[ |u(Y_t, X_t, A_t) - l(Y_t, X_t, A_t)| \right] \leq \delta$.

(b) The number of pairs in this set is finite, i.e., $|B_\delta| < \infty$.

(c) For any $(l, u) \in B_\delta$, for some $m < \infty$ which does no depend on $\delta$, $\sup_{\mathcal{P}\in\mathbf{P}, t\geq 1} \mathbb{E}_{\mathcal{P}, \pi_t^{\text{sta}}} \left[ u(Y_t, X_t, A_t)^2 \right] \leq m$ and $\sup_{\mathcal{P}\in\mathbf{P}, t\geq 1} \mathbb{E}_{\mathcal{P}, \pi_t^{\text{sta}}} \left[ l(Y_t, X_t, A_t)^2 \right] \leq m$.

Showing that we can find $B_\delta$ that satisfy (a), means that $|B_\delta|$ is an upper bound on the bracketing number of $\{m_\theta : \theta \in \Theta\}$. For more information on bracketing functions, see Van Der Vaart and Wellner [1996] and Van der Vaart [2000].

To construct $B_\delta$, we follow a similar argument to Example 19.7 of Van der Vaart [2000] (page 271). Make a grid over $\Theta$ with meshwidth $\lambda/2 > 0$ and let the points in this grid be the set $G_{\lambda/2} \subseteq \Theta$; we will specify $\lambda$ later. Note that by construction, for any $\theta \in \Theta$ we can find a $\theta \in G_{\lambda/2}$ such that $\|\theta' - \theta\| \leq \lambda$.

By our Lipschitz Condition 4, we have that for any $\theta, \theta' \in \Theta$, $|m_\theta(Y_t, X_t, A_t) - m_{\theta'}(Y_t, X_t, A_t)| \leq g(Y_t, X_t, A_t)\|\theta - \theta'\|$ for function $g$ such that for some $m_g < \infty$,

$$\sup_{\mathcal{P} \in \mathbf{P}, t \geq 1} \mathbb{E}_{\mathcal{P}, \pi_t^{\mathrm{sta}}}[g(Y_t, X_t, A_t)^2] \leq m_g. \tag{26}$$

We now show that we can choose $B_\delta = \big\{(m_\theta - g(Y_t, X_t, A_t), m_\theta + g(Y_t, X_t, A_t)) : \theta \in G_{\lambda/2}\big\}$. Note that by compactness of $\Theta$, Condition 3, the number of points in $G_{\lambda/2}$ is finite, so (b) above holds.

To show that (a) holds for our choice of $B_\delta$, recall that for any $\theta \in \Theta$ we can find a $\theta' \in G_{\lambda/2}$ such that $\|\theta' - \theta\| \leq \lambda$. Also, by the Lipschitz Condition 4, $|m_\theta(Y_t, X_t, A_t) - m_{\theta'}(Y_t, X_t, A_t)| \leq g(Y_t, X_t, A_t)\|\theta - \theta'\| \leq g(Y_t, X_t, A_t)\lambda$. Thus we have that

$$m_{\theta'}(Y_t, X_t, A_t) - g(Y_t, X_t, A_t)\lambda \leq m_\theta(Y_t, X_t, A_t) \leq m_{\theta'}(Y_t, X_t, A_t) + g(Y_t, X_t, A_t)\lambda.$$

Note that

$$\sup_{\mathcal{P} \in \mathbf{P}, t \geq 1} \mathbb{E}_{\mathcal{P}, \pi_t^{\mathrm{sta}}} \big[ m_{\theta'}(Y_t, X_t, A_t) + g(Y_t, X_t, A_t)\lambda - \{m_{\theta'}(Y_t, X_t, A_t) - g(Y_t, X_t, A_t)\lambda\}\big]$$

$$= 2\lambda \sup_{\mathcal{P} \in \mathbf{P}, t \geq 1} \mathbb{E}_{\mathcal{P}, \pi_t^{\mathrm{sta}}} [g(Y_t, X_t, A_t)] \leq 2\lambda \sqrt{m_g} < \infty.$$

The inequalities above hold by Equation (26) and since $\mathbb{E}_{\mathcal{P}, \pi_t^{\mathrm{sta}}}[g(Y_t, X_t, A_t)] \leq \sqrt{\mathbb{E}_{\mathcal{P}, \pi_t^{\mathrm{sta}}}[g(Y_t, X_t, A_t)^2]}$ by Jensen's inequality. (a) above holds for our choice of $B_\delta$ by letting meshwidth $\lambda = \delta/(2\sqrt{m_g})$.

We now show that (c) above holds. Note that

$$\sup_{\mathcal{P} \in \mathbf{P}, t \geq 1} \mathbb{E}_{\mathcal{P}, \pi_t^{\mathrm{sta}}} \Big[ \{m_\theta(Y_t, X_t, A_t) + g(Y_t, X_t, A_t)\}^2 \Big]$$

$$\leq 3 \sup_{\mathcal{P} \in \mathbf{P}, t \geq 1} \mathbb{E}_{\mathcal{P}, \pi_t^{\mathrm{sta}}} \big[ m_\theta(Y_t, X_t, A_t)^2 \big] + 3 \sup_{\mathcal{P} \in \mathbf{P}, t \geq 1} \mathbb{E}_{\mathcal{P}, \pi_t^{\mathrm{sta}}} \big[ g(Y_t, X_t, A_t)^2 \big]. \tag{27}$$

Note that the above upper bound, Equation (27), also holds for $\sup_{\mathcal{P} \in \mathbf{P}, t \geq 1} \mathbb{E}_{\mathcal{P}, \pi_t^{\mathrm{sta}}} \Big[ \{m_\theta(Y_t, X_t, A_t) - g(Y_t, X_t, A_t)\}^2 \Big]$.

Since, $m_\theta(Y_t, X_t, A_t) = m_\theta(Y_t, X_t, A_t) - m_{\theta^*(\mathcal{P})}(Y_t, X_t, A_t) + m_{\theta^*(\mathcal{P})}(Y_t, X_t, A_t)$,

$$\leq 9 \sup_{\mathcal{P} \in \mathbf{P}, t \geq 1} \mathbb{E}_{\mathcal{P}, \pi_t^{\mathrm{sta}}} \Big[ \{m_\theta(Y_t, X_t, A_t) - m_{\theta^*(\mathcal{P})}(Y_t, X_t, A_t)\}^2 \Big]$$

$$+ 9 \sup_{\mathcal{P} \in \mathbf{P}, t \geq 1} \mathbb{E}_{\mathcal{P}, \pi_t^{\mathrm{sta}}} \big[ m_{\theta^*(\mathcal{P})}(Y_t, X_t, A_t)^2 \big]$$

$$+ 3 \sup_{\mathcal{P} \in \mathbf{P}, t \geq 1} \mathbb{E}_{\mathcal{P}, \pi_t^{\mathrm{sta}}} \big[ g(Y_t, X_t, A_t)^2 \big].$$

Note that $\sup_{\mathcal{P} \in \mathbf{P}, t \geq 1} \mathbb{E}_{\mathcal{P}, \pi_t^{\mathrm{sta}}} \big[ m_{\theta^*(\mathcal{P})}(Y_t, X_t, A_t)^2 \big]$ is bounded by our moment Condition 5 and that $\sup_{\mathcal{P} \in \mathbf{P}, t \geq 1} \mathbb{E}_{\mathcal{P}, \pi_t^{\mathrm{sta}}} \big[ g(Y_t, X_t, A_t)^2 \big]$ is bounded by Equation (26).

By our Lipschitz Condition 4, for any $\theta \in \Theta$, $|m_\theta(Y_t, X_t, A_t) - m_{\theta^*(\mathcal{P})}(Y_t, X_t, A_t)| \leq g(Y_t, X_t, A_t)\|\theta - \theta^*(\mathcal{P})\|$. Thus,

$$\sup_{\mathcal{P} \in \mathbf{P}, t \geq 1} \mathbb{E}_{\mathcal{P}, \pi_t^{\mathrm{sta}}} \Big[ \{m_\theta(Y_t, X_t, A_t) - m_{\theta^*(\mathcal{P})}(Y_t, X_t, A_t)\}^2 \Big]$$

$$\leq \sup_{\mathcal{P} \in \mathbf{P}, t \geq 1} \mathbb{E}_{\mathcal{P}, \pi_t^{\mathrm{sta}}} \big[ g(Y_t, X_t, A_t)^2 \big] \|\theta - \theta^*(\mathcal{P})\|^2.$$

The above is bounded by Equation (26) and by compactness of $\Theta$, Condition 3. Thus (c) above holds for our choice of $B_\delta$.

**Main Argument:** We now show that for any $\epsilon > 0$,

$$\sup_{\mathcal{P} \in \mathbf{P}} \mathbb{P}_{\mathcal{P},\pi} \left( \sup_{\theta \in \Theta} \left\{ \frac{1}{T} \sum_{t=1}^{T} W_t m_{\theta,t} - \mathbb{E}_{\mathcal{P},\pi}[W_t m_{\theta,t} | \mathcal{H}_{t-1}] \right\} > \epsilon \right) \to 0. \tag{28}$$

An analogous argument can be made to show that
$\sup_{\mathcal{P} \in \mathbf{P}} \mathbb{P}_{\mathcal{P},\pi} \left( \sup_{\theta \in \Theta} \left\{ -\frac{1}{T} \sum_{t=1}^{T} W_t m_{\theta,t} - \mathbb{E}_{\mathcal{P},\pi}[W_t m_{\theta,t} | \mathcal{H}_{t-1}] \right\} > \epsilon \right) \to 0.$

Let $\delta > 0$; we will choose $\delta$ later. Let $B_\delta$ be the set of pairs of functions as constructed earlier.

$$\sup_{\theta \in \Theta} \left\{ \frac{1}{T} \sum_{t=1}^{T} W_t m_{\theta,t} - \mathbb{E}_{\mathcal{P},\pi}[W_t m_{\theta,t} | \mathcal{H}_{t-1}] \right\}$$

Note that by (a), we get the following upper bound:

$$\leq \max_{(l,u) \in B_\delta} \left\{ \frac{1}{T} \sum_{t=1}^{T} W_t u(Y_t, X_t, A_t) - \mathbb{E}_{\mathcal{P},\pi}[W_t l(Y_t, X_t, A_t) | \mathcal{H}_{t-1}] \right\}.$$

By adding and subtracting $\mathbb{E}_{\mathcal{P},\pi}\left[ W_t u(Y_t, X_t, A_t) | \mathcal{H}_{t-1} \right]$ and triangle inequality,

$$\leq \max_{(l,u) \in B_\delta} \left\{ \frac{1}{T} \sum_{t=1}^{T} \mathbb{E}_{\mathcal{P},\pi} \left[ W_t \{ u(Y_t, X_t, A_t) - l(Y_t, X_t, A_t) \} | \mathcal{H}_{t-1} \right] \right\}$$

$$+ \max_{(l,u) \in B_\delta} \left\{ \frac{1}{T} \sum_{t=1}^{T} W_t u(Y_t, X_t, A_t) - \mathbb{E}_{\mathcal{P},\pi} \left[ W_t u(Y_t, X_t, A_t) | \mathcal{H}_{t-1} \right] \right\}.$$

Note that by Condition 9, $W_t = \sqrt{\frac{\pi_t^{\text{sta}}(A_t, X_t)}{\pi_t(A_t, X_t, \mathcal{H}_{t-1})}} \leq \sqrt{\rho_{\max}}$ with probability 1, so
$\mathbb{E}_{\mathcal{P},\pi} \left[ W_t \{ u(Y_t, X_t, A_t) - l(Y_t, X_t, A_t) \} | \mathcal{H}_{t-1} \right] \leq \frac{1}{\sqrt{\rho_{\max}}} \mathbb{E}_{\mathcal{P},\pi} \left[ W_t^2 \{ u(Y_t, X_t, A_t) - l(Y_t, X_t, A_t) \} | \mathcal{H}_{t-1} \right]$
$= \frac{1}{\sqrt{\rho_{\max}}} \mathbb{E}_{\mathcal{P},\pi_t^{\text{sta}}} \left[ u(Y_t, X_t, A_t) - l(Y_t, X_t, A_t) \right] \leq \frac{1}{\sqrt{\rho_{\max}}} \delta$; the last equality holds by Condition 1
and the last inequality holds by (a). And since $\max_{i \in [1:n]} \{a_i\} \leq \sum_{i=1}^{n} |a_i|$,

$$\leq \frac{1}{\sqrt{\rho_{\max}}} \delta + \sum_{(l,u) \in B_\delta} \left| \frac{1}{T} \sum_{t=1}^{T} W_t u(Y_t, X_t, A_t) - \mathbb{E}_{\mathcal{P},\pi} \left[ W_t u(Y_t, X_t, A_t) | \mathcal{H}_{t-1} \right] \right|$$

By Lemma 1 and (c), for any $(l,u) \in B_\delta$, $\frac{1}{T} \sum_{t=1}^{T} W_t u(Y_t, X_t, A_t) - \mathbb{E}_{\mathcal{P},\pi} \left[ W_t u(Y_t, X_t, A_t) | \mathcal{H}_{t-1} \right] = o_{\mathcal{P} \in \mathbf{P}}(1)$. Since $|B_\delta| < \infty$ by (b), the convergence holds for all $(l,u) \in B_\delta$ simultaneously, so

$$= \frac{1}{\sqrt{\rho_{\max}}} \delta + o_{\mathcal{P} \in \mathbf{P}}(1).$$

Equation (28) holds by choosing $\delta = \sqrt{\rho_{\max}} \epsilon / 2$.

## B.5 Least-Squares Estimator

We use $\phi(X_t, A_t)$ to denote a feature vector that constructed using context $X_t$ and action $A_t$.

**Condition 10** (Linear Expected Outcome). *For all $\mathcal{P} \in \mathbf{P}$, the following holds w.p. 1,*

$$\mathbb{E}_{\mathcal{P}} [Y_t | X_t, A_t] = \phi(X_t, A_t)^\top \theta^*(\mathcal{P}).$$

**Condition 11** (Moment Conditions for Least Squares). *The fourth moments of $\phi(X_t, A_t) \left( Y_t - \phi(X_t, A_t)^\top \theta^*(\mathcal{P}) \right)$ and $\phi(X_t, A_t)$ with respect to $\mathcal{P}$ and policy $\pi_t^{\text{sta}}$ are respectively bounded uniformly over $\mathcal{P} \in \mathbf{P}$ and $t \geq 1$.*

*Also the minimum eigenvalue of $\Sigma_T(\mathcal{P}) = \frac{1}{T} \sum_{t=1}^{T} \mathbb{E}_{\mathcal{P},\pi_t^{\text{sta}}} \left[ \phi(Y_t, X_t, A_t)^{\otimes 2} \left( Y_t - \phi(Y_t, X_t, A_t)^\top \theta^*(\mathcal{P}) \right)^2 \right]$ and $\frac{1}{T} \sum_{t=1}^{T} \mathbb{E}_{\mathcal{P},\pi_t^{\text{sta}}} \left[ \phi(X_t, A_t)^{\otimes 2} \right]$ respectively are both bounded above constant some constant greater than zero for all $\mathcal{P} \in \mathbf{P}$.*

**Condition 12** (Importance Ratios for Least Squares). *Let $\rho_{\min} > 0$ and $\rho_{\max,T} > 0$ be a non-random sequence such that $\frac{\rho_{\max,T}}{T} \to 0$. $\{\pi_t^{\text{sta}}\}_{t=1}^T$ are pre-specified and do not depend on data $\{Y_t, X_t, A_t\}_{t=1}^T$. For all $\mathcal{P} \in \boldsymbol{P}$, the following holds w.p. 1,*

$$\rho_{\min} \leq \frac{\pi_t^{\text{sta}}(A_t, X_t)}{\pi_t(A_t, X_t, \mathcal{H}_{t-1})} \leq \rho_{\max,T}.$$

Note that Condition 12 allows $\pi_t(A_t, X_t, \mathcal{H}_{t-1})$ to go to zero at some rate for stabilizing policies $\{\pi_t^{\text{sta}}\}_{t \geq 1}$ that are strictly bounded away from 0 and 1.

We now define the AW-LS estimator for $\theta^*(\mathcal{P}) \in \mathbb{R}^d$:

$$\hat{\theta}_T^{\text{AW-LS}} := \underset{\theta \in \mathbb{R}^d}{\operatorname{argmax}} \left\{ -\sum_{t=1}^T W_t \left( Y_t - \phi(X_t, A_t)^\top \theta \right)^2 \right\}. \tag{29}$$

**Theorem 3** (Consistency and Asymptotic Normality of Adaptively-Weighted Least Squares Estimator). *Under Conditions 1, 10, 11, and 12,*

$$\Sigma_T(\mathcal{P})^{-1/2} \left( \frac{1}{\sqrt{T}} \sum_{t=1}^T W_t \phi(X_t, A_t)^{\otimes 2} \right) \left( \hat{\theta}_T^{\text{AW-LS}} - \theta^*(\mathcal{P}) \right) \overset{D}{\to} \mathcal{N}(0, I_d) \text{ uniformly over } \mathcal{P} \in \boldsymbol{P},$$

*where $\Sigma_T(\mathcal{P}) := \frac{1}{T} \sum_{t=1}^T \phi(X_t, A_t)^{\otimes 2} \left( Y_t - \phi(X_t, A_t)^\top \theta^*(\mathcal{P}) \right)^2$.*

**Proof:** By taking the derivative of Equation (29) with respect to the parameters, we have that

$$0 = \sum_{t=1}^T W_t \phi(X_t, A_t) \left( Y_t - \phi(X_t, A_t)^\top \hat{\theta}_T^{\text{AW-LS}} \right).$$

By rearranging terms, we have that

$$-\frac{1}{\sqrt{T}} \sum_{t=1}^T W_t \phi(X_t, A_t) \left( Y_t - \phi(X_t, A_t)^\top \theta^*(\mathcal{P}) \right)$$

$$= \frac{1}{\sqrt{T}} \sum_{t=1}^T W_t \phi(X_t, A_t)^{\otimes 2} \left( \hat{\theta}_T^{\text{AW-LS}} - \theta^*(\mathcal{P}) \right). \tag{30}$$

We first show that the following holds:

$$\Sigma_T(\mathcal{P})^{-1/2} \frac{1}{\sqrt{T}} \sum_{t=1}^T W_t \phi(X_t, A_t) \left( Y_t - \phi(X_t, A_t)^\top \theta^*(\mathcal{P}) \right) \overset{D}{\to} \mathcal{N}(0, I_d) \text{ uniformly over } \mathcal{P} \in \mathbf{P}.$$

$$\tag{31}$$

Equation (31) holds by a similar argument as that used in Section B.3.2, for $\dot{m}_\theta(Y_t, X_t, A_t) = \phi(X_t, A_t) \left( Y_t - \phi(X_t, A_t)^\top \theta^*(\mathcal{P}) \right)$ by showing that the conditions of Theorem 2 hold. It can be checked that all the arguments hold even when we allow $\rho_{\max,T}$ to grow at a rate such that $\frac{\rho_{\max,T}}{T} \to 0$.

By Equations (30) and (31),

$$\Sigma_T(\mathcal{P})^{-1/2} \frac{1}{\sqrt{T}} \sum_{t=1}^T W_t \phi(X_t, A_t)^{\otimes 2} \left( \hat{\theta}_T^{\text{AW-LS}} - \theta^*(\mathcal{P}) \right) \overset{D}{\to} \mathcal{N}(0, I_d) \text{ uniformly over } \mathcal{P} \in \mathbf{P}.$$

$$\tag{32}$$

By Equation (32), to ensure that $\hat{\theta}_T^{\text{AW-LS}} \overset{P}{\to} \theta^*(\mathcal{P})$ uniformly over $\mathcal{P} \in \mathbf{P}$, it is sufficient to show that the minimum eigenvalue of $\Sigma_T(\mathcal{P})^{-1/2} \frac{1}{\sqrt{T}} \sum_{t=1}^T W_t \phi(X_t, A_t)^{\otimes 2}$ goes to infinity uniformly over $\mathcal{P} \in \mathbf{P}$ as $T \to \infty$.

By Condition 11, the maximum eigenvalue of $\Sigma_T(\mathcal{P})$ is bounded uniformly over $\mathcal{P} \in \mathbf{P}$, so the minimum eigenvalue of $\Sigma_T(\mathcal{P})^{-1/2}$ is bounded uniformly above 0. Thus it is sufficient to show

that the minimum eigenvalue of $\frac{1}{\sqrt{T}}\sum_{t=1}^{T} W_t \phi(X_t, A_t)^{\otimes 2}$ goes to infinity uniformly over $\mathcal{P} \in \mathbf{P}$ as $T \to \infty$.

Note that by Lemma 1 and Condition 11,

$$\frac{1}{\sqrt{T}}\sum_{t=1}^{T} W_t \phi(X_t, A_t)^{\otimes 2} - \mathbb{E}_{\mathcal{P},\pi}\left[W_t \phi(X_t, A_t)^{\otimes 2} \big| \mathcal{H}_{t-1}\right] = O_{\mathcal{P}\in\mathbf{P}}(1). \tag{33}$$

Note that by law of iterated expectations,

$$\mathbb{E}_{\mathcal{P},\pi}\left[W_t \phi(X_t, A_t)^{\otimes 2} \big| \mathcal{H}_{t-1}\right]$$

$$= \mathbb{E}_{\mathcal{P}}\left[\int_{a\in\mathcal{A}} \pi_t(a, X_t, \mathcal{H}_{t-1})\mathbb{E}_{\mathcal{P}}\left[W_t \phi(X_t, A_t)^{\otimes 2} \big| \mathcal{H}_{t,1}, X_t, a\right] da \bigg| \mathcal{H}_{t-1}\right].$$

By Condition 1 and since $W_t = \sqrt{\frac{\pi_t^{\text{sta}}(A_t, X_t)}{\pi_t(A_t, X_t, \mathcal{H}_{t-1})}}$,

$$= \mathbb{E}_{\mathcal{P}}\left[\int_{a\in\mathcal{A}} \sqrt{\frac{\pi_t(a, X_t, \mathcal{H}_{t-1})}{\pi_t^{\text{sta}}(a, X_t)}} \pi_t^{\text{sta}}(a, X_t)\mathbb{E}_{\mathcal{P}}\left[\phi(X_t, A_t)^{\otimes 2} \big| X_t, a\right] da \bigg| \mathcal{H}_{t-1}\right]$$

Since by Condition 12, $\frac{\pi_t(a, X_t, \mathcal{H}_{t-1})}{\pi_t^{\text{sta}}(a, X_t)} \geq \frac{1}{\sqrt{\rho_{\max,T}}}$ and $\phi(X_t, A_t)^{\otimes 2} \succeq 0$,

$$\succeq \frac{1}{\sqrt{\rho_{\max,T}}}\mathbb{E}_{\mathcal{P}}\left[\int_{a\in\mathcal{A}} \pi_t^{\text{sta}}(a, X_t)\mathbb{E}_{\mathcal{P}}\left[\phi(X_t, A_t)^{\otimes 2} \big| X_t, a\right] da \bigg| \mathcal{H}_{t-1}\right].$$

Since $\pi_t^{\text{sta}}$ are pre-specified and since by our i.i.d. potential outcomes assumption (Condition 1) $X_t$ do not depend on $\mathcal{H}_{t-1}$,

$$= \frac{1}{\sqrt{\rho_{\max,T}}}\mathbb{E}_{\mathcal{P}}\left[\int_{a\in\mathcal{A}} \pi_t^{\text{sta}}(a, X_t)\mathbb{E}_{\mathcal{P}}\left[\phi(X_t, A_t)^{\otimes 2} \big| X_t, a\right] da\right].$$

By law of iterated expectations,

$$= \frac{1}{\sqrt{\rho_{\max,T}}}\mathbb{E}_{\mathcal{P},\pi_t^{\text{sta}}}\left[\phi(X_t, A_t)^{\otimes 2}\right].$$

The above result and Equation (33) implies that

$$\frac{1}{\sqrt{T}}\sum_{t=1}^{T} W_t \phi(X_t, A_t)^{\otimes 2} \succeq O_{\mathcal{P}\in\mathbf{P}}(1) + \sqrt{\frac{T}{\rho_{\max,T}}}\frac{1}{T}\sum_{t=1}^{T}\mathbb{E}_{\mathcal{P},\pi_t^{\text{sta}}}\left[\phi(X_t, A_t)^{\otimes 2}\right]. \tag{34}$$

By Condition 11, the minimum eigenvalue of $\frac{1}{T}\sum_{t=1}^{T}\mathbb{E}_{\mathcal{P},\pi_t^{\text{sta}}}\left[\phi(X_t, A_t)^{\otimes 2}\right]$ is bounded above some constant greater than zero for all $\mathcal{P} \in \mathbf{P}$. By Condition 12, $\sqrt{\frac{T}{\rho_{\max,T}}} \to \infty$. Thus by Equation (32) and Equation (34), we have that $\hat{\theta}_T^{\text{AW-LS}} \xrightarrow{P} \theta^*(\mathcal{P})$ uniformly over $\mathcal{P} \in \mathbf{P}$.

## C   Choice of Stabilizing Policy

### C.1   Optimal Stabilizing Policy in Multi-Arm Bandit Setting

Here we consider the multi-armed bandit setting where $\mathbb{E}_{\mathcal{P}}[Y_t(a)] = \theta_a^*(\mathcal{P})$ and $\mathrm{Var}_{\mathcal{P}}(Y_t(a)) = \sigma^2$. We consider the adaptively-weighted least-squares estimator where $m_\theta(Y_t, A_t) = -\mathbb{1}_{A_t=a}(Y_t - \theta_a^*(\mathcal{P}))^2$. By Theorem 1, we have that

$$\left( \frac{1}{T} \sum_{t=1}^{T} \mathbb{E}_{\mathcal{P}, \pi_t^{\mathrm{sta}}} \left[ \mathbb{1}_{A_t=a}(Y_t - \theta_a^*(\mathcal{P}))^2 \right] \right)^{-1/2} \left( \frac{1}{T} \sum_{t=1}^{T} W_t \mathbb{1}_{A_t=a} \right) \sqrt{T}(\hat{\theta}_{T,a}^{\mathrm{AW\text{-}LS}} - \theta_a^*(\mathcal{P})) \overset{D}{\to} \mathcal{N}(0,1).$$

While the asymptotic variance of $\sqrt{T}(\hat{\theta}_{T,a}^{\mathrm{AW\text{-}LS}} - \theta_a^*(\mathcal{P}))$ does not necessarily concentrate we can examine the following:

$$\left( \frac{1}{T} \sum_{t=1}^{T} W_t \mathbb{1}_{A_t=a} \right)^{-1} \left( \frac{1}{T} \sum_{t=1}^{T} \mathbb{E}_{\mathcal{P}, \pi_t^{\mathrm{sta}}} \left[ \mathbb{1}_{A_t=a}(Y_t - \theta_a^*(\mathcal{P}))^2 \right] \right) \left( \frac{1}{T} \sum_{t=1}^{T} W_t \mathbb{1}_{A_t=a} \right)^{-1}$$

By Lemma 1, we have that $\frac{1}{T} \sum_{t=1}^{T} W_t \mathbb{1}_{A_t=a} - \sqrt{\pi_t^{\mathrm{sta}}(a) \pi_t(A_t, \mathcal{H}_{t-1})} \overset{P}{\to} 0$. Thus we have

$$= \left( \frac{1}{T} \sum_{t=1}^{T} \pi_t^{\mathrm{sta}}(a) \sigma^2 \right) \left( o_p(1) + \frac{1}{T} \sum_{t=1}^{T} \sqrt{\pi_t^{\mathrm{sta}}(a) \pi_t(A_t, \mathcal{H}_{t-1})} \right)^{-2}.$$

As long as $\pi_t^{\mathrm{sta}}(a), \pi_t(A_t, \mathcal{H}_{t-1})$ are bounded away from zero w.p. 1, the $o_p(1)$ term is asymptotically negligible and we can just consider $\left( \frac{1}{T} \sum_{t=1}^{T} \pi_t^{\mathrm{sta}}(a) \sigma^2 \right) \left( \frac{1}{T} \sum_{t=1}^{T} \sqrt{\pi_t^{\mathrm{sta}}(a) \pi_t(A_t, \mathcal{H}_{t-1})} \right)^{-2}$.

By Cauchy-Schwartz inequality,
$$\left( \frac{1}{T} \sum_{t=1}^{T} \sqrt{\pi_t^{\mathrm{sta}}(a) \pi_t(a, \mathcal{H}_{t-1})} \right)^2 \leq \left( \frac{1}{T} \sum_{t=1}^{T} \pi_t^{\mathrm{sta}}(a) \right) \left( \frac{1}{T} \sum_{t=1}^{T} \pi_t(a, \mathcal{H}_{t-1}) \right).$$

Thus, $\frac{1}{\frac{1}{T} \sum_{t=1}^{T} \pi_t(a, \mathcal{H}_{t-1})} \leq \frac{\frac{1}{T} \sum_{t=1}^{T} \pi_t^{\mathrm{sta}}(a)}{\left( \frac{1}{T} \sum_{t=1}^{T} \sqrt{\pi_t^{\mathrm{sta}}(a) \pi_t(a, \mathcal{H}_{t-1})} \right)^2}$, so

$$\frac{\frac{1}{T} \sum_{t=1}^{T} \pi_t^{\mathrm{sta}}(a)}{\left( \frac{1}{T} \sum_{t=1}^{T} \sqrt{\pi_t(a, \mathcal{H}_{t-1}) \pi_t^{\mathrm{sta}}(a)} \right)^2} \geq \frac{1}{\frac{1}{T} \sum_{t=1}^{T} \pi_t(a, \mathcal{H}_{t-1})}.$$

Note that this lower bound is achieved when $\pi_t^{\mathrm{sta}}(a) = \pi_t(a)$. However, since $\pi_t$ is a function of $\mathcal{H}_{t-1}$ and stabilizing policies $\{\pi_t^{\mathrm{sta}}\}_{t=1}^{T}$ are pre-specified, setting $\pi_t^{\mathrm{sta}}(A_t) = \pi_{t,a}$ is generally an unfeasible choice. Thus we want to choose $\pi_t^{\mathrm{sta}}$ to be as close to $\pi_t$ as possible, subject to the constraint that the stabilizing policies are pre-specified, i.e., not a function of the data $\{Y_t, X_t, A_t\}_{t \geq 1}$.

### C.2   Approximating the Optimal Stabilizing Policy

One way to approximately choose the optimal evaluation policy is to select $\pi_t^{\mathrm{sta}}(a, x) = \mathbb{E}_{\mathcal{P}, \pi}[\pi_t(a, x, \mathcal{H}_{t-1})]$. Note that $\mathbb{E}_{\mathcal{P}, \pi}[\pi_t(a, x, \mathcal{H}_{t-1})]$ depends on the $\mathcal{P}$, which is unknown. Thus it is natural to choose $\pi_t^{\mathrm{sta}}(a, x)$ to be $\mathbb{E}_{\mathcal{P}, \pi}[\pi_t(a, x, \mathcal{H}_{t-1})]$ weighted by a prior on $\mathcal{P}$. Note that as long as the evaluation policy ensures that weights $W_t$ are bounded, the choice of evaluation policy does not affect the asymptotic validity of the estimator.

In Figure 6, we display the difference in mean squared error for the AW-LS estimator in a two-armed bandit setting for two different choices of evaluation policy: (1) the uniform evaluation policy which selects actions uniformly from $\mathcal{A}$ and (2) the expected $\pi_t(a, \mathcal{H}_{t-1})$ evaluation policy for which $\pi_t^{\mathrm{sta}}(a) = \mathbb{E}_{\mathcal{P}, \pi}[\pi_t(a, \mathcal{H}_{t-1})]$. We can see in this setting that by setting $\pi_t^{\mathrm{sta}}(a) = \mathbb{E}_{\mathcal{P}, \pi}[\pi_t(a, \mathcal{H}_{t-1})]$ we are able to decrease the mean squared error of the AW-LS estimator compared AW-LS with the uniform evaluation policy. Note though that in some cases setting $\pi_t^{\mathrm{sta}}(a) = \mathbb{E}_{\mathcal{P}, \pi}[\pi_t(a, \mathcal{H}_{t-1})]$ is equivalent to choosing the uniform evaluation policy. For example, a two-armed bandit with identical arms so under common bandit algorithms $\mathbb{E}_{\mathcal{P}, \pi}[\pi_t(a, \mathcal{H}_{t-1})] = 0.5$ for all $t \in [1:T]$, which will make the evaluation policy $\pi_t^{\mathrm{sta}}(a) = \mathbb{E}_{\mathcal{P}, \pi}[\pi_t(a, \mathcal{H}_{t-1})]$ equivalent to the uniform policy.

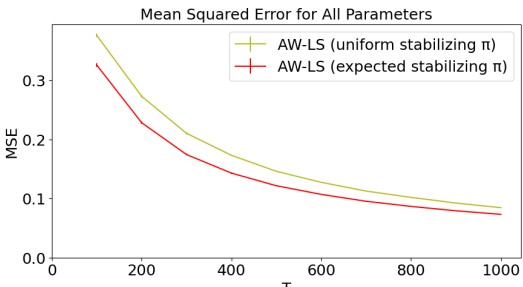

Figure 6: Above we plot the mean squared errors for the adaptively-weighted least squares estimator with evaluation policies: (1) uniform evaluation policy which selects actions uniformly from $\mathcal{A}$ and (2) expected $\pi_t(a, \mathcal{H}_{t-1})$ evaluation policy for which $\pi_t^{\mathrm{sta}}(a) = \mathbb{E}_{\mathcal{P},\pi}\left[\pi_t(a)\right]$ (oracle quantity). In a two arm bandit setting we perform Thompson Sampling with standard normal priors, 0.01 clipping, $\theta^*(\mathcal{P}) = [\theta_0^*(\mathcal{P}), \theta_1^*(\mathcal{P})] = [0, 1]$, standard normal errors, and $T = 1000$. Error bars denote standard errors computed over 5,000 Monte Carlo simulations.

# D  Need for Uniformly Valid Inference on Data Collected with Bandit Algorithms

Here we consider the two-armed bandit setting where $\mathbb{E}_{\mathcal{P}}[R_t(a)] = \theta_{0,a}(\mathcal{P})$, $\mathrm{Var}_{\mathcal{P}}(R_t(a)) = \sigma^2$, and $\mathbb{E}_{\mathcal{P}}[R_t(a)^4] < c < \infty$ for $a \in \{0, 1\}$. The unweighted least squares estimator is asymptotically normal on adaptively collected data under the following condition of Lai and Wei [1982], there exists a non-random sequence $\{b_t\}_{t \geq 1}$ such that

$$b_T \cdot \sum_{t=1}^{T} A_t \xrightarrow{P} 1. \tag{35}$$

Specifically, by Theorem 3 of Lai and Wei [1982], under (35),

$$\sqrt{\sum_{t=1}^{T} A_t}(\hat{\theta}_{T,1}^{\mathrm{OLS}} - \theta_1^*(\mathcal{P})) = \frac{\sum_{t=1}^{T} A_t(R_t - \theta_1^*(\mathcal{P}))}{\sqrt{\sum_{t=1}^{T} A_t}} \xrightarrow{D} \mathcal{N}(0, \sigma^2).$$

However, as discussed in Deshpande et al. [2018] and Zhang et al. [2020], (35) can fail to to hold for common bandit algorithms when there is no unique optimal policy, i.e., when $\theta_0^*(\mathcal{P}) - \theta_1^*(\mathcal{P}) = 0$. For example, in Figure 7 we plot $\frac{1}{T} \sum_{t=1}^{T} A_t$ for Thompson Sampling and $\epsilon$-greedy for a bandit with two identical arms.

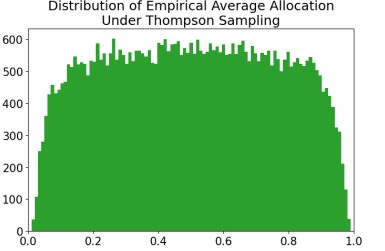 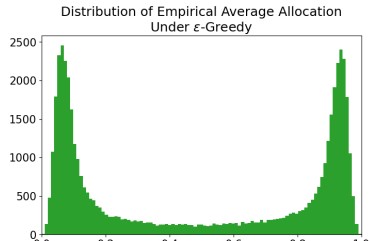

Figure 7: Above we plot empirical allocations, $\frac{1}{T} \sum_{t=1}^{T} A_t$, under both Thompson Sampling (standard normal priors, 0.01 clipping) and $\epsilon$-greedy ($\epsilon = 0.1$) under zero margin $\theta_0^*(\mathcal{P}) = \theta_1^*(\mathcal{P}) = 0$. For our simulations $T = 100$, errors are standard normal, and we use $50k$ Monte Carlo repetitions.

In order to construct reliable confidence intervals using asymptotic approximations, it is crucial that that estimators converge uniformly in distribution. To illustrate the importance of uniformity, consider the following example. We can modify Thompson Sampling to ensure that $\frac{1}{T} \sum_{t=1}^{T} A_t \xrightarrow{P} 0.5$ when $\theta_1^*(\mathcal{P}) - \theta_0^*(\mathcal{P}) = 0$. For example, we could do this by using an algorithm we call Thompson Sampling Hodges (inspired by the Hodges estimator; see Van der Vaart [2000, Page 109]), defined below:

$$\pi_t(1, \mathcal{H}_{t-1}) = \mathbb{P}(\tilde{\theta}_1 > \tilde{\theta}_0 | \mathcal{H}_{t-1}) \mathbb{1}_{|\mu_{1,t} - \mu_{0,t}| > t^{-4}} + 0.5 \mathbb{1}_{|\mu_{1,t} - \mu_{0,t}| \leq t^{-4}}$$

Under standard Thompson Sampling arm one is chosen according to the posterior probability that is optimal, so $\pi_t(1, \mathcal{H}_{t-1}) = \mathbb{P}(\tilde{\theta}_1 > \tilde{\theta}_0 | \mathcal{H}_{t-1})$. Above, $\mu_{a,t}$ denotes the posterior mean for the mean reward for arm $a$ at time $t$. Under TS-Hodges, if difference between the posterior means, $|\mu_{1,t} - \mu_{0,t}|$, is less than $t^{-4}$, $\pi_t$ is set to 0.5. Additionally, we clip the action selection probabilities to bound them strictly away from 0 and 1 for some constant $\pi_{\min}$ in the following sense $\mathrm{clip}(\pi_t) = (1 - \pi_{\min}) \wedge (\pi_t \vee \pi_{\min})$. Under TS-Hodges with clipping, we can show that

$$\frac{1}{T} \sum_{t=1}^{T} A_t \xrightarrow{P} \begin{cases} 1 - \pi_{\min} & \text{if } \theta_1^*(\mathcal{P}) - \theta_0^*(\mathcal{P}) > 0 \\ \pi_{\min} & \text{if } \theta_1^*(\mathcal{P}) - \theta_0^*(\mathcal{P}) < 0 \\ 0.5 & \text{if } \theta_1^*(\mathcal{P}) - \theta_0^*(\mathcal{P}) = 0 \end{cases} \tag{36}$$

By equation (36), we satisfy (35) pointwise for every fixed $\mathcal{P}$ and we have that the OLS estimator is asymptotically normal *pointwise* [Lai and Wei, 1982]. However, equation (36) fails to hold uniformly over $\mathcal{P} \in \mathbf{P}$. Specifically, it fails to hold for any sequence of $\{\mathcal{P}_t\}_{t=1}^{\infty}$ such that $\theta_1^*(\mathcal{P}_t) - \theta_0^*(\mathcal{P}_t) = t^{-4}$.

In Figure 8, we show that confidence intervals constructed using normal approximations fail to provide reliable confidence intervals, even for very large sample sizes for the worst case values of $\theta_1^*(\mathcal{P}) - \theta_0^*(\mathcal{P})$.

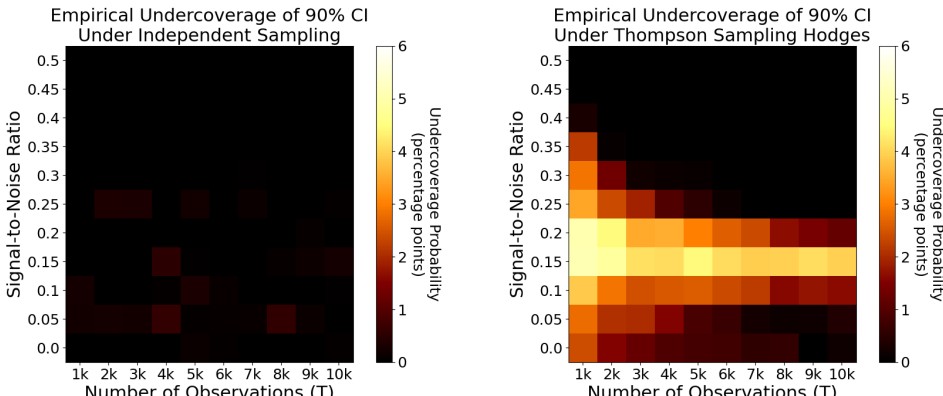

Figure 8: Above we construct confidence intervals for $\theta_1^*(\mathcal{P}) - \theta_0^*(\mathcal{P})$ using a normal approximation for the OLS estimator. We compare independent sampling ($\pi_t = 0.5$) and TS Hodges, both with standard normal priors, $0.01$ clipping, standard normal errors, and $T = 10,000$. We vary the value of $\theta_1^*(\mathcal{P}) - \theta_0^*(\mathcal{P})$ in the simulations to demonstrate the non-uniformity of the confidence intervals.

# E    Discussion of Chen et al. [2020]

Here we show formally that Theorem 3.1 in Chen et al. [2020], which proves that the OLS estimator is asymptotically normal on data collected with an $\epsilon$-greedy algorithm, does not cover the case in which there is no unique optimal policy.

They assume that for rewards $R_t$, context vectors $X_t$, and binary actions $A_t \in \{0, 1\}$,

$$\mathbb{E}[R_t | X_t, A_t] = A_t \mathbf{X}_t^\top \boldsymbol{\beta}_1 + (1 - A_t) \mathbf{X}_t^\top \boldsymbol{\beta}_0.$$

They define $\boldsymbol{\beta} := \boldsymbol{\beta}_1 - \boldsymbol{\beta}_0$.

Specifically at part 1(b) of their proof on page 4 of the supplementary material, they claim that $g(\hat{\boldsymbol{\beta}}_t, \epsilon) \xrightarrow{P} g(\boldsymbol{\beta}, \epsilon)$, where $\hat{\boldsymbol{\beta}}_t$ is the OLS estimator for $\boldsymbol{\beta} := \boldsymbol{\beta}_1 - \boldsymbol{\beta}_0$ and $g$ is defined as follows:

$$g(\boldsymbol{\beta}_0, \boldsymbol{\beta}_1, \epsilon) = \frac{\epsilon}{2} \int \mathbf{v}^\top \mathbf{x} \mathbf{x}^\top \mathbf{v} d\mathcal{P}_x + (1 - \epsilon) \int \mathbb{1}_{\boldsymbol{\beta}^\top \mathbf{x} \geq 0} \mathbf{v}^\top \mathbf{x} \mathbf{x}^\top \mathbf{v} d\mathcal{P}_x$$

Above $v \in \mathbb{R}^d$ is arbitrary fixed vector and $x \in \mathbb{R}^d$ are the context vectors. $\mathcal{P}_x$ is the distribution of the context vectors $\boldsymbol{X}_t$.

Specifically, they claim that $g(\hat{\boldsymbol{\beta}}_t, \epsilon) \xrightarrow{P} g(\boldsymbol{\beta}, \epsilon)$ because $\hat{\boldsymbol{\beta}}_t \xrightarrow{P} \boldsymbol{\beta}$ (Corollary 3.1) and by continuous mapping theorem.

Recall the continuous mapping theorem for convergence in probability [Van der Vaart, 2000, Theorem 2.3]:

**Theorem 4** (Continuous Mapping Theorem). *Let $g : \mathbb{R}^k \to \mathbb{R}^m$ be continuous at every point of a set $C$ such that $\mathbb{P}(X \in C) = 1$. If $X_n \xrightarrow{P} X$, then $g(X_n) \xrightarrow{P} g(X)$.*

Note that $g$ is not continuous in $\boldsymbol{\beta}$ at the value $\boldsymbol{\beta} = \mathbf{0} \in \mathbb{R}^d$; this is due to the indicator term $\mathbb{1}_{\boldsymbol{\beta}^\top \mathbf{x} \geq 0}$. Thus, the standard continuous mapping theorem can not be applied in this setting. Note that the case that $0 = \boldsymbol{\beta} = \boldsymbol{\beta}_1 - \boldsymbol{\beta}_0$, is exactly when there is no unique optimal policy. This means that Theorem 3.1 in Chen et al. [2020] does not cover the setting in which there is no unique optimal policy.