# OpenReview forum: "Statistical Inference with M-Estimators on Adaptively Collected Data"
_NeurIPS.cc/2021/Conference — NeurIPS 2021 Poster_

### Official Review · Reviewer_55GN · 2021-06-27

**Rating:** 6
**Confidence:** 5

**Summary:**

The authors analyze square root importance sampling weighted M-estimators over parametric classes for adaptively collected data.

This work is motivated by the fact that the usual unweigthed M-estimators that one would use for i.i.d. data fail to be asymptotically normally distributed under adaptively collected data.

Under appropriate conditions, the proposed weighting scheme makes the resulting M-estimator asymptotically normal, uniformly over the parameter space.

The authors demonstrate their claims through simulations.

**Limitations And Societal Impact:**

It seems to me that condition 7 easily breaks when the reward model is incorrectly specified.

(Say for instance that $m_\theta(x,a,y) = -(y - \theta^\top_a x)^2$. Suppose that $ X_1^{(1)},...,X_d^{(1)} $ and $ X_1^{(2)},...,X_d^{(2)} $ are two subsequences of the context sequence corresponding to rounds with $A_t = 1$, such that the corresponding empirical covariance matrices are full-rank, Then, if condition 7 holds, for $a=1$, $\theta^*_{a, \mathcal{P}}$ will be a minimizer of both

$$ \sum_{s=1}^d E[(Y^{(1)}_s - \theta^\top_a X^{(1)}_s | X^{(1)}_s]$$

and

$$ \sum_{s=1}^d E[(Y^{(2)}_s - \theta^\top_a X^{(2)}_s | X^{(2)}_s].$$

Since the design matrices $(X_1^{(1)},...,X_d^{(1)})^\top $ and $(X_1^{(2)},...,X_d^{(1)})^\top $
are full rank, the minimizers are unique, and are a priori different if the model is misspecified and the design matrices are distinct.)

I think the authors should clearly state that in general condition 7 does not hold under misspecification. I think this is the main caveat of this work which has serious practical implications, as parametric models are hardly ever correctly specified in practical applications.

This also raises another issue. Even if we had guarantees for inference on the coefficients of a misspecified parametric model, I think the practical usefulness of such guarantees of such results is limited. For instance this cannot be used to perform causal inference on the mean potential outcome of a given an arm, as no function of the coefficients equals that statistical parameter in general if the model is misspecified.


## Additional minor comments:

Line 297:
Why do these bounds need to be anytime ?  The asymptotic normality result stated in the article allows to construct an approximate CI at a given sample size, but if you wanted to give an approximate anytime confidence sequence, you’d need a stronger type of asymptotic distributional result (such as a weak invariance principle) that would characterize the joint asymptotic distribution of the sequence of estimators.
In that sense, it seems to me that the finite sample counterpart of your work is a concentration bound at a specific time point - not anytime.

Line 143-144 and 149-154. I think these identities would be better stated as lemmas and proven in appendix.

Line 187: There is an issue with the sentence.

Line 195: $\dot{m}$ and $\ddot{m}$ are not defined (although their meaning is clear)

Line 201: Condition 6: I think it'd be nice to define $\left\lVert B \right\rVert_1$ at the beginning of the assumption statement, not at the end, so that the reader doesn't run into this notation before it is defined.

Line 305: Minor comment: I think there is no need to introduce the additional notation $R_t$. You could just say that the outcome $Y_t$ has the interpretation of a reward.

**Main Review:**

- The paper is clearly written, even though I see some directions of improvement which I list further down.
- Inference from adaptively collected data is well motivated and of great practical significance.
- The contribution is novel to the best of my knowledge.
- Most conditions are well discussed.
- I've checked the proofs and they are correct.
- Simulations are compelling. In particular the existing baseline methods are clearly inferior and non-usable for inference.



### Significance:

The main weakness is that the guarantees do not seem to hold under model misspecification, which reduces the scope and practical applicability the contribution. I discuss this further in the next text box. I will update my rating depending on answers to this concern.



**Time Spent Reviewing:**

8

---

> ### Author Response · Authors · 2021-08-10
> **Reply to Official Review of Paper6140 by Reviewer 55GN**
>
> Thank you so much for taking the time to read our paper and for your thoughtful comments! We really appreciate that you think our paper is well-written and is of great practical significance.
>
> We completely agree that it would be very interesting to extend our work to more general settings for which Condition 7 may not hold. Condition 7 is restrictive, as it requires the model for the expected outcome to be correctly specified. For this reason, in our Discussion section, we discuss how it would be great to be able to extend our results to best "projected" solutions, i.e., the parameter that maximizes a given criterion regardless of whether the model is correctly specified. Additionally, in our Problem Formulation we explicitly introduce this correctly specified model assumption and say that it is an implicit modelling assumption (line 87); in the revision we can mention this again when introducing Condition 7.
>
> Regarding your minor comments:
> - Thank you for pointing out the typos and ways to improve the presentation / writing. We will make these changes in the revision.
> - Regarding your comments related to Line 297 in the Related Work section: The differences between the confidence regions we construct using asymptotic approximations vs. those constructed using high probability bounds differ in two respects (1) the high-probaiblity bounds hold in finite samples, and (2) the high-probability bounds used in the bandit literature are also often anytime. Given your comments we will reword this section to only focus on point (1) to avoid confusion, since this is the primary difference we wanted to highlight in comparison to our work.

---

### Official Review · Reviewer_yw7d · 2021-07-14

**Rating:** 7
**Confidence:** 5

**Summary:**

This paper extends classical M-estimator results to settings where the data are collected sequentially.

**Limitations And Societal Impact:**

yes

**Main Review:**

From a technical perspective, the contributions seem relatively unsurprising, combining classical M-estimator conditions with the notion of stabilizing weights arising in the bandit/optimal treatment literatures. That said, putting these technical arguments together correctly requires expertise that wouldn't be available to quite a few potential users of this approach. Also, the uniformity established by the authors is also useful for helping to ensure that the asymptotics will provide meaningful finite sample insights. So, I have an overall favorable view of this work.

I have a few comments/questions:

1. It would be worth describing cases where the condition that $\theta^*(P)$ exists for a given $m_\theta$ (below Eq. 1) is expected to hold, and also settings where it is not expected to hold. E.g., in a correctly specified parametric/semiparametric model is this always plausible provided that $\theta^*(P)$ is some (function) of the indexing parameter? How about more generally?

2. Along the same lines as above, could you develop your results for the case where the objective function for $\theta^*(\mathcal{P})$ is defined as some kind of average over possible values of $(X_t,A_t)$ or so, resulting in the estimand being some kind of least squares/KL projection that doesn't rely on the existence of a $\theta^*(P)$ that satisfies the almost sure assumption in Eq. 1?

3. The theoretical section allows the action space $\mathcal{A}$ to be finite or infinite, but then the experiments all focus on the case were $\mathcal{A}=\{0,1\}$. It would be helpful to see an example with a larger $\mathcal{A}$, to provide some assurance that the asymptotics provided are actually relevant at realistic sample sizes (the current uniform theoretical results show that a sample size exists at which the asymptotic results are relevant, but doesn't tell you what that sample size actually is).

Clarifying questions/comments:

1. On pg 5 line 192, it says "when $m_\theta$ is concave". Concave in what? In $\theta$ for a fixed $x,a,y$, or something else?

2. I think that Condition 7 should read: "For all $P\in\mathcal{P}$, there exists a $\theta^*(\mathcal{P})\in\Theta$ such that $\theta^*(\mathcal{P})\in\arg\max_{\theta\in\Theta}\mathbb{E}_{\mathcal{P}}[m_\theta(Y_t,X_t,A_t)|X_t,A_t]$ w.p. 1."

There also appears to be a typo at the bottom of page 2: $f\hat{\theta}$.

**Time Spent Reviewing:**

2.5

---

> ### Author Response · Authors · 2021-08-10
> **Reply to Official Review of Paper6140 by Reviewer yw7d**
>
> Thank you so much for taking the time to read our paper and for your thoughtful comments! We really appreciate that you have an overall favorable view of our work and that you thought our work would be useful to practitioners.
>
> Regarding the comments/questions that you had:
>
> - Regarding your first two questions: We completely agree that it would be very interesting to extend our work to more general settings in which equation (1) may not hold. Condition in equation (1) is relatively restricted, as it requires the model for the expected outcome to be correctly specified. For this reason, in our Discussion section, we discuss how it would be great to be able to extend our results to best "projected" solutions, i.e., the parameter that maximizes a given criterion regardless of whether the model is correctly specified.
> - We agree that additional simulations with more arms would be interesting. This is something we will consider for the revision.
> - Clarifying point 1: $m_\theta(Y, X, A)$ is a concave function of $\theta$ for all $Y, A, X$
> - Clarifying point 2: Thank you for this correction!

---

### Official Review · Reviewer_GB2D · 2021-07-15

**Rating:** 6
**Confidence:** 4

**Summary:**

This paper introduces an M-estimation-based method that provides valid inference for adaptively collected data. In particular, the contribution of the new inference technique includes: 1) it applies to a wide range of estimands that can be written as the maximizer of a loss function; 2) it provides valid inference in the presence of contexts; 3) uniform convergence is established.


**Limitations And Societal Impact:**

A discussion on the potential negative societal impact of the work should be added.


**Main Review:**

This is in general a well-written paper, and I find its contribution to the inference technique for adaptively collected data quite significant. I have two major questions regarding the paper:

1. It is assumed in Condition 9 that the ratio between \pi^{sta} and \pi_t is bounded from above and below by a constant. If, as the authors suggest at the beginning of Section 3, we take \pi^{sta} to be 1/|A|, this is essentially the overlap condition. This immediately assumes away many technical difficulties and does not seem to be very natural, since the main feature of the adaptively collected/bandit-collected data is that there are vanishing propensity scores. Is it possible to improve on this condition? If not, what are the difficulties?

2. I notice that there is a parallel paper (Zhan et al., 2021) that studies a similar problem. Their paper allows the propensity score to go to zero, but the results they provide are different from this paper. I am curious to see if there is any equivalence and difference between the results given by the two papers.

Reference:

Zhan, Ruohan, et al. "Off-Policy Evaluation via Adaptive Weighting with Data from Contextual Bandits." arXiv preprint arXiv:2106.02029 (2021).




**Time Spent Reviewing:**

3 hours

---

> ### Author Response · Authors · 2021-08-10
> **Reply to Official Review of Paper6140 by Reviewer GB2D**
>
> Thank you so much for taking the time to read our paper and for your thoughtful comments! We really appreciate that you found our paper to be well-written and that it makes a significant contribution to the area.
>
> Regarding the questions that you had:
>
> 1. We agree that allowing the action selection probabilities go to zero is interesting (note, we discuss this question in our Discussion section). We were able show that we can allow the action selection probabilities go to zero for the adaptively-weighted least squares estimator (see Appendix B.5 for more details) and believe that perhaps with more sophisticated mathematical techniques / slightly different assumptions it is possible to show this for more general estimators as well. We found it in general challenging to show consistency of the weighted M-estimators when allowing the action selection probabilities go to zero; we conjecture we could get around this issue by using methods recently proposed in this recent paper (https://arxiv.org/abs/2106.01723) on empirical risk minimization on bandit data, which is also being reviewed at NeurIPS 2021.
> 2. Regarding the relationship of our work to that of off-policy evaluation on adaptively collected data, the main difference is that in off-policy evaluation the estimand is the value under a pre-specified policy (note this is a scalar value). Instead we are interested in constructing confidence regions for parameters of a model for an outcome (that could be the reward)---for example, this could be parameters of a logistic regression model for a binary outcome. We believe that there could be theory in the future that could unify these methods for these different estimands.
> 3. Regarding potential negative societal impacts, in our next version we will write about how someone might apply our methods without understanding the necessary assumptions, which could lead to inaccurate inference.

---

### Official Review · Reviewer_H36N · 2021-07-17

**Rating:** 7
**Confidence:** 4

**Summary:**

This paper studies how to conduct statistical inference with M-estimators based on data collected from adaptive algorithms. It proposes an adaptively weighted M-estimator with square-root importance weights and proves that under fairly standard assumptions, the estimator (uniformly) has asymptotic normal distribution and it can be used to construct uniformly valid confidence intervals. By studying M-estimation problems, this paper substantially broadens the scope of estimation problems in this recent literature of statistical inference with adaptively collected data.

**Limitations And Societal Impact:**

Yes. They discuss the limitations in their section 6.

**Main Review:**

This paper studies statistical inference with adaptively collected data, a very important problem that attracts a lot of recent attention. I think this paper's study of general M-estimation problems, not just only simple expectations or linear models, makes a significant contribution in this area. Moreover, this paper is well-written, and it is overall easy to follow. I particularly section 3.1 about the intuition for the proposed weights, which is really helpful, and section 6 that discusses many questions I had while I was reading sections before this part.

I only have some minor comments:

1. eq (7) doesn't look right: it seems that from equalities (a) on, there shouldn't be an indicator function (At = a), right? It should cancel with the inverse propensity weights in the denominator, and instead the inner conditional expectation should be conditioned on (At = a) instead.

2. I'm a bit confused by condition 7 (i) and condition 8. The authors state that the parameter thata^* is not necessarily unique sol to the conditional maximization problem, but condition 8 assumes an identification condition, which usually needs uniqueness. How should I understand this? So does this mean that even though theta^* is nonunique, the stablized expectation in condition (8) somehow rules out all of them but only a single identifiable one? If this is the case, then I do not think it is very plausible, although I get that the current presentation is in some sense more general than assuming unique solution to the conditional maximization problem.

**Time Spent Reviewing:**

1.5

---

> ### Author Response · Authors · 2021-08-10
> **Reply to Official Review of Paper6140 by Reviewer H36N**
>
> Thank you so much for taking the time to read our paper and for your thoughtful comments! We really appreciate that you found our paper to be well-written and that it makes a significant contribution to the area.
>
> Regarding the minor comments that you had:
>
> 1. The step we take in equality (a) actually can be thought of as $A_t = a$ in the intermediate steps (we write out in more detail below for clarity). Overall, equality (a) is changing the measure of the expectation from that of $\pi_t$ to $\pi^{sta}$.
>
> Recall the first line of equation (7):
> $$\mathbb{E_{\mathcal{P}}} \left[ \mathbb{E_{\mathcal{P}}} \left[ \frac{\pi^{\mathrm{sta}}(A_t, X_t)}{\pi_t(A_t, X_t, \mathcal{H_{t-1}})} 1_{A_t=a} X_t X_t^\top \left( Y_t - X_t^\top \theta_a^*(\mathcal{P}) \right)^2 \bigg| \mathcal{H_{t-1}}, X_t \right] \bigg| \mathcal{H_{t-1}} \right]$$
>
> The inner expectation above can be simplified as follows:
> $$\mathbb{E_{\mathcal{P}}} \left[ \frac{\pi^{\mathrm{sta}}(A_t, X_t)}{\pi_t(A_t, X_t, \mathcal{H_{t-1}})} 1_{A_t=a} X_t X_t^\top \left( Y_t - X_t^\top \theta_a^*(\mathcal{P}) \right)^2 \bigg| \mathcal{H_{t-1}}, X_t \right]$$
>
> By law of total expectation,
> $$= \sum_{a' \in \mathcal{A}} \pi_t(a', X_t, \mathcal{H_{t-1}}) \mathbb{E_{\mathcal{P}}} \left[ \frac{\pi^{\mathrm{sta}}(A_t, X_t)}{\pi_t(A_t, X_t, \mathcal{H_{t-1}})} 1_{A_t=a} X_t X_t^\top \left( Y_t - X_t^\top \theta_a^*(\mathcal{P}) \right)^2 \bigg| \mathcal{H}_{t-1}, X_t, A_t = a' \right]$$
>
> $$= \sum_{a' \in \mathcal{A}} \pi_t(a', X_t, \mathcal{H_{t-1}})  \frac{\pi^{\mathrm{sta}}(a', X_t)}{\pi_t(a', X_t, \mathcal{H_{t-1}})} \mathbb{E_{\mathcal{P}}} \left[ 1_{A_t=a} X_t X_t^\top \left( Y_t - X_t^\top \theta_a^*(\mathcal{P}) \right)^2 \bigg| \mathcal{H}_{t-1}, X_t, A_t = a' \right]$$
>
> $$= \sum_{a' \in \mathcal{A}} \pi^{\mathrm{sta}}(a', X_t) \mathbb{E_{\mathcal{P}}} \left[  1_{A_t=a} X_t X_t^\top \left( Y_t - X_t^\top \theta_a^*(\mathcal{P}) \right)^2 \bigg| \mathcal{H_{t-1}}, X_t, A_t = a' \right]$$
>
> By law of total expectation again,
> $$= \mathbb{E_{\mathcal{P}, \pi^{\mathrm{sta}}}} \left[ 1_{A_t=a} X_t X_t^\top \left( Y_t - X_t^\top \theta_a^*(\mathcal{P}) \right)^2 \bigg| \mathcal{H}_{t-1}, X_t \right].$$
>
> The above term is in the second line in equation (7) after equality (a). We can clarify these steps in the revision.
>
> 2. The difference between Condition 7(i) and Condition 8 is that the former is a conditional statement (conditional on $X_t, A_t$) and the latter is a marginal statement (marginal over $X_t, A_t$---specifically according to the distribution when $A_t$ is chosen according to stabilizing policies $\pi_t^{sta}$). Condition 7(i) is saying there may be a non-unique solution for some contexts $X_t$ and actions $A_t$, however Condition 8 assumes that marginally over $X_t, A_t$ there is a well-separated solution. We will clarify this difference in the revision.

---

### Decision · Program_Chairs · 2021-09-27

**Decision:**

Accept (Poster)

**Comment:**

The expert reviewers all appreciated the paper and agree it provides useful results and that the paper should be accepted. The authors are to be congratulated for an interesting contribution that nicely handles a very timely topic. The authors are expected to address the points raised by reviewers in a final version, including as they outlined in their response. In my opinion the most important thing to address is to make very clear early on the setting where the method is appropriate, namely well-specified parametric models, and that this excludes, for example, a constant model for the mean reward of an arm or policy. This is not to detract from the work -- rather, making clear what the paper sets out to do and what it does not will significantly improve clarity for the reader and therefore the paper's impact. A discussion regarding what would actually happen under misspecification (and whether slight misspecification results in only slight aberrations from the results) would potentially add a lot too (I believe the target estimand would then depend on the logging policies, if I understand correctly).